# Posterior Sampling Reinforcement Learning with Gaussian Processes for Continuous Control: Sublinear Regret Bounds for Unbounded State Spaces

**Hamish Flynn** [1]   **Joe Watson** [2 3 4]   **Ingmar Posner** [2]   **Jan Peters** [3 4 5 6]

## Abstract

We analyze the Bayesian regret of the Gaussian process posterior sampling reinforcement learning (GP-PSRL) algorithm. Posterior sampling is a heuristic for decision-making under uncertainty that has been used to develop successful algorithms for a variety of continuous control problems. However, theoretical work on GP-PSRL is limited. All known regret bounds either have a sub-optimal growth rate, require strong smoothness assumptions, or fail to properly account for the fact that the set of possible system states is unbounded. Through a recursive application of the Borell-Tsirelson-Ibragimov-Sudakov inequality, we show that, with high probability, the states actually visited by the algorithm are contained within a ball of near-constant radius. We then use the chaining method to control the regret suffered by GP-PSRL under weak smoothness conditions. Our main result is a Bayesian regret bound of the order $\widetilde{\mathcal{O}}(H\sqrt{\gamma_T T})$, where $H$ is the horizon, $T$ is the number of time steps and $\gamma_T$ is the expected information gain. With this result, we resolve the limitations with prior theoretical work on PSRL, and provide the theoretical foundation and tools for analyzing PSRL in complex settings.

## 1. Introduction

The study of decision-making under uncertainty deals with the exploration-exploitation trade-off using statistical techniques. Posterior sampling, also known as Thompson sampling or probability matching (Thompson, 1933), is an approach based on random sampling from a belief over optimal decisions. Despite its heuristic origins, posterior sampling is now understood to be theoretically sound (Russo & Van Roy, 2014). Compared to other principled methods, such as "optimistic" approaches, posterior sampling usually performs better and is often simpler to implement, especially when conjugate priors are available (Osband & Van Roy, 2017).

In the posterior sampling reinforcement learning (PSRL) algorithm, a belief is maintained over Markov decision processes (MDPs) and an optimal control oracle is used to compute an optimal policy for a sampled MDP in an episodic fashion, resulting in random exploration over the space of possible optimal policies. For MDPs with continuous states and actions, Gaussian processes can be used as a tractable yet versatile prior belief over MDPs. However, while posterior sampling has been shown to enjoy strong theoretical performance guarantees across a range of decision-making problems, theoretical work on PSRL with Gaussian processes is limited. We identify several limitations with previous results. Individually, some of these problems have been (partially) addressed in previous work (cf. Appendix B).

**Problem one: unbounded state spaces.** In the setting that we consider, the system states are corrupted by Gaussian noise, which means that the set of possible states is unbounded. If this is not properly accounted for, kernel-dependent quantities such as the maximum information gain (cf. Section 3.4) can grow linearly with the number of time steps. In addition, arguments used to control suprema of Gaussian processes fail when the Gaussian process is defined on an unbounded domain. To obtain rigorous theoretical guarantees, it is necessary to show that the states actually encountered by PSRL lie within a bounded subset of the state space with high probability.

**Problem two: sub-optimal rates.** Most regret bounds for GP-PSRL use an argument due to Osband et al. (2013), in which one constructs confidence sets that contain the true MDP and all the sampled MDPs. One can then show that the Bayesian regret of PSRL is as good as the worst-case regret of any optimistic algorithm. If tight confidence sets are available, this argument can yield near-optimal regret bounds. However, due to the difficulty of constructing confidence sets for functions in reproducing kernel Hilbert spaces (RKHSs) (Lattimore, 2023), when applied to GP-PSRL, this

[1] Carnegie Mellon University [2] University of Oxford [3] Technical University of Darmstadt [4] DFKI [5] hessian.AI [6] Robotics Institute Germany. Correspondence to: Hamish Flynn <hamishflynn.gm@gmail.com>, Joe Watson <joewatson@robots.ox.ac.uk>.

*Proceedings of the 43rd International Conference on Machine Learning*, Seoul, South Korea. PMLR 306, 2026. Copyright 2026 by the author(s).

| | Algorithm | Regret Bound | Regret Type | Prior/Model Class |
|---|---|---|---|---|
| Theorem 1 in Osband & Van Roy (2014) | PSRL | $\widetilde{\mathcal{O}}(L\Gamma_T\sqrt{T})$ | Bayesian | RKHS ball |
| Theorem 1 in Chowdhury & Gopalan (2019) | UCB | $\widetilde{\mathcal{O}}(L\Gamma_{d_sT}\sqrt{T})$ | Worst-case | RKHS ball |
| Theorem 2 in Chowdhury & Gopalan (2019) | PSRL | $\widetilde{\mathcal{O}}(L\Gamma_{d_sT}\sqrt{T})$ | Bayesian | RKHS ball |
| Theorem 3 in Chowdhury & Gopalan (2019) | UCB | $\widetilde{\mathcal{O}}(Le^{\Gamma_{d_sH}}\sqrt{\Gamma_{d_sT}T})$ | Bayesian | GP |
| Theorem 4 in Chowdhury & Gopalan (2019) | PSRL | $\widetilde{\mathcal{O}}(Le^{\Gamma_{d_sH}}\sqrt{\Gamma_{d_sT}T})$ | Bayesian | GP |
| Theorem 3.2 in Kakade et al. (2020) | UCB | $\widetilde{\mathcal{O}}(H\gamma_T\sqrt{T})$ | Worst-case | RKHS ball |
| Theorem 3 in Curi et al. (2020) | UCB | $\widetilde{\mathcal{O}}(H^{3/2}\Gamma_T^{(H+1)/2}\sqrt{T})$ | Worst-case | RKHS ball |
| Theorem 1 in Fan & Ming (2021) | PSRL | $\widetilde{\mathcal{O}}(H^{3/2}\Gamma_N\sqrt{T})$ | Bayesian | GP |
| Theorem 4.11 (ours) | PSRL | $\widetilde{\mathcal{O}}(H\sqrt{\gamma_T T})$ | Bayesian | GP |

*Table 1.* State-of-the-art regret bounds for PSRL and UCB algorithms in the setting that we consider (cf. Section 3). We only show the dependence of the regret on $H$ and $T$, and suppress all polylogarithmic factors. Here, $L$ is a problem-dependent quantity which hides dependence on $H$ (cf. Equation 3 in Osband & Van Roy, 2014). Since the eluder dimension and the maximum information gain ($\Gamma_T$) are equivalent for RKHSs (Huang et al., 2021), we state the regret bound from Theorem 1 of Osband & Van Roy (2014) in terms of the maximum information gain. The expected information gain $\gamma_T$ is an instance-dependent quantity with a growth rate that is never worse than that of $\Gamma_T$ (cf. Section 3.4). For any of the Bayesian regret bounds, the prior "RKHS ball" indicates that the prior is allowed to be any distribution with support contained within a ball of an RKHS.

approach results in regret bounds that have at least linear dependence on the maximum information gain.

**Problem three: limited priors.** Many existing regret bounds allow for limited choices of the prior. Approaches based on the "PSRL is as good as optimism" argument mentioned previously only allow for priors with support contained within a ball of the RKHS associated with the kernel function. In particular, this does not include Gaussian process priors. Other works have used bounds for suprema of Gaussian processes to upper bound the regret of GP-PSRL (cf. Theorem 4 in Chowdhury & Gopalan, 2019). While these approaches do allow one to use a Gaussian process prior, they typically impose strong smoothness assumptions on the covariance kernel. For instance, Theorem 4 of Chowdhury & Gopalan (2019) requires that the kernel function is four times differentiable.

**Contributions.** Our work offers a regret analysis for GP-PSRL that simultaneously addresses all three of the problems described above. We show that, with high-probability, the states visited by GP-PSRL (or indeed, any algorithm) are contained within a Euclidean ball whose radius grows only logarithmically with the total number of time steps $T$. We prove this using a recursive application of a tail bound for suprema of Gaussian processes, which is known as the Borell-Tsirelson-Ibragimov-Sudakov inequality.

We then prove a Bayesian regret bound with square root dependence on the expected information gain and linear dependence on the horizon $H$ (cf. Table 1). The expected information gain is an instance-dependent quantity that never has a worse growth rate than the maximum information gain (cf. Section 3.4). Our regret analysis accommodates Gaussian process priors, and only requires that the kernel function is bounded and Hölder continuous. We achieve this by replacing the confidence sets or discretization techniques

used in most previous works by a form of the chaining method (cf. Section 4.6).

**Outline.** Section 2 summarizes some known results about posterior sampling and reinforcement learning with GPs. In Section 3, we formally introduce the problem, we state our modeling assumptions and we describe the GP-PSRL algorithm. Section 4 contains our main results. In Section 5, we present some empirical results that support our theoretical results. We summarize our findings in Section 6. Some further discussion can be found in Appendix A.

**Notation.** For any positive integer $m$, $[m] := \{1, \ldots, m\}$. For any set $\mathcal{Z}$, any metric $d$ and any $\varepsilon > 0$, $\mathsf{N}(\mathcal{Z}, d, \varepsilon)$ is the $\varepsilon$-covering number of $\mathcal{Z}$ w.r.t. $d$. We define $d_2(\boldsymbol{x}, \boldsymbol{y}) := \|\boldsymbol{x} - \boldsymbol{y}\|_2$ to be the Euclidean metric. For any point $\boldsymbol{x} \in \mathbb{R}^d$, $\mathbb{B}_{\boldsymbol{x}}^d(R) := \{\boldsymbol{y} \in \mathbb{R}^d : d_2(\boldsymbol{x}, \boldsymbol{y}) \le R\}$ is the $d$-dimensional (closed) Euclidean ball of radius $R$, centered at $\boldsymbol{x}$. We define $\mathbb{B}^d(R) := \mathbb{B}_0^d(R)$ to be the ball around the origin. For any event $A$, we define $\mathbb{I}\{A\}$ to be the indicator function that equals $1$ if $A$ occurs and $0$ otherwise.

## 2. Related Work

We give a broad overview of some related work. A more detailed discussion about related work and the three problems listed in the introduction can be found in Appendix B.

Posterior sampling originated as a heuristic for decision-making under uncertainty (Thompson, 1933). However, it has since been shown to satisfy (near-)optimal regret bounds for a variety of multi-armed bandit problems (Agrawal & Goyal, 2012; Kaufmann et al., 2012), including linear bandits (Agrawal & Goyal, 2013; Russo & Van Roy, 2013; Abeille & Lazaric, 2017a) and Gaussian process bandits, which is also known as kernelized bandits (Russo & Van Roy, 2014; Chowdhury & Gopalan, 2017).

Posterior sampling was extended to reinforcement learning by Strens (2000). It was later shown that PSRL satisfies sublinear regret bounds in tabular MDPs (Osband et al., 2013; Osband & Van Roy, 2017) and in MDPs with bounded eluder dimension (Osband & Van Roy, 2014). Several works have applied PSRL to linear quadratic control problems (Abeille & Lazaric, 2017b; 2018; Ouyang et al., 2019; Faradonbeh et al., 2020), as well as other parametric control problems (Abbasi-Yadkori & Szepesvári, 2015), often focusing on the infinite-horizon setting. Chowdhury & Gopalan (2019) proved Bayesian regret bounds for PSRL in the setting that we consider. However, it is assumed that the state space is compact and that the prior either has bounded support or is a Gaussian process with a four times differentiable kernel. Fan & Ming (2021) proved Bayesian regret bounds for PSRL in MDPs with linear dynamics with respect to a fixed feature space. Kakade et al. (2020) and Curi et al. (2020) developed optimistic algorithms that satisfy worst-case regret bounds for MDPs with dynamics governed by a fixed function with bounded RKHS norm.

In our regret analysis, we use a version of the chaining method (Dudley, 1967; Talagrand, 2014) to control a sum of model estimation errors (cf. Section 4.6). Various forms of the chaining method have previously been used to control regret or estimation errors in Gaussian process bandits (Contal et al., 2015; Contal & Vayatis, 2016) and sequential level set estimation problems (Shekhar & Javidi, 2019).

# 3. Preliminaries

We provide the problem statement and the general definitions that appear throughout the paper, and we introduce the GP-PSRL algorithm.

## 3.1. Markov Decision Processes

A finite-horizon MDP is a tuple $\mathcal{M} = (\mathcal{S}, \mathcal{A}, r, P, \rho, H)$, where $\mathcal{S}$ is the state space, $\mathcal{A}$ is the action space, $r : \mathcal{S} \times \mathcal{A} \to \mathbb{R}$ is the (deterministic) reward function, $P : \mathcal{S} \times \mathcal{A} \to \mathcal{P}(\mathcal{S})$ is the transition kernel, $\rho \in \mathcal{P}(\mathcal{S})$ is the initial state distribution and $H$ is the horizon. A policy $\pi : \mathcal{S} \times [H] \to \mathcal{P}(\mathcal{A})$ is a time-dependent behavior rule that maps any state and any step to a probability distribution on the set of actions. For any MDP $\mathcal{M}$, any policy $\pi$ and any time step $h$, the value function $V_{\pi,h}^{\mathcal{M}} : \mathcal{S} \to \mathbb{R}$ is defined by

$$V_{\pi,h}^{\mathcal{M}}(\boldsymbol{s}) := \mathbb{E}_{\mathcal{M},\pi}\left[\sum_{j=h}^{H} r(\boldsymbol{s}_j, \boldsymbol{a}_j) \,\big|\, \boldsymbol{s}_h = \boldsymbol{s}\right],$$

where $\mathbb{E}_{\mathcal{M},\pi}[\cdot]$ denotes the expectation with respect to the random sequence of states and actions $\boldsymbol{s}_1, \boldsymbol{a}_1, \ldots, \boldsymbol{s}_H, \boldsymbol{a}_H$ generated by the policy $\pi$, the transition kernel $P$ and the initial state distribution $\rho$ over $H$ steps. For $h = H + 1$, define $V_{\pi,H+1}^{\mathcal{M}}(\boldsymbol{s}) := 0$. We say that a policy $\pi$ is optimal

for an MDP $\mathcal{M}$ if for all states $\boldsymbol{s} \in \mathcal{S}$,

$$V_{\pi,1}^{\mathcal{M}}(\boldsymbol{s}) \geq \max_{\pi'} V_{\pi',1}^{\mathcal{M}}(\boldsymbol{s}).$$

For any MDP $\mathcal{M}$, any policy $\pi$, any time step $h$ and any function $V : \mathcal{S} \to \mathbb{R}$, we define the Bellman operator $\mathsf{T}_{\pi,h}^{\mathcal{M}} : \mathbb{R}^{\mathcal{S}} \to \mathbb{R}^{\mathcal{S}}$ by,

$$\mathsf{T}_{\pi,h}^{\mathcal{M}} V(\boldsymbol{s}) := \int_{\mathcal{A}} r(\boldsymbol{s}, \boldsymbol{a}) + \langle P(\boldsymbol{s}, \boldsymbol{a}), V\rangle \mathrm{d}\pi(\boldsymbol{a}|\boldsymbol{s}, h),$$

where for any (signed) measure $P$ on $\mathcal{S}$ and any function $V : \mathcal{S} \to \mathbb{R}$, $\langle P, V\rangle := \int_{\mathcal{S}} V(\boldsymbol{s})\mathrm{d}P(\boldsymbol{s})$. From this definition, for any state $\boldsymbol{s} \in \mathcal{S}$ we have

$$\mathsf{T}_{\pi,h}^{\mathcal{M}} V_{\pi,h+1}^{\mathcal{M}}(\boldsymbol{s}) = V_{\pi,h}^{\mathcal{M}}(\boldsymbol{s}). \qquad (1)$$

## 3.2. Regret Minimization in MDPs

We consider the following episodic interaction protocol. At the start of the interaction, the true MDP $\mathcal{M}_\star = (\mathcal{S}, \mathcal{A}, r, P^\star, \rho, H)$ is drawn randomly from a known prior distribution (which will be specified in Section 3.3). The interaction then proceeds for a total of $T$ time steps, which is divided into $N$ episodes, each of length $H$ (so $T = NH$). At the start of each episode $n = 1, \ldots, N$, the agent chooses a policy $\pi_n$, and an initial state $\boldsymbol{s}_{n,1}$ is drawn from the initial state distribution $\rho$. Then for each time step $h = 1, \ldots, H$: (a) the agent observes the state $\boldsymbol{s}_{n,h}$; (b) the agent draws an action $\boldsymbol{a}_{n,h} \sim \pi_n(\boldsymbol{s}_{n,h}, h)$ from a policy $\pi_n$; (c) the next state $\boldsymbol{s}_{n,h+1} \sim P^\star(\boldsymbol{s}_{n,h}, \boldsymbol{a}_{n,h})$ is drawn from the transition kernel $P^\star$ (unless $h = H$). The agent is therefore defined by the sequence of policies $\pi_1, \ldots, \pi_N$, which we will also refer to as "the algorithm". The interaction between the agent and the MDP produces a random sequence of outcomes $\boldsymbol{s}_{1,1}, \boldsymbol{a}_{1,1}, \ldots, \boldsymbol{s}_{N,H}, \boldsymbol{a}_{N,H}$. The goal of the agent is to minimize the regret, which is the loss of reward caused by playing the polices $\pi_1, \ldots, \pi_N$ instead of the optimal policy. We define the regret after $T$ steps as

$$\mathcal{R}_T := \mathbb{E}\left[\sum_{n=1}^{N} V_{\pi^\star,1}^{\mathcal{M}_\star}(\boldsymbol{s}_{n,1}) - V_{\pi_n,1}^{\mathcal{M}_\star}(\boldsymbol{s}_{n,1})\right].$$

The expectation is taken with respect to all sources of randomness, including the random draw of $\mathcal{M}_\star$ at the start of the interaction and any randomness involved in the agent's selection of the policies $\pi_1, \ldots, \pi_N$. This notion of regret is usually referred to as the Bayesian regret.

## 3.3. Modeling Assumptions

We consider the continuous setting, where $\mathcal{S} = \mathbb{R}^{d_s}$ and $\mathcal{A} \subseteq \mathbb{R}^{d_a}$. Note that the state space is the entirety of $\mathbb{R}^{d_s}$, whereas the action set is only a subset of $\mathbb{R}^{d_a}$. In fact, we assume that $\mathcal{A}$ is contained within a Euclidean ball.

**Assumption 3.1.** There exists a positive constant $R_a$ such that $\mathcal{A} \subseteq \mathbb{B}^{d_a}(R_a)$.

We assume that the reward function is a known and uniformly (on $\mathcal{S} \times \mathcal{A}$) bounded function.

**Assumption 3.2.** There exists a positive constant $R_{\max}$ such that for all $(\boldsymbol{s}, \boldsymbol{a}) \in \mathcal{S} \times \mathcal{A}$, $|r(\boldsymbol{s}, \boldsymbol{a})| \leq R_{\max}$.

The transition kernel $P$ is assumed to be of the form

$$P(\boldsymbol{s}, \boldsymbol{a}) = \mathcal{N}(f(\boldsymbol{s}, \boldsymbol{a}), \sigma^2 \boldsymbol{I}),$$

where $f : \mathcal{S} \times \mathcal{A} \to \mathcal{S}$ is a function that maps state-action pairs to states and $\sigma$ is the (known) standard deviation of the noise. We sometimes refer to $f$ as the dynamics of the MDP. We assume that the initial state distribution is $\rho = \mathcal{N}(0, \sigma^2 \boldsymbol{I})$. We write this as $\boldsymbol{s}_{n,1} = \boldsymbol{\varepsilon}_{n,1}$ and

$$\boldsymbol{s}_{n,h+1} = f^\star(\boldsymbol{s}_{n,h}, \boldsymbol{a}_{n,h}) + \boldsymbol{\varepsilon}_{n,h+1},$$

where for all $n \in [N]$ and $h \in [H]$, $\boldsymbol{\varepsilon}_{n,h} \sim \mathcal{N}(0, \sigma^2 \boldsymbol{I})$. We will frequently use $\mathcal{X} = \mathcal{S} \times \mathcal{A} \subset \mathbb{R}^{d_s + d_a}$ to denote the set of possible state action pairs. We will also use the notations $\boldsymbol{x} = (\boldsymbol{s}, \boldsymbol{a})$, $r(\boldsymbol{x}) = r(\boldsymbol{s}, \boldsymbol{a})$ and $f(\boldsymbol{x}) = f(\boldsymbol{s}, \boldsymbol{a})$ interchangeably. We will typically use $\mathcal{Z}$ to denote an arbitrary subset of $\mathbb{R}^{d_s + d_a}$. For each $i \in [d_s]$, we use $f_i : \mathcal{X} \to \mathbb{R}$ to denote the $i^{\text{th}}$ component of $f$, so $f = (f_1, \ldots, f_{d_s})$.

We model each component $f_i^\star$ of the dynamics $f^\star$ of the true $\mathcal{M}_\star$ as an independent random draw from a zero-mean Gaussian process with covariance kernel $c : \mathbb{R}^{d_s + d_a} \times \mathbb{R}^{d_s + d_a} \to \mathbb{R}$. We will write this as $f_1^\star, \ldots, f_{d_s}^\star \sim \mathcal{GP}(0, c(\boldsymbol{x}, \boldsymbol{y}))$, where $\boldsymbol{x}, \boldsymbol{y}$ are understood to be elements of $\mathcal{X}$. This is the prior on MDPs described in Section 3.2. Let us define $\mathcal{F}_n := \sigma(\boldsymbol{s}_{1,1}, \boldsymbol{a}_{1,1}, \ldots, \boldsymbol{s}_{n,H}, \boldsymbol{a}_{n,H})$ to be the $\sigma$-algebra generated by the interaction history of the agent up to the end of episode $n$. Conditioned on any history $\mathcal{F}_n$, the Bayesian posterior on each component $f_i^\star$ is another Gaussian process, with mean and covariance

$$\mu_{n,i}(\boldsymbol{x}) = \boldsymbol{c}_n^\top(\boldsymbol{x})(\boldsymbol{C}_n + \sigma^2 \boldsymbol{I})^{-1} \boldsymbol{y}_{n,i}, \tag{2}$$

$$c_n(\boldsymbol{x}, \boldsymbol{y}) = c(\boldsymbol{x}, \boldsymbol{y}) - \boldsymbol{c}_n^\top(\boldsymbol{x})(\boldsymbol{C}_n + \sigma^2 \boldsymbol{I})^{-1} \boldsymbol{c}_n(\boldsymbol{y}), \tag{3}$$

where $\boldsymbol{y}_{n,i} \in \mathbb{R}^{n(H-1)}$ is the vector containing the $i^{\text{th}}$ component of every observed next state from the first $n$ episodes, $\boldsymbol{c}_n(\boldsymbol{x}) := [c(\boldsymbol{x}, \boldsymbol{x}_{1,1}), \ldots, c(\boldsymbol{x}, \boldsymbol{x}_{n,H-1})]^\top \in \mathbb{R}^{n(H-1)}$ and $\boldsymbol{C}_n$ is the $n(H-1) \times n(H-1)$ kernel matrix constructed from all state-action pairs in the first $H-1$ steps of the first $n$ episodes. Let us define $\sigma_n^2(\boldsymbol{x}) = c_n(\boldsymbol{x}, \boldsymbol{x})$ to be the posterior predictive variance at the state-action pair $\boldsymbol{x}$.

To facilitate our theoretical analysis, we will impose the following assumptions on the kernel function of the prior. First, we assume that the kernel is bounded.

**Assumption 3.3.** There exists a positive constant $C$ such that for all $\boldsymbol{x} \in \mathbb{R}^{d_s + d_a}$, $0 < c(\boldsymbol{x}, \boldsymbol{x}) \leq C$.

The assumption that $c(\boldsymbol{x}, \boldsymbol{x})$ is positive is just for convenience. It ensures that the prior and posterior variance at

every point is non-zero, so we don't have to be careful about dividing by the variance. We will also assume that the kernel is Hölder continuous.

**Assumption 3.4.** There exist positive constants $L > 0$ and $\alpha \in (0, 1]$ such that for all $\boldsymbol{x}, \boldsymbol{y}, \boldsymbol{z} \in \mathbb{R}^{d_s + d_a}$,

$$|c(\boldsymbol{x}, \boldsymbol{y}) - c(\boldsymbol{x}, \boldsymbol{z})| \leq L \|\boldsymbol{y} - \boldsymbol{z}\|_2^\alpha.$$

We consider these assumptions on the kernel function (especially the smoothness assumption) to be quite mild. They are satisfied by the commonly used squared exponential and Matérn kernels. We introduce two quite fundamental objects associated with the kernel functions of Gaussian processes. First, we define the natural distance $d_c : \mathbb{R}^{d_s + d_a} \times \mathbb{R}^{d_s + d_a} \to \mathbb{R}$ associated with the kernel $c$ by

$$d_c(\boldsymbol{x}, \boldsymbol{y}) := \sqrt{c(\boldsymbol{x}, \boldsymbol{x}) - 2c(\boldsymbol{x}, \boldsymbol{y}) + c(\boldsymbol{y}, \boldsymbol{y})}. \tag{4}$$

The natural distance arises naturally when working with Gaussian processes, since if $f \sim \mathcal{GP}(0, c(\boldsymbol{x}, \boldsymbol{y}))$, then

$$\mathbb{E}[(f(\boldsymbol{x}) - f(\boldsymbol{y}))^2] = d_c^2(\boldsymbol{x}, \boldsymbol{y}).$$

This will appear in the chaining arguments in Section 4.

### 3.4. Information Gain

We define two kernel-dependent quantities that appear in regret bounds for reinforcement learning with Gaussian processes. For any radius $R \geq 0$, let $A = \{\sup_{n \in [N], h \in [H]} \|\boldsymbol{x}_{n,h}\|_2 \leq R\}$ be the event that all observed state action pairs have norm at most $R$. We define the maximum information gain as

$$\Gamma_T(\sigma^2, R) := \sup_{\boldsymbol{x}_{1,1}, \ldots, \boldsymbol{x}_{N,H}} \mathbb{I}\{A\} \frac{1}{2} \log \det\left(\frac{1}{\sigma^2} \widetilde{\boldsymbol{C}}_T + \boldsymbol{I}\right),$$

where $\widetilde{\boldsymbol{C}}_T$ is the $T \times T$ kernel matrix constructed from the state-action pairs $\boldsymbol{x}_{1,1}, \ldots, \boldsymbol{x}_{N,H}$. The maximum information gain can be interpreted as the maximum amount of information that a sequence of $T$ state action pairs can provide about the model $f^\star$. The presence of the indicator variable in the definition of $\Gamma_T(\sigma^2, R)$ is somewhat non-standard. The maximum information gain is usually defined as a supremum over the entire state-action space. However, if the set over which the supremum is taken is unbounded, then the maximum information gain is usually linear in $T$. For instance, if the kernel is either the squared exponential or the Matérn kernel, then

$$\lim_{R \to \infty} \Gamma_T(\sigma^2, R) = \frac{T}{2} \log(1 + 1/\sigma^2).$$

The maximum information gain is a worst-case quantity in the sense that it contains a supremum over all possible (bounded) sequences of state-action pairs. Regret bounds

**Algorithm 1** GP-PSRL

**Input:** Prior covariance $c$, noise $\sigma^2$, reward function $r$.
**for** $n = 1$ **to** $N$ **do**
    Sample $\mathcal{M}_n$ from the posterior over MDPs.
    Compute an optimal policy $\pi_n$ for $\mathcal{M}_n$.
    **for** $h = 1$ **to** $H$ **do**
        Observe the state $s_{n,h}$.
        Draw an action $a_{n,h} \sim \pi_n(s_{n,h}, h)$.
    **end for**
    Update the posterior over MDPs.
**end for**

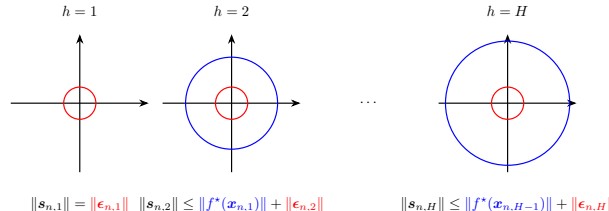

$\|s_{n,1}\| = \|\epsilon_{n,1}\| \quad \|s_{n,2}\| \le \|f^\star(x_{n,1})\| + \|\epsilon_{n,2}\| \qquad \|s_{n,H}\| \le \|f^\star(x_{n,H-1})\| + \|\epsilon_{n,H}\|$

*Figure 1.* The first state in each episode is drawn from an isotropic Gaussian, and so its norm is sub-Gaussian. As long as the norm of the current state is bounded, the next state is sub-Gaussian, and so its norm is also sub-Gaussian. The bound on the norm of the state at step $h$ will grow with $h$. The challenge is to show that this bound does not grow too quickly.

that depend on the maximum information gain may therefore be overly pessimistic–even if they have the "correct" dependence on $\Gamma_T$. We would like to have regret bounds that depend on a less pessimistic quantity. For example, we could replace the supremum inside the maximum information gain with an expectation. We define the expected information gain as

$$\gamma_T(\sigma^2, R) := \mathbb{E}\left[\mathbb{I}\{A\}\frac{1}{2}\log\det\left(\frac{1}{\sigma^2}\widetilde{C}_T + I\right)\right]. \quad (5)$$

The expected information gain can be interpreted as the average amount of information that a sequence of $T$ state-action pairs collected by a given algorithm provides about the model. For large $T$, we can expect $\gamma_T(\sigma^2, R)$ to be considerably smaller than $\Gamma_T(\sigma^2, R)$ because an algorithm that has sublinear regret will eventually stop collecting informative data. In any case, the expected information gain is never larger than the maximum information gain. Therefore, everything else being equal, one should always prefer a regret bound that depends on the expected information gain instead of the maximum information gain.

### 3.5. Posterior Sampling Reinforcement Learning

PSRL begins with a prior distribution on (the dynamics of) MDPs, which is updated at the end of each episode as new data are collected. At the start of the $n^{\text{th}}$ episode, PSRL samples a function $f^{(n)}$ from the Bayesian posterior distribution of $f^\star$ conditioned on the history $\mathcal{F}_{n-1}$. Let us define $P^{(n)}$ to be the transition kernel with dynamics $f^{(n)}$ and $\mathcal{M}_n := (\mathcal{S}, \mathcal{A}, r, P^{(n)}, \rho, H)$ to be the MDP sampled by PSRL at the beginning of episode $n$. PSRL then plays any policy $\pi_n$ which is an optimal policy for the sampled MDP $\mathcal{M}_n$. For the model we consider with a conjugate Gaussian process prior, we refer to PSRL as GP-PSRL.

As discussed in Section 3.3, the posterior of each component $f_i^\star$ of $f^\star$ conditioned on $\mathcal{F}_{n-1}$ is a Gaussian process with the mean $\mu_{n-1,i}$ and the covariance $c_{n-1}$ given in (2) and (3). Therefore, conditioned on $\mathcal{F}_{n-1}$, we have $f_i^{(n)} \sim \mathcal{GP}(\mu_{n-1,i}(x), c_{n-1}(x, y))$ and $f^{(n)} = (f_1^{(n)}, \ldots, f_{d_s}^{(n)})$.

## 4. Regret Analysis

We state and sketch the proof of our regret bound for GP-PSRL. The method of proof can be split into two parts. In the first part, we show that with high probability, all the states observed by the algorithm are contained within a Euclidean ball with a radius that grows only logarithmically with $T$. In the second part, under the event that all the observed states are bounded, we prove an upper bound for the Bayesian regret of GP-PSRL.

The general idea behind the first part of the proof is illustrated in Figure 1. We exploit the fact that as long as the norm of the current state is bounded, then the norm of the next state has sub-Gaussian tail behavior, which means it can be bounded with high probability. In particular, since $s_{n,h+1}$ can be written as

$$s_{n,h+1} = f^\star(s_{n,h}, a_{n,h}) + \varepsilon_{n,h+1},$$

the norm of $s_{n,h+1}$ is sub-Gaussian as long as the norm of $f^\star(s_{n,h}, a_{n,h})$ is sub-Gaussian. If the norm of $s_{n,h}$ is bounded, then for some finite radius $R$, the norm of $f^\star(s_{n,h}, a_{n,h})$ can be upper bounded by the supremum $\sup_{x \in \mathbb{B}^{d_s+d_s}(R)} \|f^\star(x)\|_2$. To show that this supremum is sub-Gaussian, we use a version of the Borell-Tsirelson-Ibragimov-Sudakov (BTIS) inequality (Borell, 1975; Tsirelson et al., 1976), which states that the supremum of a Gaussian process is sub-Gaussian (cf. Lemma 4.1). Within each episode, the bounds on the norms of the states will increase as the step $h$ increases. It remains to show that this increase is not too fast (cf. Section 4.3).

The second part of the proof can be split into two parts. We show that the regret of GP-PSRL can be upper bounded by a sum of model estimation errors (cf. Section 4.5) and then we upper bound the model estimation errors (cf. Section 4.6). The main novelty in this part of the proof is in how we obtain an upper bound for the estimation errors that depends on the expected information gain (as opposed to the maximum information gain) and holds under weak smoothness assumptions on the kernel.

## 4.1. Suprema of Gaussian Processes

We describe the tools that we use to control suprema of Gaussian processes. First, we focus on the supremum $\sup_{\boldsymbol{x} \in \mathcal{Z}} f(\boldsymbol{x})$, where $\mathcal{Z}$ is some subset of $\mathbb{R}^{d_s+d_a}$ and $f \sim \mathcal{GP}(0, c(\boldsymbol{x}, \boldsymbol{y}))$. For any (sufficiently large) threshold $u$, we would like to have an upper bound for the tail probability $\mathbb{P}(\sup_{\boldsymbol{x} \in \mathcal{Z}} f(\boldsymbol{x}) > u)$. We use the BTIS inequality to obtain such a tail bound (Lemma 4.1).

**Lemma 4.1** (Borell-Tsirelson-Ibragimov-Sudakov). *Let $\mathcal{Z}$ be a subset of $\mathbb{R}^{d_s+d_a}$, let $f \sim \mathcal{GP}(0, c(\boldsymbol{x}, \boldsymbol{y}))$ and suppose that $\mathbb{E}[\sup_{\boldsymbol{x} \in \mathcal{Z}} f(\boldsymbol{x})] < \infty$. Let $v := \sup_{\boldsymbol{x} \in \mathcal{Z}} \mathbb{E}[f(\boldsymbol{x})^2]$. For every $u > 0$,*

$$\mathbb{P}\left( \sup_{\boldsymbol{x} \in \mathcal{Z}} f(\boldsymbol{x}) - \mathbb{E}\left[ \sup_{\boldsymbol{x} \in \mathcal{Z}} f(\boldsymbol{x}) \right] \geq u \right) \leq e^{-\frac{u^2}{2v}}.$$

If Assumption 3.3 is satisfied, then $\sup_{\boldsymbol{x} \in \mathcal{Z}} \mathbb{E}[f(\boldsymbol{x})^2] \leq C$. For $u$ much larger than $\mathbb{E}[\sup_{\boldsymbol{x} \in \mathcal{Z}} f(\boldsymbol{x})]$, we therefore have the estimate

$$\mathbb{P}\left( \sup_{\boldsymbol{x} \in \mathcal{Z}} f(\boldsymbol{x}) \geq u \right) \sim e^{-\frac{u^2}{2C}}. \tag{6}$$

To determine how large $u$ needs to be for this to be a reasonable estimate of the tail probability, we need to upper bound the expected supremum of $f$. To do so, we can use the chaining method (Dudley, 1967).

**Lemma 4.2** (Dudley). *Let $\mathcal{Z}$ be a subset of $\mathbb{R}^{d_s+d_a}$. If $f \sim \mathcal{GP}(0, c(\boldsymbol{x}, \boldsymbol{y}))$, then*

$$\mathbb{E}\left[ \sup_{\boldsymbol{x} \in \mathcal{Z}} f(\boldsymbol{x}) \right] \leq 12 \int_0^\infty \sqrt{\log \mathsf{N}(\mathcal{Z}, d_c, \varepsilon)} \, \mathrm{d}\varepsilon.$$

We recall that $\mathsf{N}(\mathcal{Z}, d_c, \varepsilon)$ is the $\varepsilon$-covering number of $\mathcal{Z}$ with respect to the natural distance $d_c$ (cf. (4)). The quantity $\log \mathsf{N}(\mathcal{Z}, d_c, \varepsilon)$ is known as the metric entropy of $\mathcal{Z}$ (w.r.t. $d_c$), and consequently, the integral in Lemma 4.2 is sometimes called the entropy integral. For any metric $d$, let us define the diameter of $\mathcal{Z}$ as

$$\mathrm{diam}_d(\mathcal{Z}) := \sup_{\boldsymbol{x}, \boldsymbol{y} \in \mathcal{Z}} d(\boldsymbol{x}, \boldsymbol{y}).$$

Notice that for any $\varepsilon \geq \mathrm{diam}_{d_c}(\mathcal{Z})$, any point in $\mathcal{Z}$ forms an $\varepsilon$-covering of $\mathcal{Z}$ w.r.t. $d_c$. This means that the upper limit of the entropy integral can be replaced by $\mathrm{diam}_{d_c}(\mathcal{Z})$. Therefore, explicit upper bounds for the expected supremum can be obtained by plugging in upper bounds for the covering number and the diameter of $\mathcal{Z}$ with respect to $d_c$.

## 4.2. Suprema of Vector-Valued Gaussian Processes

We turn our attention to suprema of vector-valued Gaussian processes. In particular, let $f_1, \ldots, f_{d_s} \sim \mathcal{GP}(0, c(\boldsymbol{x}, \boldsymbol{y}))$ be independent GPs, and let $f = (f_1, \ldots, f_{d_s})$. For any radius $R > 0$, we would like to have a bound for the tail

probability $\mathbb{P}(\sup_{\boldsymbol{x} \in \mathbb{B}^{d_s+d_a}(R)} \|f(\boldsymbol{x})\|_2 > u)$. Due to the following Gaussian concentration inequality, the supremum of $\|f(\boldsymbol{x})\|_2$ can be dealt with using tools from Section 4.1.

**Lemma 4.3** (Theorem 2.26 in Wainwright, 2019). *Let $(X_1, \ldots, X_n)$ be a vector of i.i.d. standard Gaussian random variables, and let $g : \mathbb{R}^n \to \mathbb{R}$ be $L$-Lipschitz with respect to the Euclidean metric. Then the random variable $g(X_1, \ldots, X_n)$ is $L$-sub-Gaussian.*

Since the Euclidean norm is 1-Lipschitz w.r.t. the Euclidean metric, Lemma 4.3 tells us that for any $\boldsymbol{x}$, the $\|f(\boldsymbol{x})\|_2$ exhibits concentration similar to each of the components $f_i(\boldsymbol{x})$. Using Lemma 4.3, one can show that a version of the BTIS inequality holds for the supremum $\sup_{\boldsymbol{x}} \|f(\boldsymbol{x})\|_2$.

**Lemma 4.4.** *Suppose that Assumption 3.3 is satisfied. Let $f_1, \ldots, f_{d_s} \sim \mathcal{GP}(0, c(\boldsymbol{x}, \boldsymbol{y}))$ be independent GPs and let $f = (f_1, \ldots, f_{d_s})$. For any $R > 0$, let $\mathcal{Z} = \mathbb{B}^{d_s+d_a}(R)$. For any $u > 0$,*

$$\mathbb{P}\left( \sup_{\boldsymbol{x} \in \mathcal{Z}} \|f(\boldsymbol{x})\|_2 - \mathbb{E}\left[ \sup_{\boldsymbol{x} \in \mathcal{Z}} \|f(\boldsymbol{x})\|_2 \right] \geq u \right) \leq e^{-\frac{u^2}{2C}}.$$

The proof of Lemma 4.4 can be found in Appendix D.2, and is inspired by the proof of the standard BTIS inequality from van Handel (2016) (cf. Lemma 6.12). The idea is to establish that for any finite subset $\mathcal{Z} \subset \mathbb{B}^{d_s+d_a}(R)$, the supremum $\sup_{\boldsymbol{x} \in \mathcal{Z}} \|f(\boldsymbol{x})\|_2$ is equal in distribution to a $\sqrt{C}$-Lipschitz function of a finite-dimensional standard Gaussian random vector. By Lemma 4.3, this means that the supremum is $\sqrt{C}$-sub-Gaussian. One can then use separability (cf. Definition C.1) to show that this holds even when $\mathcal{Z} = \mathbb{B}^{d_s+d_a}(R)$. As one might expect, the chaining argument can be applied to $\sup_{\boldsymbol{x}} \|f(\boldsymbol{x})\|_2$ as well.

**Lemma 4.5.** *Let $f_1, \ldots, f_{d_s} \sim \mathcal{GP}(0, c(\boldsymbol{x}, \boldsymbol{y}))$ be independent Gaussian processes and let $f = (f_1, \ldots, f_{d_s})$. For any $R > 0$, let $\mathcal{Z} = \mathbb{B}^{d_s+d_a}(R)$. If Assumption 3.3 is satisfied,*

$$\mathbb{E}\left[ \sup_{\boldsymbol{x} \in \mathcal{Z}} \|f(\boldsymbol{x})\|_2 \right] \leq 12 \int_0^\infty \sqrt{\log \mathsf{N}(\mathcal{Z}, d_c, \varepsilon)} \, \mathrm{d}\varepsilon$$
$$+ (12 + \sqrt{d_s})\sqrt{C}.$$

The proof of Lemma 4.5 can be found in Appendix D.1, and is inspired by the proof of the chaining argument from van Handel (2016) (cf. Theorem 5.24). Compared to the bound in Lemma 4.2, the bound in Lemma 4.5 has an extra term. This is because $\|f(\boldsymbol{x})\|_2$ is not zero-mean. In Appendix D.1, we derive bounds on the covering number and diameter of $\mathbb{B}^{d_s+d_a}(R)$ w.r.t. $d_c$, where $c$ is any kernel that satisfies our smoothness and boundedness assumptions. As described in Section 4.1, these bounds can be combined with Lemma 4.5 to obtain an explicit bound for the supremum (cf. Lemma D.5). Lemma 4.6 shows that when $u$ is at least twice as big as this upper bound, $\sup_{\boldsymbol{x} \in \mathbb{B}^{d_s+d_a}(R)} \|f(\boldsymbol{x})\|_2$ satisfies a tail bound similar to the one in (6).

**Lemma 4.6.** *Suppose that Assumption 3.3 and Assumption 3.4 are satisfied. Let $f_1, \ldots, f_{d_s} \sim \mathcal{GP}(0, c(\boldsymbol{x}, \boldsymbol{y}))$ be independent GPs and let $f = (f_1, \ldots, f_{d_s})$. For any $R > 0$ and any $u$ such that*

$$u \geq 84\alpha^{-1/2}\sqrt{C(d_s + d_a)\log(5 + 5R^\alpha L/C)}\,,$$

*we have*

$$\mathbb{P}\big(\sup_{\boldsymbol{x} \in \mathbb{B}^{d_s + d_a}(R)}\|f(\boldsymbol{x})\|_2 \geq u\big) \leq e^{-\frac{u^2}{8C}}\,. \quad (7)$$

The proof of Lemma 4.6 can be found in Appendix D.3.

### 4.3. Tail Bound for the Norm of the Largest State

We can now show that the states generated by running the algorithm remain close to the origin. We begin by formalizing the argument described at the beginning of this section. Let $R_1, \ldots, R_H$ be a sequence of radii and let us define $A_{n,h}$ to be the event that the state observed at step $h$ of episode $n$ has norm less than $R_h$, i.e.,

$$A_{n,h} := \{\|\boldsymbol{s}_{n,h}\|_2 \leq R_h\}\,.$$

In Appendix E.1 (see also Lemma E.4), we prove that

$$\mathbb{I}\{\cup_{n=1}^N \cup_{h=1}^H A_{n,h}^\mathsf{c}\} \leq \sum_{n=1}^N \sum_{h=1}^H \mathbb{I}\{A_{n,h}^\mathsf{c}\}\mathbb{I}\{A_{n,h-1}\}\,.$$

From the definition of $A_{n,h}$, this means that

$$\mathbb{P}(\cup_{n=1}^N \cup_{h=1}^H \{\|\boldsymbol{s}_{n,h}\|_2 > R_h\}) \quad (8)$$
$$\leq \sum_{n=1}^N \sum_{h=1}^H \mathbb{E}[\mathbb{I}\{\|\boldsymbol{s}_{n,h}\|_2 > R_h\}\mathbb{I}\{\|\boldsymbol{s}_{n,h-1}\|_2 \leq R_{h-1}\}]\,.$$

This inequality tells us that if for all $n \in [N]$ and $h \in [H]$ we can control the probability that $\|\boldsymbol{s}_{n,h}\|_2$ exceeds $R_h$, given that the $\|\boldsymbol{s}_{n,h-1}\|_2 \leq R_{h-1}$, then this automatically controls the probability that the norm of any state, say $\boldsymbol{s}_{n,h}$, exceeds the value $R_h$. It remains to determine how large $R_1, \ldots, R_H$ need to be to ensure that the right-hand side of (8) is of the order $1/T$. In Appendix E.2, we use the tail bound in Lemma 4.6 to upper bound each term on the right-hand side of (8). We then determine specific values of $R_1, \ldots, R_H$ that ensure that this bound is at most $2/T$. Finally, we prove by induction that if $T$ is sufficiently large, then these values of $R_1, \ldots, R_H$ satisfy $R_h \leq R$ for all $h \in [H]$, where $R$ is given in (9). Since

$$\mathbb{P}\big(\sup_{n \in [N], h \in [H]}\|\boldsymbol{s}_{n,h}\|_2 > R\big) \leq \mathbb{P}\big(\cup_{n=1}^N \cup_{h=1}^H A_{n,h}^\mathsf{c}\big)\,,$$

this leads to the following result.

**Lemma 4.7.** *Suppose that Assumption 3.1, Assumption 3.3 and Assumption 3.4 are satisfied. Let us define*

$$D := 168\alpha^{-1/2}\sqrt{\max(C, \sigma^2)(d_s + d_a)}\,,$$

*and*

$$R := D\sqrt{\log(10(T + R_a)\max(1, L/C))}\,. \quad (9)$$

*If the total number of rounds $T$ satisfies*

$$T \geq D\sqrt{2\log(10D\max(1, L/C)(R_a + 1))}\,, \quad (10)$$

*then*

$$\mathbb{P}\big(\sup_{n \in [N], h \in [H]}\|\boldsymbol{s}_{n,h}\|_2 > R\big) \leq \frac{2}{T}\,.$$

The proof can be found in Appendix E.3. Roughly speaking, whenever $T$ is larger than $\sqrt{d_s + d_a}$, we have

$$\mathbb{P}\big(\sup_{n \in [N], h \in [H]}\|\boldsymbol{s}_{n,h}\|_2 \geq \sqrt{(d_s + d_a)\log(T)}\big) \sim \frac{1}{T}\,.$$

Since $\|\boldsymbol{x}_{n,h}\|_2^2 = \|\boldsymbol{s}_{n,h}\|_2^2 + \|\boldsymbol{a}_{n,h}\|_2^2$, Lemma 4.7 also gives us a bound on the norm of the largest state-action pair. In particular, let us define

$$\widetilde{R} := \sqrt{R^2 + R_a^2}\,, \quad (11)$$

where $R$ is given in (9). Then

$$\mathbb{P}\big(\sup_{n \in [N], h \in [H]}\|\boldsymbol{x}_{n,h}\|_2 > \widetilde{R}\big) \leq \frac{2}{T}\,.$$

### 4.4. Bounding Regret by Value Estimation Error

We turn to the second part of the proof of our regret bound. The first step is to re-write the Bayesian regret as a sum of value estimation errors $V_{\pi_n,1}^{\mathcal{M}_n}(\boldsymbol{s}_{n,1}) - V_{\pi_n,1}^{\mathcal{M}^\star}(\boldsymbol{s}_{n,1})$. The $n^{\text{th}}$ term in the sum is the difference between the value of the policy $\pi_n$ in the sampled MDP $\mathcal{M}_n$ and the true MDP $\mathcal{M}_\star$. The following lemma is a re-statement of Lemma 1 in Osband et al. (2013).

**Lemma 4.8.** *If $\pi_n$ is an optimal policy for $\mathcal{M}_n$, then*

$$\mathcal{R}_T = \mathbb{E}\left[\sum_{n=1}^N V_{\pi_n,1}^{\mathcal{M}_n}(\boldsymbol{s}_{n,1}) - V_{\pi_n,1}^{\mathcal{M}^\star}(\boldsymbol{s}_{n,1})\right]\,.$$

The reason that the regret can be re-written in this way is that given any history $\mathcal{F}_{n-1}$, $f^\star$ and $f^{(n)}$ have the same conditional distribution.

### 4.5. Bounding Regret by Model Estimation Error

Let us now define the good event $A$ as

$$A := \{\sup_{n \in [N], h \in [H]}\|\boldsymbol{s}_{n,h}\|_2 \leq R\}\,.$$

Having established that the bad event $A^\mathsf{c}$ occurs with low probability, we conduct the rest of the regret analysis under the good event, which ensures that $\sup_{n \in [N], h \in [H]}\|\boldsymbol{x}_{n,h}\|_2 \leq \widetilde{R}$ (cf. (11)). The next step is a well-established idea in the analysis of model-based reinforcement learning algorithms. We show that the sum of the value estimation errors can be controlled by a sum of model estimation errors. Results of this type are sometimes referred to as simulation lemmas (Kearns & Singh, 2002).

**Lemma 4.9.** *Suppose that Assumption 3.2 is satisfied. For any event A,*

$$\mathbb{E}\left[\mathbb{I}\{A\} \sum_{n=1}^{N} \left( V_{\pi_n,1}^{\mathcal{M}_n}(\boldsymbol{s}_{n,1}) - V_{\pi_n,1}^{\mathcal{M}^\star}(\boldsymbol{s}_{n,1}) \right) \right]$$

$$\leq \frac{HR_{\max}}{\sigma} \mathbb{E}\left[\mathbb{I}\{A\} \sum_{n=1}^{N} \sum_{h=1}^{H-1} \|f^{(n)}(\boldsymbol{x}_{n,h}) - f^\star(\boldsymbol{x}_{n,h})\|_2\right]$$

$$+ 2R_{\max}H\sqrt{2\pi T}\,.$$

Lemma 4.9 is more or less a combination of two known results. The first is a result from Section 5.1 of Osband et al. (2013), which expresses the sum of the value estimation errors as a sum of Bellman errors. In Appendix F, we prove a slight extension of this result (cf. Lemma F.1), which accounts for the fact that both the value estimation errors and the Bellman errors are multiplied by a random indicator variable. The second half of Lemma 4.9 is essentially the same as Lemma 1 in Fan & Ming (2021), and allows us to upper bound the sum of the Bellman errors by the sum of the model estimation errors. In Appendix F we give a simpler (but less general) proof of this result using Pinsker's inequality (cf. Lemma F.2).

### 4.6. Bounding the Model Estimation Error

The final step is to upper bound the sum of the model estimation errors. One way to do this is to construct confidence sets that contain both $f^\star$ and $f^{(n)}$ with high probability. The sum of the estimation errors can then be upper bounded by the sum of the widths of these confidence sets. We believe that this idea was first used in the analysis of PSRL by Osband et al. (2013) (see also Osband & Van Roy, 2014; 2017). The problem with this approach, at least if one is interested in GP-PSRL, is that it tends to give bounds on the model estimation error that have linear dependence on $\Gamma_T$ (or related quantities such as the Eluder dimension Russo & Van Roy, 2013), which would lead to a final regret bound with linear dependence on $\Gamma_T$.

A second way to upper bound the sum of the estimation errors is to exploit the fact that, conditioned on any history $\mathcal{F}_{n-1}$, $f^{(n)}$ and $f^\star$ are both Gaussian processes, and use bounds for suprema of Gaussian processes. This approach is commonly used in proofs of Bayesian regret bounds for Gaussian process bandits (Srinivas et al., 2012). However, even in recent work (e.g., Takeno et al., 2023; 2024), it is typically assumed that the kernel satisfies much stronger smoothness conditions than the Hölder continuity condition in Assumption 3.4. Namely, Srinivas et al. (2012) and Takeno et al. (2023; 2024) assume that samples from the GP are differentiable, and that the partial derivatives are uniformly bounded with high probability. This is satisfied for stationary kernels that are four times differentiable (Ghosal

& Roy, 2006; Srinivas et al., 2012), which rules out Matérn kernels with $\nu \leq 2$. Contal & Vayatis (2016) improved upon this idea by replacing the single-step discretization techniques used in other works with a form of the chaining method, which allows similar results to be proved under much weaker smoothness assumptions (similar to our Assumption 3.4). We adopt this approach in our setting, and show that under weak smoothness conditions, one can derive bounds on the estimation error that have square root dependence on the expected information gain.

**Lemma 4.10.** *Suppose that assumptions 3.1, 3.3 and 3.4 are satisfied. Then*

$$\mathbb{E}\left[\mathbb{I}\{A\} \sum_{n=1}^{N} \sum_{h=1}^{H-1} \|f^{(n)}(\boldsymbol{x}_{n,h}) - f^\star(\boldsymbol{x}_{n,h})\|_2\right] =$$

$$\mathcal{O}\left(\sqrt{(d_s + d_a)\gamma_T(\sigma^2, \widetilde{R})T\log(T)}\right)\,.$$

The proof of Lemma 4.10 is in Appendix G.3. The general idea is to separate the estimation error into a discretized estimation error and two discretization error terms. For some $\varepsilon \in (0, \widetilde{R}]$, let $B_\varepsilon$ be a minimal $\varepsilon$-cover of $\mathbb{B}^{d_s+d_a}(\widetilde{R})$ w.r.t. $d_2$ and let $\omega : \mathbb{B}^{d_s+d_a}(\widetilde{R}) \to B_\varepsilon$ be any function that maps points in $\mathbb{B}^{d_s+d_a}(\widetilde{R})$ to the closest point in $B_\varepsilon$. We upper bound the discretized estimation error by the supremum of a normalized estimation error (over $B_\varepsilon$) multiplied by the posterior variance. Since the kernel function is assumed to be bounded, the posterior variance is a bounded random variable. We show that the supremum of the normalized estimation error is conditionally sub-Gaussian (cf. Lemma G.1). We then use a trick from Schwartz et al. (2025) (cf. their Lemma 18), which allows us to upper bound the expected value of the product of a sub-Gaussian random variable and a bounded, non-negative random variable (cf. Lemma G.4). At this point, we have reduced the problem of bounding the discretized estimation error to the problem of bounding the expectation of a sum of posterior variances. Using a version of the elliptical potential lemma from Vakili & Olkhovskaya (2024) (cf. Lemma G.8), we can upper bound this sum by a quantity of order $\sqrt{(d_s + d_a)\gamma_T T\log(1/\varepsilon)}$. It remains to upper bound the discretization errors. Each of the discretization error terms is at most

$$\mathbb{E}\left[\sup_{\boldsymbol{x} \in \mathbb{B}^{d_s+d_a}(R)} \sum_{n=1}^{N} \sum_{h=1}^{H-1} \|f(\boldsymbol{x}) - f(\omega(\boldsymbol{x}))\|_2\right]\,, \quad (12)$$

where $f_1, \ldots, f_{d_s} \sim \mathcal{GP}(0, c(\boldsymbol{x}, \boldsymbol{y}))$ are independent GPs and $f = (f_1, \ldots, f_{d_s})$. The challenge is to show that if $\varepsilon$ is sufficiently small, then (under weak smoothness conditions) the discretization error is negligible. We notice that each component of $\widetilde{f}(\boldsymbol{x}) := f(\boldsymbol{x}) - f(\omega(\boldsymbol{x}))$ is another Gaussian process with the kernel

$$\widetilde{c}(\boldsymbol{x}, \boldsymbol{y}) := c(\omega(\boldsymbol{x}), \omega(\boldsymbol{y})) - c(\omega(\boldsymbol{x}), \boldsymbol{y})$$
$$- c(\boldsymbol{x}, \omega(\boldsymbol{y})) + c(\boldsymbol{x}, \boldsymbol{y})\,.$$

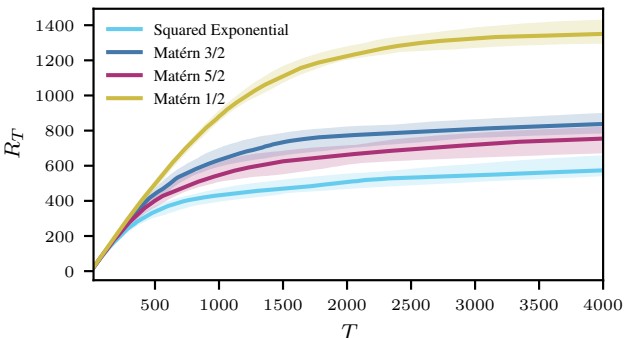

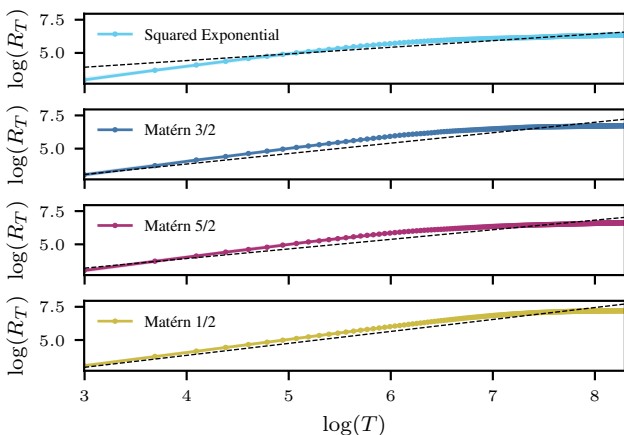

*Figure 2.* An empirical validation of the GP-PSRL algorithm, showing the Bayesian regret for GP-PSRL over 20 seeds and across four different GP priors. As our regret analysis would suggest, GP-PSRL has sublinear regret with all of these priors, and smoother priors result in lower regret.

While this kernel function is not continuous, it is still piecewise Hölder continuous. This allows us to suitably upper bound the covering number of $\mathbb{B}^{d_s+d_a}(\widetilde{R})$ w.r.t. the distance $d_{\widetilde{c}}$ (cf. Lemma G.12). It can also be shown that the diameter of $\mathbb{B}^{d_s+d_a}(\widetilde{R})$ w.r.t. $d_{\widetilde{c}}$ can be made arbitrarily small by decreasing $\varepsilon$ (cf. Lemma G.13). Therefore, when we use the chaining argument to upper bound the supremum in (12), the resulting entropy integral can be made arbitrarily small as well (cf. Lemma G.17). The proof of Lemma 4.10 then boils down to choosing a suitable value of $\varepsilon$.

### 4.7. Main Result

We can now state our main result.

**Theorem 4.11.** *Suppose that assumptions 3.1-3.4 are satisfied. Define $\widetilde{R}$ as in (11). For all $T$ that satisfy (10),*

$$\mathcal{R}_T = \mathcal{O}\left(H\sqrt{(d_s + d_a)\gamma_T(\sigma^2, \widetilde{R})T\log(T)}\right).$$

The proof is in Appendix H, though it is mainly just a matter of combining all the steps described previously.

### 4.8. Specialization to Matérn Kernels

Using a bound on the maximum information gain from Vakili & Olkhovskaya (2023), we can specialize our main result to Matérn kernels.

**Lemma 4.12** (Lemma 2 in Vakili & Olkhovskaya (2023)). *For the Matérn kernel with parameter $\nu > 0$,*

$$\Gamma_T(\sigma^2, \widetilde{R}) = \mathcal{O}\left(T^{\frac{d_s+d_a}{2\nu+d_s+d_a}}\log^{\frac{2\nu}{2\nu+d_s+d_a}}(T)\widetilde{R}^{\frac{2\nu(d_s+d_a)}{2\nu+d_s+d_a}}\right).$$

Since in Theorem 4.11, $\widetilde{R} = \mathcal{O}(\sqrt{(d_s + d_a)\log(T)})$, we obtain the following regret bound for Matérn kernels.

**Corollary 4.13.** *Suppose that $c$ is the Matérn kernel with parameter $\nu > 0$. Suppose that assumptions 3.1-3.2 are*

*Figure 3.* A log-log plot of cumulative regret against steps. The dotted line shows the best fit of a 1/2 slope to verify our proposed $\sqrt{T}$ rate for the squared exponential kernel, and slopes of 9/10, 11/14 and 13/18 for the Matérn 1/2, 3/2 and 5/2 kernels respectively, following the specialized rates in Section 4.8.

*satisfied. Let $d := d_s + d_a$. For all $T$ that satisfy (10),*

$$\mathcal{R}_T = \mathcal{O}\left(Hd^{\frac{\nu+1}{2}}T^{\frac{\nu+d}{2\nu+d}}\log^{\max(\frac{\nu+1}{2}, \frac{3\nu+1}{2\nu+2})}(T)\right).$$

Up to logarithmic factors, this recovers the best known rate in $T$–even in the special case of Gaussian process bandits.

## 5. Experiments

We perform an experimental study to check whether our regret bound agrees with the empirical performance of GP-PSRL. We consider a 2D navigation task with a 2-dimensional state space, a 2-dimensional action space and a known reward. Figure 2 shows the Bayesian regret over 100 episodes with a horizon of 20. Consistent with Theorem 4.11, smoother kernels exhibited lower regret. Figure 3 verifies that the rates given by the regret bound in Corollary 4.13 match the actual growth rates quite closely. Due to the necessity of approximations in both the GP and optimal control, a mismatch between theory and practice is expected, but minimal. Appendix I describes the details of the experiments in more detail and contains a plot of the Bayesian regret against $H$. We find that the empirical growth rate of the Bayesian regret is approximately linear in $H$.

## 6. Conclusion

We have established a regret bound for GP-based reinforcement learning that has an improved growth-rate in $T$, and accommodates both unbounded state spaces and weak smoothness assumptions. We provide some further discussion about our work in Appendix A.

## Acknowledgements

We would like to thank Gergely Neu, Antoine Moulin, Ludovic Schwartz, Lorenzo Croissant and Aad van der Vaart for helpful discussions and correspondence. We would also like to thank our anonymous reviewers for their helpful feedback. Hamish was funded by the European Research Council (ERC), under the European Union's Horizon 2020 research and innovation programme (grant agreement 950180). Joe and Ingmar are supported by an EPSRC Programme Grant (EP/V000748/1). Joe and Jan are supported by the grant "Einrichtung eines Labors des Deutschen Forschungszentrum für Künstliche Intelligenz (DFKI) an der Technischen Universität Darmstadt" and partially supported by the German Federal Ministry of Research, Technology and Space (BMFTR) under the Robotics Institute Germany (RIG).

## Impact Statement

This paper presents work whose goal is to advance the field of Machine Learning. There are many potential societal consequences of our work, none of which we feel must be specifically highlighted here.

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

## A. Discussion

We provide some discussion about our work below.

**Optimality of the regret bound.** The regret bound in Theorem 4.11 has an improved growth rate in $T$ compared to the other regret bounds listed in Table 1. However, it is not clear whether it has the best possible growth rate in $T$ in the sense that it matches the growth rate of a lower bound of some sort. We are not aware of any lower bounds for the Bayesian regret in the setting that we consider, but some lower bounds are available for the Bayesian regret in Gaussian process bandits. This corresponds to the special case where $H = 1$ and there is no state (or a single fixed state). Scarlett (2018) showed that for Gaussian process bandits with one-dimensional actions, the Bayesian regret behaves as $\Omega(\sqrt{T})$ and $\mathcal{O}(\sqrt{T \log(T)})$. Even when the actions are one-dimensional, the information gain can grow faster than logarithmically in $T$, suggesting that the rate of $\sqrt{\gamma_T T}$ is not tight in general. However, Rusmevichientong & Tsitsiklis (2010) showed that dimension one is something of a special case when it comes to the Bayesian regret. In particular, the Bayesian regret for one-dimensional linear bandits behaves as $\mathcal{O}(\log(T))$, whereas in dimension greater than one, the Bayesian regret behaves as $\Theta(\sqrt{T})$. In summary, the rate of $\sqrt{\gamma_T T}$ may or may not be optimal.

**Worst-case regret bounds.** It is often desirable to have worst-case regret bounds, which hold for any fixed MDP in some pre-specified set of MDPs. For instance, one could design an algorithm that works well whenever the dynamics of the MDP are given by a function in a ball of an RKHS (Chowdhury & Gopalan, 2019; Kakade et al., 2020; Curi et al., 2020). It would be interesting to investigate whether GP-PSRL satisfies a worst-case regret bound of the order $H\sqrt{\gamma_T T}$. One of the main difficulties that would need to be resolved is that Lemma 4.8 no longer holds, since $f^\star$ and $f^{(n)}$ no longer have the same conditional distribution. A potential solution is to use an "optimistic" or "feel-good" version of posterior sampling (Zhang, 2022; Neu, Papini, and Schwartz, 2024), in which one replaces $f^{(n)}$, a random draw from the Bayesian posterior, with a random draw from an optimistic version of the posterior. This would fix the previous problem (i.e. an inequality similar to the identity in Lemma 4.8 now holds). However, the optimistic posterior is not a Gaussian process, which complicates the matter of bounding the estimation errors. Nevertheless, the regret analyses of Zhang (2022) and Neu, Papini, and Schwartz (2024) both work by decoupling the standard likelihood part of the optimistic posterior from the optimistic correction, and then dealing with two parts separately, which means there is a bit of hope that some of the techniques we used to bound the estimation error might also be useful for proving worst-case regret bounds.

**Locally bounded kernel functions.** The assumption that the kernel function is uniformly bounded rules out some commonly used kernels, such as the linear kernel, which are only locally bounded. By locally bounded, we mean that there is a known function $C : (0, \infty) \to \mathbb{R}$ such that for all $R > 0$, the kernel function is bounded by $C(R)$ on a ball of radius $R$. While our analysis does not break completely when the uniform boundedness assumption is relaxed to local boundedness, our tail bound for the norm of the largest state in Lemma 4.7 would become much worse. In particular, the value of $R$ in (9) may pick up exponential dependence on $H$. To obtain a result comparable to Lemma 4.7 with locally bounded kernels, we expect that it is necessary to restrict the algorithm to play stabilizing policies.

## B. Additional Related Work

We describe some related work in more detail. In this section, we only show the dependence of the regret on $H, T$ and any problem-dependent quantities. As well as this, $\widetilde{\mathcal{O}}(\cdot)$ suppresses all polylogarithmic factors.

**Bayesian regret bounds.** Several works have studied the Bayesian regret in the setting described in Section 3, or in very similar settings. Osband & Van Roy (2014) assume that the transition kernel has a mean function ($f^\star$) that belongs to a set of functions $\mathcal{F}$ that has bounded eluder dimension (Russo & Van Roy, 2013). In addition, it is assumed that the prior used by PSRL has support contained in $\mathcal{F}$. Osband & Van Roy (2014) prove a regret bound of the order $\widetilde{\mathcal{O}}(L\sqrt{d_K(\mathcal{F})d_E(\mathcal{F})T})$, where $L$ is a problem-dependent quantity that hides dependence on $H$ (cf. Equation 3 in Osband & Van Roy, 2014), $d_K(\mathcal{F})$ is the Kolmogorov dimension of $\mathcal{F}$ (more or less the metric entropy/log-covering number of $\mathcal{F}$), and $d_E(\mathcal{F})$ is the eluder dimension of $\mathcal{F}$ at precision $1/T$. Most relevant to our setting is the case where $\mathcal{F}$ is a ball in an RKHS. In this case, it turns out that both the Kolmogorov dimension and the eluder dimension are roughly proportional to the maximum information gain (cf. Huang et al., 2021). For the sake of comparison, we can think of this bound as being roughly of the order $\widetilde{\mathcal{O}}(L\Gamma_T\sqrt{T})$. Note, however, that the eluder dimension of an RKHS ball is typically only sublinear in $T$ when the state space is bounded.

In the same setting that we consider, Chowdhury & Gopalan (2019) proved Bayesian regret bounds for PSRL with two kinds of priors. If the prior is any distribution with support contained within an RKHS ball, then Theorem 2 in Chowdhury &

Gopalan (2019) gives a Bayesian regret bound of the order $\widetilde{\mathcal{O}}(L\Gamma_{d_s T}\sqrt{T})$, where $L$ is the same problem-dependent quantity introduced in Osband & Van Roy (2014). If the prior is a Gaussian process, then Theorem 4 in Chowdhury & Gopalan (2019) gives a Bayesian regret bound of the order $\widetilde{\mathcal{O}}(Le^{\Gamma_{d_s H}}\sqrt{\Gamma_{d_s T}T})$. All of the regret bounds in Chowdhury & Gopalan (2019) assume that the state space is bounded, which is incompatible with the assumption that the prior is a Gaussian process and the assumption that the states are subject to Gaussian noise.

Fan & Ming (2021) studied the Bayesian regret of GP-PSRL in the same setting that we consider. They removed the quantity $L$ that appeared in previous regret bounds using a so-called "property derived from noises with symmetric probability distribution". We use the same idea in our proof of Lemma 4.9. Although the result is stated for linear kernels, Theorem 1 in Fan & Ming (2021) would give a Bayesian regret bound for general GP priors of the order $\widetilde{\mathcal{O}}(H^{3/2}\Gamma_N\sqrt{T})$. However, the proof of Lemma 3 in Fan & Ming (2021) does not account for the fact that there is dependence between $f^\star$ and the state-action pairs visited by the algorithm. In particular, $f^\star(\boldsymbol{x}_{n,h})$ is treated as a (conditionally on $\mathcal{F}_{n-1}$) Gaussian random variable, which is not necessarily the case.

**Worst-case regret bounds.** Several works have studied the worst-case regret in a "frequentist" version of the problem in Section 3, in which the mean function $f^\star$ is a fixed function in an RKHS ball. Theorem 1 in Chowdhury & Gopalan (2019) gives a worst-case regret bound for an optimistic UCRL algorithm, which is of the order $\widetilde{\mathcal{O}}(L\Gamma_{d_s T}\sqrt{T})$. Kakade et al. (2020) developed confidence sets for the dynamics using a regularized least squares estimate and designed an algorithm that plays optimistically with respect to this confidence set. This algorithm satisfies a worst-case regret bound of the order $\widetilde{\mathcal{O}}(H\widetilde{\gamma}_T\sqrt{T})$, where $\widetilde{\gamma}_T$ is a different version of the expected information gain. Unlike $\gamma_T(\sigma^2, R)$ defined in (5), $\widetilde{\gamma}_T$ is maximized with respect to the algorithm used to generate the states and actions. In addition, $\widetilde{\gamma}_T$ does not contain an indicator variable of the form $\mathbb{I}\{\sup_{n\in[N],h\in[H]}\|\boldsymbol{x}_{n,h}\|_2 \leq R\}$ inside the expectation. This means that standard upper bounds for the maximum information gain over a bounded domain (see e.g. Srinivas et al., 2012; Vakili et al., 2021) cannot be used as upper bounds for $\widetilde{\gamma}_T$. As a result, Kakade et al. (2020) are only able to give fully explicit growth rates for special cases where the underlying RKHS is finite-dimensional.

Curi et al. (2020) proposed another optimistic algorithm for this setting, which introduces "hallucinated controls" into the predictions of the model and then performs a greedy policy search in this augmented system. The resulting algorithm is shown to satisfy a worst-case regret bound of the order $\widetilde{\mathcal{O}}(H^{3/2}\Gamma_T^{(H+1)/2}\sqrt{T})$. The issue of the unbounded state space is explicitly addressed by Curi et al. (2020). Curi et al. (2020) derived a tail bound for the norm of the largest state encountered by their algorithm (cf. their Lemma 26) and used it to prove another worst-case regret bound (cf. their Theorem 5) that accounts for the fact that the state space is unbounded. However, the tail bound in Lemma 26 in Curi et al. (2020) only ensures that, with high probability, all states are contained within a ball with a radius that grows exponentially in $T$. Recall that Lemma 4.7 states that, when $f^\star$ is a random draw from a GP, the norm of the largest state grows as $\sqrt{(d_s + d_s)\log(T)}$.

**Posterior sampling with multi-output GPs.** Recent work has applied posterior sampling to MDPs in which the reward function and the dynamics are drawn from a multi-output GP. In this setting, correlations between the state-dimensions and the reward function can be exploited to reduce the regret suffered. Bayrooti et al. (2025a) propose an optimistic posterior sampling algorithm, but do not provide a regret bound. Bayrooti et al. (2025b) establish a regret bound for PSRL with multi-output GPs, which captures correlations between the reward function and the dynamics. However, the regret bound only applies to MDPs with deterministic dynamics, so it is not comparable to ours. Moreover, it is implicitly assumed that the state space is bounded, and all value functions are assumed to have bounded first and second derivatives.

## C. Additional Definitions

### C.1. Separable Gaussian Processes

For the chaining arguments to go through, we need to make a technical assumption, which is that the Gaussian processes we work with are separable. Fortunately, all the Gaussian processes that we consider are separable, so this is not a problem. Any random process is separable if it satisfied the property in the definition below.

**Definition C.1.** A random process $(f(\boldsymbol{x}))_{\boldsymbol{x}\in\mathcal{Z}}$ is called separable if there is a countable set $\mathcal{Z}_0 \subseteq \mathcal{Z}$ such that

$$f(\boldsymbol{x}) \in \lim_{\boldsymbol{z}\to\boldsymbol{x}, \boldsymbol{z}\in\mathcal{Z}_0} f(\boldsymbol{z}) \text{ for all } \boldsymbol{x} \in \mathcal{Z} \text{ a.s.}.$$

A consequence of $f$ being separable is that there exists a countable subset $\mathcal{Z}_0 \subseteq \mathcal{Z}$ such that $\sup_{\boldsymbol{x}\in\mathcal{Z}} f(\boldsymbol{x}) = \sup_{\boldsymbol{x}\in\mathcal{Z}_0} f(\boldsymbol{x})$ almost surely.

## C.2. Covering Numbers and Proper Covering Numbers

In several places, we will make use of $\varepsilon$-covers of subsets of $\mathbb{R}^{d_s+d_a}$. In general, it is preferable that we do not restrict the elements of a cover of some set to themselves be elements of the set being covered. On one occasion however, it will be important that the elements of the cover of some set are also elements of the set being covered. We therefore distinguish between two types of covers.

**Definition C.2** ($\varepsilon$-covers and covering numbers). Consider the metric space $(\mathbb{R}^{d_s+d_a}, d)$. A set $M \subseteq \mathbb{R}^{d_s+d_a}$ is an $\varepsilon$-cover for $\mathcal{Z} \subseteq \mathbb{R}^{d_s+d_a}$ (w.r.t. the metric $d$) if for all $x \in \mathcal{Z}$, there exists $y \in M$ such that $d(x, y) \le \varepsilon$. The $\varepsilon$-covering number $\mathsf{N}(\mathcal{Z}, d, \varepsilon)$ of $\mathcal{Z}$ w.r.t. the metric $d$ is the cardinality of the smallest $\varepsilon$-cover for $\mathcal{Z}$.

Note that an $\varepsilon$-cover of $\mathcal{Z}$ is not required to be a subset of $\mathbb{R}^{d_s+d_a}$. This has the advantage that the $\varepsilon$-covering number satisfies a certain monotonicity property. In particular, for two sets $\mathcal{Z}_1 \subseteq \mathcal{Z}_2 \subseteq \mathbb{R}^{d_s+d_a}$ and any metric $d$, any $\varepsilon$-cover for $\mathcal{Z}_2$ is also an $\varepsilon$-cover for $\mathcal{Z}_1$, which means

$$\mathsf{N}(\mathcal{Z}_1, d, \varepsilon) \le \mathsf{N}(\mathcal{Z}_2, d, \varepsilon).$$

Next, we define proper $\varepsilon$-covers and proper covering numbers.

**Definition C.3** (Proper $\varepsilon$-covers and proper covering numbers). Consider the metric space $(\mathbb{R}^{d_s+d_a}, d)$ and a subset $\mathcal{Z} \subseteq \mathbb{R}^{d_s+d_a}$. A set $M \subseteq \mathcal{Z}$ is a proper $\varepsilon$-cover for $\mathcal{Z}$ if for all $x \in \mathcal{Z}$, there exists $y \in M$ such that $d(x, y) \le \varepsilon$. The proper $\varepsilon$-covering number $\mathsf{N}_{\mathrm{pr}}(\mathcal{Z}, d, \varepsilon)$ of $\mathcal{Z}$ w.r.t. the metric $d$ is the cardinality of the smallest proper $\varepsilon$-cover for $\mathcal{Z}$.

The proper $\varepsilon$-covering number does not satisfy the same monotonicity property. For example, $\mathsf{N}_{\mathrm{pr}}([0, 1], d_2, 1/2) = 1$, whereas $\mathsf{N}_{\mathrm{pr}}([0, 1/2) \cup (1/2, 1], d_2, 1/2) = 2$.

# D. Tail Bounds for Suprema of Gaussian Processes

We prove the results given in Section 4.2. In Appendix D.1, we prove Lemma 4.5 and we use bounds on the covering number and diameter of $\mathbb{B}^{d_s+d_a}(R)$ w.r.t $d_c$ to obtain an explicit upper bound for the expected supremum. In Appendix D.2, we prove Lemma 4.4. Finally, in Appendix D.3, we prove Lemma 4.6.

## D.1. Expected Suprema of Gaussian Processes

To apply the chaining method to GPs with kernels that satisfy our smoothness and boundedness assumptions, we need upper bounds on both the covering numbers and the diameter (w.r.t $d_c$) of a Euclidean ball with a given radius. We will use the following bound on the covering number of a Euclidean ball (not necessarily centered at the origin) with respect to the Euclidean metric.

**Lemma D.1** (Lemma 5.7 in Wainwright (2019)). *For any $R > 0$, $\varepsilon > 0$ and $x \in \mathbb{R}^{d_s+d_a}$,*

$$\mathsf{N}(\mathbb{B}_x^{d_s+d_a}(R), d_2, \varepsilon) \le \left(1 + \frac{2R}{\varepsilon}\right)^{d_s+d_a}.$$

If, $\varepsilon \le R$, then this upper bound can be simplified slightly to

$$\mathsf{N}(\mathbb{B}_x^{d_s+d_a}(R), d_2, \varepsilon) \le \left(\frac{3R}{\varepsilon}\right)^{d_s+d_a}.$$

For any kernel that satisfies Assumption 3.4, we can turn this into an upper bound for the covering number of a Euclidean ball with respect to the natural distance associated with the kernel.

**Lemma D.2.** *Suppose that a kernel $c$ satisfies Assumption 3.4, and let $d_c$ denote the natural distance associated with $c$. Then for every $R \ge 0$,*

$$\mathsf{N}(\mathbb{B}^{d_s+d_a}(R), d_c, \varepsilon) \le \left(1 + \frac{2R(2L)^{1/\alpha}}{\varepsilon^{2/\alpha}}\right)^{d_s+d_a}.$$

*Proof.* Lemma D.1 tells us that

$$\mathsf{N}(\mathbb{B}^{d_s+d_a}(R), d_2, \varepsilon) \leq \left(1 + \frac{2R}{\varepsilon}\right)^{d_s+d_a}.$$

Thus for any $\varepsilon_0 > 0$, we can construct an $\varepsilon_0$-cover $M$ of $\mathbb{B}^{d_s+d_a}(R)$ in the metric $d_2$, such that $|M| \leq (1 + \frac{2R}{\varepsilon_0})^{d_s+d_a}$. By Assumption 3.4, we have

$$\begin{aligned}
d_c^2(\boldsymbol{x}, \boldsymbol{y}) &= c(\boldsymbol{x}, \boldsymbol{x}) - 2c(\boldsymbol{x}, \boldsymbol{y}) + c(\boldsymbol{y}, \boldsymbol{y}) \\
&\leq |c(\boldsymbol{x}, \boldsymbol{x}) - c(\boldsymbol{x}, \boldsymbol{y})| + |c(\boldsymbol{y}, \boldsymbol{x}) - c(\boldsymbol{y}, \boldsymbol{y})| \\
&\leq 2L \|\boldsymbol{x} - \boldsymbol{y}\|_2^{\alpha}.
\end{aligned}$$

In particular, $d_c(\boldsymbol{x}, \boldsymbol{y}) \leq \sqrt{2L d_2^{\alpha}(\boldsymbol{x}, \boldsymbol{y})}$. Therefore, if we choose $\varepsilon_0 = (\frac{\varepsilon^2}{2L})^{1/\alpha}$, then $M$ is an $\varepsilon$-covering of $\mathbb{B}^{d_s+d_a}(R)$ in the metric $d_c$. $\qquad\square$

Recall that for any subset $\mathcal{Z} \subseteq \mathbb{R}^{d_s+d_a}$ and any metric $d$, the diameter of $\mathcal{Z}$ is $\operatorname{diam}_d(\mathcal{Z}) := \sup_{\boldsymbol{x}, \boldsymbol{y} \in \mathcal{Z}} d(\boldsymbol{x}, \boldsymbol{y})$. For any kernel that satisfies Assumption 3.3 we can upper bound the diameter of $\mathbb{B}^{d_s+d_a}(R)$ w.r.t. the natural distance $d_c$.

**Lemma D.3.** *Suppose that a kernel $c : \mathbb{R}^{d_s+d_a} \times \mathbb{R}^{d_s+d_a} \to \mathbb{R}$ satisfies Assumption 3.3, and let $d_c$ denote the natural distance associated with c. Then for every $R \geq 0$,*

$$\operatorname{diam}_{d_c}(\mathbb{B}^{d_s+d_a}(R)) \leq 2\sqrt{C}.$$

*Proof.* By Assumption 3.3, we obtain

$$d_c(\boldsymbol{x}, \boldsymbol{y}) = \sqrt{c(\boldsymbol{x}, \boldsymbol{x}) - 2c(\boldsymbol{x}, \boldsymbol{y}) + c(\boldsymbol{y}, \boldsymbol{y})} \leq \sqrt{4C}.$$

This concludes the proof. $\qquad\square$

The last tool we need before we can apply the chaining method is a concentration inequality for $\|f(\boldsymbol{x})\|_2$. Since the Euclidean norm is 1-Lipschitz w.r.t. the Euclidean metric, we can use Lemma 4.3 to show that $\|f(\boldsymbol{x})\|_2$ is sub-Gaussian.

**Corollary D.4.** *Let $f_1, \ldots, f_{d_s} \sim \mathcal{GP}(0, c(\boldsymbol{x}, \boldsymbol{y}))$ be independent, centered Gaussian processes and let $f = (f_1, \ldots, f_{d_s})$. For any $\boldsymbol{x}, \boldsymbol{y} \in \mathbb{R}^{d_s+d_a}$, $\|f(\boldsymbol{x}) - f(\boldsymbol{y})\|_2$ is $d_c(\boldsymbol{x}, \boldsymbol{y})$-sub-Gaussian. In particular, for all $\lambda \in \mathbb{R}$,*

$$\mathbb{E}\left[\exp\left(\lambda\left(\|f(\boldsymbol{x}) - f(\boldsymbol{y})\|_2 - \mathbb{E}[\|f(\boldsymbol{x}) - f(\boldsymbol{y})\|_2]\right)\right)\right] \leq \exp\left(\frac{\lambda^2 d_c^2(\boldsymbol{x}, \boldsymbol{y})}{2}\right). \tag{13}$$

*Proof.* Let $g : \mathbb{R}^{d_s} \to \mathbb{R}$ be the function $g(\boldsymbol{x}) := \|\boldsymbol{x}\|_2$. The reverse triangle inequality states that, for all $\boldsymbol{v}, \boldsymbol{u} \in \mathbb{R}^{d_s}$

$$g(\boldsymbol{v}) - g(\boldsymbol{u}) = \|\boldsymbol{v}\|_2 - \|\boldsymbol{u}\|_2 \leq \|\boldsymbol{v} - \boldsymbol{u}\|_2.$$

It follows that $g$ is 1-Lipschitz w.r.t. the Euclidean metric. If $\boldsymbol{x}$ and $\boldsymbol{y}$ are such that $\|f(\boldsymbol{x}) - f(\boldsymbol{y})\|_2$ is identically equal to 0, then (13) is trivially satisfied. Suppose that $\boldsymbol{x}$ and $\boldsymbol{y}$ are such that $\|f(\boldsymbol{x}) - f(\boldsymbol{y})\|_2$ is not identically 0. In this case, one can verify that

$$f(\boldsymbol{x}) - f(\boldsymbol{y}) \sim \mathcal{N}(0, d_c^2(\boldsymbol{x}, \boldsymbol{y})\boldsymbol{I}).$$

Thus $f(\boldsymbol{x}) - f(\boldsymbol{y}))/d_c(\boldsymbol{x}, \boldsymbol{y})$ is a vector of i.i.d. standard Gaussian random variables. By Lemma 4.3, for all $\lambda_0 \in \mathbb{R}$,

$$\mathbb{E}\left[\exp\left(\frac{\lambda_0}{d_c(\boldsymbol{x}, \boldsymbol{y})}\left(\|f(\boldsymbol{x}) - f(\boldsymbol{y})\|_2 - \mathbb{E}[\|f(\boldsymbol{x}) - f(\boldsymbol{y})\|_2]\right)\right)\right] \leq \exp\left(\frac{\lambda_0^2}{2}\right).$$

The claim follows by substituting $\lambda = \lambda_0/d_c(\boldsymbol{x}, \boldsymbol{y})$. $\qquad\square$

We can now prove the chaining argument in Lemma 4.5.

*Proof of Lemma 4.5.* We first prove the result in the finite case, where $\mathbb{B}^{d_s+d_a}(R)$ is replaced by any finite non-empty subset $\mathcal{Z} \subset \mathbb{B}^{d_s+d_a}(R)$. We then use separability (cf. Definition C.1) to remove this restriction.

By Lemma D.3, we know that $\mathcal{Z}$ has finite diameter. Let $k_0$ be the largest integer such that $2^{-k_0} \geq \operatorname{diam}_{d_c}(\mathcal{Z})$. For $k \geq k_0$, let $M_k$ be a $2^{-k}$-cover of $\mathcal{Z}$ w.r.t. $d_c$ such that $|M_k| = \mathsf{N}(\mathcal{Z}, d_c, 2^{-k})$. Also, for each $k \geq k_0$, define $\omega_k : \mathcal{Z} \to M_k$ such that $d_c(\boldsymbol{x}, \omega_k(\boldsymbol{x})) \leq 2^{-k}$ for all $\boldsymbol{x} \in \mathcal{Z}$. Since $2^{-k_0} \geq \operatorname{diam}_{d_c}(\mathcal{Z})$, we can take $M_{k_0} := \{\boldsymbol{x}_0\}$, where $\boldsymbol{x}_0$ is any point in $\mathcal{Z}$. Fix any $m \geq k_0$. By introducing a telescoping sum and then using the triangle inequality, we obtain

$$
\begin{aligned}
\mathbb{E}\left[\sup_{\boldsymbol{x}\in\mathcal{Z}}\|f(\boldsymbol{x})\|_2\right] &= \mathbb{E}\left[\sup_{\boldsymbol{x}\in\mathcal{Z}}\|f(\boldsymbol{x})\|_2 - \sup_{\boldsymbol{x}\in\mathcal{Z}}\|f(\omega_m(\boldsymbol{x}))\|_2\right] + \mathbb{E}\left[\|f(\boldsymbol{x}_0)\|_2\right] \\
&\quad + \sum_{k=k_0+1}^{m} \mathbb{E}\left[\sup_{\boldsymbol{x}\in\mathcal{Z}}\|f(\omega_k(\boldsymbol{x}))\|_2 - \sup_{\boldsymbol{x}\in\mathcal{Z}}\|f(\omega_{k-1}(\boldsymbol{x}))\|_2\right] \\
&\leq \mathbb{E}\left[\sup_{\boldsymbol{x}\in\mathcal{Z}}\left\{\|f(\boldsymbol{x}) - f(\omega_m(\boldsymbol{x}))\|_2\right\}\right] + \mathbb{E}\left[\|f(\boldsymbol{x}_0)\|_2\right] \\
&\quad + \sum_{k=k_0+1}^{m} \mathbb{E}\left[\sup_{\boldsymbol{x}\in\mathcal{Z}}\left\{\|f(\omega_k(\boldsymbol{x})) - f(\omega_{k-1}(\boldsymbol{x}))\|_2\right\}\right].
\end{aligned}
$$

Since $\mathcal{Z}$ is finite, we can choose $m$ to be large enough such that $d_c(\boldsymbol{x}, \omega_m(\boldsymbol{x})) = 0$ for all $\boldsymbol{x} \in \mathcal{Z}$, and so the first term disappears. Due to Assumption 3.3, the second term satisfies the upper bound

$$
\mathbb{E}\left[\|f(\boldsymbol{x}_0)\|_2\right] \leq \sqrt{\sum_{i=1}^{d_s}\mathbb{E}[(f_i(\boldsymbol{x}_0))^2]} \leq \sqrt{d_s c(\boldsymbol{x}_0, \boldsymbol{x}_0)} \leq \sqrt{C d_s}.
$$

All that remains is to control the sum. For all $\boldsymbol{x} \in \mathcal{Z}$,

$$
d_c(\omega_k(\boldsymbol{x}), \omega_{k-1}(\boldsymbol{x})) \leq d_c(\boldsymbol{x}, \omega_k(\boldsymbol{x})) + d_c(\boldsymbol{x}, \omega_{k-1}(\boldsymbol{x})) \leq 3 \cdot 2^{-k}.
$$

By Corollary D.4, it follows that $\|f(\omega_k(\boldsymbol{x})) - f(\omega_{k-1}(\boldsymbol{x}))\|_2$ is $3 \cdot 2^{-k}$-sub-Gaussian. We upper bound the $k^{\text{th}}$ term by adding and subtracting $\sup_{\boldsymbol{x}\in\mathcal{Z}}\mathbb{E}[\|f(\omega_k(\boldsymbol{x})) - f(\omega_{k-1}(\boldsymbol{x}))\|_2]$.

$$
\begin{aligned}
\mathbb{E}\left[\sup_{\boldsymbol{x}\in\mathcal{Z}}\left\{\|f(\omega_k(\boldsymbol{x})) - f(\omega_{k-1}(\boldsymbol{x}))\|_2\right\}\right] &\leq \sup_{\boldsymbol{x}\in\mathcal{Z}}\mathbb{E}[\|f(\omega_k(\boldsymbol{x})) - f(\omega_{k-1}(\boldsymbol{x}))\|_2] \\
&\quad + \mathbb{E}\left[\sup_{\boldsymbol{x}\in\mathcal{Z}}\left\{\|f(\omega_k(\boldsymbol{x})) - f(\omega_{k-1}(\boldsymbol{x}))\|_2 - \mathbb{E}[\|f(\omega_k(\boldsymbol{x})) - f(\omega_{k-1}(\boldsymbol{x}))\|_2]\right\}\right].
\end{aligned}
$$

Since $\|f(\omega_k(\boldsymbol{x})) - f(\omega_{k-1}(\boldsymbol{x}))\|_2$ is $3 \cdot 2^{-k}$-sub-Gaussian, the first term satisfies

$$
\sup_{\boldsymbol{x}\in\mathcal{Z}}\mathbb{E}[\|f(\omega_k(\boldsymbol{x})) - f(\omega_{k-1}(\boldsymbol{x}))\|_2] \leq \sup_{\boldsymbol{x}\in\mathcal{Z}}\left(\mathbb{E}[\|f(\omega_k(\boldsymbol{x})) - f(\omega_{k-1}(\boldsymbol{x}))\|_2^2]\right)^{1/2} \leq 3 \cdot 2^{-k}.
$$

The second term can be upper bounded by using Jensen's inequality and the sub-Gaussian property. To save space, let us write

$$
g(\boldsymbol{y}, \boldsymbol{z}) := \|f(\boldsymbol{y}) - f(\boldsymbol{z})\|_2 - \mathbb{E}[\|f(\boldsymbol{y}) - f(\boldsymbol{z})\|_2]\}.
$$

Also, let us define the set

$$
M := \{(\omega_k(\boldsymbol{x}), \omega_{k-1}(\boldsymbol{x})) : \boldsymbol{x} \in \mathcal{Z}\} \subseteq M_k \times M_{k-1}.
$$

The cardinality of $M$ at most $|M_k||M_{k-1}| \leq |M_k|^2$. From our above discussion, for each $(\boldsymbol{y}, \boldsymbol{z}) \in M$, $g(\boldsymbol{y}, \boldsymbol{z})$ is centered

and $3 \cdot 2^{-k}$-sub-Gaussian. By Jensen's inequality, for any $\lambda > 0$,

$$
\begin{aligned}
\mathbb{E}\left[\sup_{\boldsymbol{x} \in \mathcal{Z}} g(\omega_k(\boldsymbol{x}), \omega_{k-1}(\boldsymbol{x}))\right] &= \mathbb{E}\left[\frac{1}{\lambda} \log \exp\left(\lambda \sup_{\boldsymbol{x} \in \mathcal{Z}} g(\omega_k(\boldsymbol{x}), \omega_{k-1}(\boldsymbol{x}))\right)\right] \\
&\leq \frac{1}{\lambda} \log \mathbb{E}\left[\sup_{\boldsymbol{x} \in \mathcal{Z}} \exp\left(\lambda g(\omega_k(\boldsymbol{x}), \omega_{k-1}(\boldsymbol{x}))\right)\right] \\
&\leq \frac{1}{\lambda} \log \sum_{(\boldsymbol{y}, \boldsymbol{z}) \in M} \mathbb{E}\left[\exp\left(\lambda g(\boldsymbol{y}, \boldsymbol{z})\right)\right] \\
&\leq \frac{2}{\lambda} \log |M_k| + \frac{\lambda(3 \cdot 2^{-k})^2}{2} \, .
\end{aligned}
$$

If we choose $\lambda = \frac{2\sqrt{\log|M_k|}}{3 \cdot 2^{-k}}$, we get the inequality

$$
\mathbb{E}\left[\sup_{\boldsymbol{x} \in \mathcal{Z}} g(\omega_k(\boldsymbol{x}), \omega_{k-1}(\boldsymbol{x}))\right] \leq 6 \cdot 2^{-k} \sqrt{\log|M_k|} \, .
$$

Combining everything so far, we have

$$
\mathbb{E}\left[\sup_{\boldsymbol{x} \in \mathcal{Z}} \|f(\boldsymbol{x})\|_2\right] \leq \sqrt{Cd_s} + \sum_{k=k_0+1}^{m} \left\{6 \cdot 2^{-k} \sqrt{\log \mathsf{N}(\mathcal{Z}, d_c, 2^{-k})} + 3 \cdot 2^{-k}\right\} \, .
$$

We separate the sum, and upper bound each part separately. For any $D \in \mathbb{R}$ and any $k \in \mathbb{Z}$, we have

$$
2^{-k} D = 2(2^{-k} - 2^{-k-1})D = \int_{2^{-k-1}}^{2^{-k}} D \mathrm{d}\varepsilon \, .
$$

Therefore, the first part of the sum satisfies

$$
\begin{aligned}
\sum_{k=k_0+1}^{m} 6 \cdot 2^{-k} \sqrt{\log \mathsf{N}(\mathcal{Z}, d_c, 2^{-k})} &\leq \sum_{k \in \mathbb{Z}} 6 \cdot 2^{-k} \sqrt{\log \mathsf{N}(\mathcal{Z}, d_c, 2^{-k})} \\
&= \sum_{k \in \mathbb{Z}} 12 \int_{2^{-k-1}}^{2^{-k}} \sqrt{\log \mathsf{N}(\mathcal{Z}, d_c, 2^{-k})} \mathrm{d}\varepsilon \\
&\leq \sum_{k \in \mathbb{Z}} 12 \int_{2^{-k-1}}^{2^{-k}} \sqrt{\log \mathsf{N}(\mathcal{Z}, d_c, \varepsilon)} \mathrm{d}\varepsilon \\
&= 12 \int_0^{\infty} \sqrt{\log \mathsf{N}(\mathcal{Z}, d_c, \varepsilon)} \mathrm{d}\varepsilon \, .
\end{aligned}
$$

The second part of the sum satisfies

$$
\sum_{k=k_0+1}^{m} 3 \cdot 2^{-k} \leq \sum_{k=k_0+1}^{\infty} 3 \cdot 2^{-k} = 3 \cdot 2^{-k_0} \sum_{k=1}^{\infty} 2^{-k} = 3 \cdot 2^{-k_0} \, .
$$

By Lemma D.3 and the fact that $k_0$ is the largest integer such that $2^{-k_0} \geq \mathrm{diam}_{d_c}(\mathcal{Z})$,

$$
3 \cdot 2^{-k_0} = 6 \cdot 2^{-k_0-1} \leq 6 \cdot \mathrm{diam}_{d_c}(\mathcal{Z}) \leq 12\sqrt{C} \, .
$$

Therefore, since the covering number is monotone w.r.t. $\mathcal{Z}$, for any finite subset $\mathcal{Z} \subset \mathbb{B}^{d_s+d_a}(R)$,

$$
\mathbb{E}\left[\sup_{\boldsymbol{x} \in \mathcal{Z}} \|f(\boldsymbol{x})\|_2\right] \leq 12 \int_0^{\infty} \sqrt{\log \mathsf{N}(\mathbb{B}^{d_s+d_a}(R), d_c, \varepsilon)} \mathrm{d}\varepsilon + 12\sqrt{C} + \sqrt{Cd_s} \, .
$$

Now suppose that $\mathcal{Z} \subseteq \mathbb{B}^{d_s+d_a}(R)$ does not necessarily have finite cardinality. By separability, there exists a countable subset $\mathcal{Z}_0 \subseteq \mathcal{Z}$ such that $\sup_{\boldsymbol{x} \in \mathcal{Z}} \|f(\boldsymbol{x})\|_2 = \sup_{\boldsymbol{x} \in \mathcal{Z}_0} \|f(\boldsymbol{x})\|_2$ almost surely. Let $\mathcal{Z}_0^{(n)}$ be the first $n$ elements $\mathcal{Z}_0$. By the monotone convergence theorem,

$$
\begin{aligned}
\mathbb{E}\left[\sup_{\boldsymbol{x} \in \mathcal{Z}} \|f(\boldsymbol{x})\|_2\right] &= \mathbb{E}\left[\sup_{\boldsymbol{x} \in \mathcal{Z}_0} \|f(\boldsymbol{x})\|_2\right] \\
&= \mathbb{E}\left[\lim_{n \to \infty} \sup_{\boldsymbol{x} \in \mathcal{Z}^{(n)}} \|f(\boldsymbol{x})\|_2\right] \\
&= \lim_{n \to \infty} \mathbb{E}\left[\sup_{\boldsymbol{x} \in \mathcal{Z}^{(n)}} \|f(\boldsymbol{x})\|_2\right] \\
&\leq 12 \int_0^\infty \sqrt{\log \mathsf{N}(\mathbb{B}^{d_s+d_a}(R), d_c, \varepsilon)} \mathrm{d}\varepsilon + 12\sqrt{C} + \sqrt{Cd_s} \,.
\end{aligned}
$$

This concludes the proof. $\qquad\qquad\square$

We specialize this result to obtain the following bound on the expected supremum.

**Lemma D.5.** *Suppose that Assumption 3.3 and Assumption 3.4 are satisfied. Let $f_1, \ldots, f_{d_s} \sim \mathcal{GP}(0, c(\boldsymbol{x}, \boldsymbol{y}))$ be independent Gaussian processes and let $f = (f_1, \ldots, f_{d_s})$. For any $R > 0$,*

$$
\mathbb{E}\left[\sup_{\boldsymbol{x} \in \mathbb{B}^{d_s+d_a}(R)} \|f(\boldsymbol{x})\|_2\right] \leq 42\alpha^{-1/2}\sqrt{C(d_s + d_a)\log(5 + 5R^\alpha L/C)} \,.
$$

*Proof.* Due to Lemma 4.5, we just need to bound the entropy integral $\int_0^\infty \sqrt{\log \mathsf{N}(\mathbb{B}^{d_s+d_a}(R), d_c, \varepsilon)} \mathrm{d}\varepsilon$. Using the bounds on the covering numbers (cf. Lemma D.2) and diameter (cf. Lemma D.3) of $\mathbb{B}^{d_s+d_a}(R)$ w.r.t. $d_c$, we obtain

$$
\int_0^\infty \sqrt{\log \mathsf{N}(\mathbb{B}^{d_s+d_a}(R), d_c, \varepsilon)} \mathrm{d}\varepsilon \leq \int_0^{2\sqrt{C}} \sqrt{(d_s + d_a)\log(1 + 2R(2L)^{1/\alpha}/\varepsilon^{2/\alpha})} \mathrm{d}\varepsilon \,.
$$

Using the substitution $\delta = \varepsilon/(2\sqrt{C})$, we can re-write this integral as

$$
\int_0^{2\sqrt{C}} \sqrt{(d_s + d_a)\log(1 + 2R(2L)^{1/\alpha}/\varepsilon^{2/\alpha})} \mathrm{d}\varepsilon = 2\sqrt{C} \int_0^1 \sqrt{(d_s + d_a)\log(1 + 2R(2L)^{1/\alpha}/(2\sqrt{C}\delta)^{2/\alpha})} \mathrm{d}\delta \,.
$$

For all $\delta \in (0, 1]$, $1 \leq 1/\delta^{2/\alpha}$, which means

$$
\sqrt{\log(1 + 2R(2L)^{1/\alpha}/(2\sqrt{C}\delta)^{2/\alpha})} \leq \sqrt{\log(1 + 2R(2L)^{1/\alpha}/(4C)^{1/\alpha})} + \sqrt{(2/\alpha)\log(1/\delta)} \,.
$$

Using this inequality and the identity $\int_0^1 \sqrt{\log(1/x)}\mathrm{d}x = \sqrt{\pi}/2$ (see e.g., 4.215 in Gradshteyn & Ryzhik, 2014), we obtain

$$
\int_0^1 \sqrt{\log(1 + 2R(2L)^{1/\alpha}/(2\sqrt{C}\delta)^{2/\alpha})}\mathrm{d}\delta \leq \sqrt{\log(1 + 2R(2L)^{1/\alpha}/(4C)^{1/\alpha})} + \alpha^{-1/2}\sqrt{\pi/2} \,.
$$

Combining everything so far, we have

$$
\int_0^\infty \sqrt{\log \mathsf{N}(\mathbb{B}^{d_s+d_a}(R), d_c, \varepsilon)} \mathrm{d}\varepsilon \leq 2\sqrt{C(d_s + d_a)}\left(\sqrt{\log(1 + 2R(2L)^{1/\alpha}/(4C)^{1/\alpha})} + \alpha^{-1/2}\sqrt{\pi/2}\right) \,. \tag{14}
$$

This upper bound on the entropy integral can be replaced by a simpler but looser bound. Since $x^{1/\alpha} + y^{1/\alpha} \leq (x + y)^{1/\alpha}$ for all $\alpha \in (0, 1]$ and $x, y \geq 0$, we have

$$
1 + \frac{2R(2L)^{1/\alpha}}{(4C)^{1/\alpha}} = 1^{1/\alpha} + \left(\frac{(2R)^\alpha 2L}{4C}\right)^{1/\alpha} \leq (1 + R^\alpha L/C)^{1/\alpha} \,.
$$

Using this inequality, and $\sqrt{a} + \sqrt{b} \le \sqrt{2(a+b)}$ for all $a, b \ge 0$, we can replace the bound in (14) by

$$
\int_0^\infty \sqrt{\log \mathsf{N}(\mathbb{B}^{d_s+d_a}(R), d_c, \varepsilon)} d\varepsilon \le 2\alpha^{-1/2}\sqrt{C(d_s + d_a)}\left(\sqrt{\log(1 + R^\alpha L/C)} + \sqrt{\pi/2}\right)
$$
$$
\le 2\sqrt{2}\alpha^{-1/2}\sqrt{C(d_s + d_a)\log(\exp(\pi/2)(1 + R^\alpha L/C))}
$$
$$
\le 2\sqrt{2}\alpha^{-1/2}\sqrt{C(d_s + d_a)\log(5 + 5R^\alpha L/C)},
$$

where in the last step we used the inequality $\exp(\pi/2) \le 5$. Plugging this bound on the entropy integral into Lemma 4.5, we obtain

$$
\mathbb{E}\left[\sup_{\boldsymbol{x} \in \mathbb{B}^{d_s+d_a}(R)} \|f(\boldsymbol{x})\|_2\right] \le 24\sqrt{2}\alpha^{-1/2}\sqrt{C(d_s + d_a)\log(5 + 5R^\alpha L/C)} + 12\sqrt{C} + \sqrt{Cd_s}.
$$

The sum of the last two terms satisfies

$$
12\sqrt{C} + \sqrt{Cd_s} \le (6\sqrt{2} + 1)\sqrt{C(d_s + d_a)} \le \frac{6\sqrt{2} + 1}{\sqrt{\log(5)}}\alpha^{-1/2}\sqrt{C(d_s + d_a)\log(5 + 5R^\alpha L/C)}.
$$

The result now follows from the inequality $24\sqrt{2} + (6\sqrt{2} + 1)\log^{-1/2}(5) \le 42$. $\qquad\square$

## D.2. Concentration of Suprema of Gaussian Processes

We prove Lemma 4.4, which states that $\sup_{\boldsymbol{x} \in \mathbb{B}^{d_s+d_a}(R)} \|f(\boldsymbol{x})\|_2$ is a sub-Gaussian random variable. First, we state and prove a technical lemma.

**Lemma D.6.** *For each $i \in [n]$, let $g_i : \mathbb{R}^{d_s+d_a} \to \mathbb{R}$ be a function which is L-Lipschitz w.r.t. the Euclidean norm. Then the function $\max(g_1, \dots, g_n) : \mathbb{R}^{d_s+d_a} \to \mathbb{R}$ given by $\max(g_1, \dots, g_n)(\boldsymbol{x}) := \max_{i \in [n]} g_i(\boldsymbol{x})$ is also L-Lipschitz w.r.t. the Euclidean norm.*

*Proof.* Fix $\boldsymbol{x}, \boldsymbol{y} \in \mathbb{R}^{d_s+d_a}$ and let $i^\star := \arg\max_{i \in [n]} g_i(\boldsymbol{x})$. Since $g_{i^\star}$ is $L$-Lipschitz, we have

$$
\max_{i \in [n]} g_i(\boldsymbol{x}) - \max_{i \in [n]} g_i(\boldsymbol{y}) \le g_{i^\star}(\boldsymbol{x}) - g_{i^\star}(\boldsymbol{y}) \le L\|\boldsymbol{x} - \boldsymbol{y}\|_2.
$$

A similar argument shows that $-(\max_{i \in [n]} g_i(\boldsymbol{x}) - \max_{i \in [n]} g_i(\boldsymbol{y})) \le L\|\boldsymbol{x} - \boldsymbol{y}\|_2$. $\qquad\square$

We can now prove Lemma 4.4.

*Proof of Lemma 4.4.* We first prove the result in the finite case, where $\mathbb{B}^{d_s+d_a}(R)$ is replaced by any finite subset $\mathcal{Z} \subset \mathbb{B}^{d_s+d_a}(R)$. We then use separability to remove this assumption (cf. Definition C.1).

Let $\mathcal{Z} = \{\boldsymbol{x}_1, \dots, \boldsymbol{x}_n\} \subset \mathbb{B}^{d_s+d_a}$ and for all $i \in [d_s]$ and $j \in [n]$, let $Z_{i,j} \sim \mathcal{N}(0, 1)$ be a standard Gaussian random variable. We would like to show that $\sup_{\boldsymbol{x} \in \mathcal{Z}} \|f(\boldsymbol{x})\|_2$ is equal in distribution to a Lipschitz function of $Z_{1,1}, \dots, Z_{d_s,n}$. By Lemma 4.3, we would then have that $\sup_{\boldsymbol{x} \in \mathcal{Z}} \|f(\boldsymbol{x})\|_2$ is sub-Gaussian. Since each component $f_i$ of $f$ is a zero-mean GP with covariance kernel $c$, we have $[f_i(\boldsymbol{x}_1), \dots, f_i(\boldsymbol{x}_n)]^\top \sim \mathcal{N}(0, \boldsymbol{C})$, where the $(i, j)^{\text{th}}$ entry of $\boldsymbol{C}$ is $C_{i,j} = c(\boldsymbol{x}_i, \boldsymbol{x}_j)$. Since the components $f_1, \dots, f_{d_s}$ are independent, we therefore have

$$
\begin{bmatrix} f_1(\boldsymbol{x}_1) \\ \vdots \\ f_1(\boldsymbol{x}_n) \\ \vdots \\ f_{d_s}(\boldsymbol{x}_1) \\ \vdots \\ f_{d_s}(\boldsymbol{x}_n) \end{bmatrix} \sim \mathcal{N}\left( \begin{bmatrix} 0 \\ \vdots \\ 0 \\ \vdots \\ 0 \\ \vdots \\ 0 \end{bmatrix}, \begin{bmatrix} C_{1,1} & \cdots & C_{1,n} & \cdots & 0 & \cdots & 0 \\ \vdots & \ddots & \vdots & & \vdots & \ddots & \vdots \\ C_{n,1} & \cdots & C_{n,n} & \cdots & 0 & \cdots & 0 \\ \vdots & & \vdots & \ddots & \vdots & & \vdots \\ 0 & \cdots & 0 & \cdots & C_{1,1} & \cdots & C_{1,n} \\ \vdots & \ddots & \vdots & & \vdots & \ddots & \vdots \\ 0 & \cdots & 0 & \cdots & C_{n,1} & \cdots & C_{n,n} \end{bmatrix} \right). \tag{15}
$$

For each $i \in [d_s]$ and $j \in [n]$, let us define the function $g_{i,j} : \mathbb{R}^{d_s n} \to \mathbb{R}$ by

$$g_{i,j}(z_{1,1}, \ldots, z_{d_s,n}) := \sum_{k=1}^{n} C_{j,k}^{1/2} z_{i,k} \,,$$

where $C_{j,k}^{1/2}$ is the $(j,k)^{\text{th}}$ entry of $\boldsymbol{C}^{1/2}$, and $\boldsymbol{C}^{1/2}$ is a symmetric square root of $\boldsymbol{C}$ (so $\boldsymbol{C}^{1/2}\boldsymbol{C}^{1/2} = \boldsymbol{C}$). In light of (15), we have $f_i(\boldsymbol{x}_j) \stackrel{\text{d}}{=} g_{i,j}(Z_{1,1}, \ldots, Z_{d_s,n})$, where $\stackrel{\text{d}}{=}$ means equality in distribution. For each $j \in [n]$, let us define $g_j : \mathbb{R}^{d_s n} \to \mathbb{R}$ by $g_j(z_{1,1}, \ldots, z_{d_s,n}) := (\sum_{i=1}^{d_s} g_{i,j}(z_{1,1}, \ldots, z_{d_s,n}))^{1/2}$. We will show that $g_j$ is $\sqrt{C}$-Lipschitz. We introduce the shorthand $\boldsymbol{z} := [z_{1,1}, \ldots, z_{d_s,n}]^\top$. Using the chain rule, we see that for any $\boldsymbol{z} \neq 0$, $l \in [d_s]$ and $m \in [n]$, the partial derivative of $g_j$ w.r.t. $z_{m,l}$ at $\boldsymbol{z}$ is

$$\begin{aligned}
\frac{\partial g_j}{\partial z_{l,m}}(\boldsymbol{z}) &= \frac{\partial}{\partial z_{l,m}} \left( \sum_{i=1}^{d_s} \left( \sum_{k=1}^{n} C_{j,k}^{1/2} z_{i,k} \right)^2 \right)^{1/2} \\
&= \frac{1}{2} \left( \frac{\partial}{\partial z_{l,m}} \sum_{i=1}^{d_s} \left( \sum_{k=1}^{n} C_{j,k}^{1/2} z_{i,k} \right)^2 \right) \left( \sum_{i=1}^{d_s} \left( \sum_{k=1}^{n} C_{j,k}^{1/2} z_{i,k} \right)^2 \right)^{-1/2} \\
&= C_{j,m}^{1/2} \sum_{k=1}^{n} C_{j,k}^{1/2} z_{l,k} \left( \sum_{i=1}^{d_s} \left( \sum_{k=1}^{n} C_{j,k}^{1/2} z_{i,k} \right)^2 \right)^{-1/2} .
\end{aligned}$$

Therefore, for any $\boldsymbol{z} \neq 0$, we have

$$\begin{aligned}
\|\nabla g_j(\boldsymbol{z})\|_2 &= \left( \sum_{l=1}^{d_s} \sum_{m=1}^{n} C_{j,m}^{1/2} C_{j,m}^{1/2} \left( \sum_{k=1}^{n} C_{j,k}^{1/2} z_{l,k} \right)^2 \left( \sum_{i=1}^{d_s} \left( \sum_{k=1}^{n} C_{j,k}^{1/2} z_{i,k} \right)^2 \right)^{-1} \right)^{1/2} \\
&= \left( \sum_{m=1}^{n} C_{j,m}^{1/2} C_{j,m}^{1/2} \sum_{l=1}^{d_s} \left( \sum_{k=1}^{n} C_{j,k}^{1/2} z_{l,k} \right)^2 \left( \sum_{i=1}^{d_s} \left( \sum_{k=1}^{n} C_{j,k}^{1/2} z_{i,k} \right)^2 \right)^{-1} \right)^{1/2} \\
&= \left( \sum_{m=1}^{n} C_{j,m}^{1/2} C_{j,m}^{1/2} \right)^{1/2} \\
&= \sqrt{C_{j,j}} \,,
\end{aligned}$$

where the identity $\sum_{m=1}^{n} C_{j,m}^{1/2} C_{j,m}^{1/2} = C_{j,j}$ follows from $\boldsymbol{C}^{1/2}\boldsymbol{C}^{1/2} = \boldsymbol{C}$. Since (by Assumption 3.3) $C_{j,j} \leq C$, we have $\|\nabla g_j(\boldsymbol{z})\|_2 \leq \sqrt{C}$ for all $z \in \mathbb{R}^{d_s n} \setminus \{0\}$. Thus we can conclude that $g_j$ is $\sqrt{C}$-Lipschitz on $\mathbb{R}^{d_s n}$. Let us now define the function $g : \mathbb{R}^{d_s n} \to \mathbb{R}$ by $g(\boldsymbol{z}) = \max_{j \in [n]} g_j(\boldsymbol{z})$. By Lemma D.6, $g$ is also $\sqrt{C}$-Lipschitz on $\mathbb{R}^{d_s n}$. Together with Lemma 4.3, this tells us that the random variable $g(Z_{1,1}, \ldots, Z_{d_s,n})$ is $\sqrt{C}$-sub-Gaussian. Since

$$\sup_{\boldsymbol{x} \in \mathcal{Z}} \|f(\boldsymbol{x})\|_2 \stackrel{\text{d}}{=} \sup_{j \in [n]} \sqrt{\sum_{i=1}^{d_s} g_{i,j}^2(Z_{1,1}, \ldots, Z_{d_s,n})} = g(Z_{1,1}, \ldots, Z_{d_s,n}) \,.$$

the random variable $\sup_{\boldsymbol{x} \in \mathcal{Z}} \|f(\boldsymbol{x})\|_2$ is also $\sqrt{C}$-sub-Gaussian. In particular, for any $u > 0$.

$$\mathbb{P}\left( \sup_{\boldsymbol{x} \in \mathcal{Z}} \|f(\boldsymbol{x})\|_2 - \mathbb{E}\left[ \sup_{\boldsymbol{x} \in \mathcal{Z}} \|f(\boldsymbol{x})\|_2 \right] \geq u \right) \leq \exp\left( -\frac{u^2}{2C} \right). \tag{16}$$

Now suppose that $\mathcal{Z} \subseteq \mathbb{B}^{d_s + d_a}(R)$ does not necessarily have finite cardinality. By separability, there exists a countable subset $\mathcal{Z}_0 \subseteq \mathcal{Z}$ such that $\sup_{\boldsymbol{x} \in \mathcal{Z}} \|f(\boldsymbol{x})\|_2 = \sup_{\boldsymbol{x} \in \mathcal{Z}_0} \|f(\boldsymbol{x})\|_2$ almost surely. Let us define $\mathcal{Z}_0^{(n)}$ to be the set that contains only the first $n$ elements of $\mathcal{Z}$, and let us define the random variable $E_n$ as

$$E_n := \mathbb{I}\left\{ \sup_{\boldsymbol{x} \in \mathcal{Z}_0^{(n)}} \|f(\boldsymbol{x})\|_2 - \mathbb{E}\left[ \sup_{\boldsymbol{x} \in \mathcal{Z}_0} \|f(\boldsymbol{x})\|_2 \right] \geq u \right\}.$$

Since $\mathbb{E}[\sup_{\boldsymbol{x}\in\mathcal{Z}_0}\|f(\boldsymbol{x})\|_2] \geq \mathbb{E}[\sup_{\boldsymbol{x}\in\mathcal{Z}_0^{(n)}}\|f(\boldsymbol{x})\|_2]$ for all $n \geq 1$, (16) tells us that $\mathbb{E}[E_n] \leq \exp(-u^2/(2C))$ for all $n \geq 1$. Whenever $n \leq n'$, $\sup_{\boldsymbol{x}\in\mathcal{Z}_0^{(n)}}\|f(\boldsymbol{x})\|_2 \leq \sup_{\boldsymbol{x}\in\mathcal{Z}_0^{(n')}}\|f(\boldsymbol{x})\|_2$ almost surely. This means that $E_n \leq E_{n'}$ almost surely. By the monotone convergence theorem,

$$
\begin{aligned}
\mathbb{P}\left(\sup_{\boldsymbol{x}\in\mathcal{Z}}\|f(\boldsymbol{x})\|_2 - \mathbb{E}\left[\sup_{\boldsymbol{x}\in\mathcal{Z}}\|f(\boldsymbol{x})\|_2\right] \geq u\right) &= \mathbb{P}\left(\sup_{\boldsymbol{x}\in\mathcal{Z}_0}\|f(\boldsymbol{x})\|_2 - \mathbb{E}\left[\sup_{\boldsymbol{x}\in\mathcal{Z}_0}\|f(\boldsymbol{x})\|_2\right] \geq u\right) \\
&= \mathbb{E}\left[\lim_{n\to\infty} E_n\right] \\
&= \lim_{n\to\infty}\mathbb{E}[E_n] \\
&\leq \exp\left(-\frac{u^2}{2C}\right).
\end{aligned}
$$

This concludes the proof. $\qquad\square$

### D.3. Proof of Lemma 4.6

Using the results from Appendix D.1 and Appendix D.2, we can now prove Lemma 4.6.

*Proof of Lemma 4.6.* Let us define $H := \mathbb{E}[\sup_{\boldsymbol{x}\in\mathbb{B}^{d_s+d_a}(R)}\|f(\boldsymbol{x})\|_2]$. By Lemma D.5, we have

$$
H \leq 42\alpha^{-1/2}\sqrt{C(d_s+d_a)\log(5+5R^\alpha L/C)} < \infty.
$$

By Lemma 4.4, for any $v \geq 0$,

$$
\mathbb{P}\left(\sup_{\boldsymbol{x}\in\mathbb{B}^{d_s+d_a}(R)}\|f(\boldsymbol{x})\|_2 \geq v + H\right) \leq \exp\left(-\frac{v^2}{2C}\right).
$$

Letting $u = v + H$, we have that for any $u \geq H$,

$$
\mathbb{P}\left(\sup_{\boldsymbol{x}\in\mathbb{B}^{d_s+d_a}(R)}\|f(\boldsymbol{x})\|_2 \geq u\right) \leq \exp\left(-\frac{(u-H)^2}{2C}\right).
$$

Whenever $u \geq 2H$, we have $u - H \geq u/2$, and so

$$
\exp\left(-\frac{(u-H)^2}{2C}\right) \leq \exp\left(-\frac{u^2}{8C}\right).
$$

The inequality $u \geq 2H$ is satisfied whenever $u \geq 84\alpha^{-1/2}\sqrt{C(d_s+d_a)\log(5+5R^\alpha L/C)}$ is satisfied. $\qquad\square$

## E. Tail Bounds for the Norm of the Largest State

In Appendix E.1, we state and prove some lemmas involving indicator functions that allow us to formalize the argument that the tails of the norm of each state decay rapidly as long as the previous state is bounded. In Appendix E.2, we upper bound the probability that at least one state has norm greater than some pre-specified threshold, and we then derive explicit values for the thresholds that ensure that this probability is of the order $1/T$. In Appendix E.3, we prove Lemma 4.7.

### E.1. A Trick With Indicator Functions

For $h = 1$, we use the convention $\mathbb{I}\{\cap_{j=1}^{h-1}A_j\} := 1$.

**Lemma E.1.** *For any finite collection of events $(A_h)_{h=1}^H$,*

$$
\mathbb{I}\{\cup_{h=1}^H A_h^{\mathsf{c}}\} = \sum_{h=1}^H \mathbb{I}\{A_h^{\mathsf{c}}\}\mathbb{I}\{\cap_{j=1}^{h-1}A_j\}.
$$

*Proof.* We use induction on $H$. For the base case $H = 1$, we have

$$\mathbb{I}\{\cup_{h=1}^{H} A_h^{\mathsf{c}}\} = \mathbb{I}\{A_1^{\mathsf{c}}\} = \sum_{h=1}^{H} \mathbb{I}\{A_h^{\mathsf{c}}\}\mathbb{I}\{\cap_{j=1}^{h-1} A_j\}.$$

Suppose inductively that for some $H \geq 1$,

$$\mathbb{I}\{\cup_{h=1}^{H} A_h^{\mathsf{c}}\} = \sum_{h=1}^{H} \mathbb{I}\{A_h^{\mathsf{c}}\}\mathbb{I}\{\cap_{j=1}^{h-1} A_j\}.$$

From this, it follows that

$$\begin{aligned}
\mathbb{I}\{\cup_{h=1}^{H+1} A_h^{\mathsf{c}}\} &= 1 - \mathbb{I}\{\cap_{h=1}^{H+1} A_h\} \\
&= 1 - \mathbb{I}\{A_{H+1}\}\mathbb{I}\{\cap_{h=1}^{H} A_h\} \\
&= 1 - (1 - \mathbb{I}\{A_{H+1}^{\mathsf{c}}\})(1 - \mathbb{I}\{\cup_{h=1}^{H} A_h^{\mathsf{c}}\}) \\
&= \mathbb{I}\{A_{H+1}^{\mathsf{c}}\}(1 - \mathbb{I}\{\cup_{h=1}^{H} A_h^{\mathsf{c}}\}) + \mathbb{I}\{\cup_{h=1}^{H} A_h^{\mathsf{c}}\} \\
&= \mathbb{I}\{A_{H+1}^{\mathsf{c}}\}\mathbb{I}\{\cap_{h=1}^{H} A_h\} + \sum_{h=1}^{H} \mathbb{I}\{A_h^{\mathsf{c}}\}\mathbb{I}\{\cap_{j=1}^{h-1} A_j\} \\
&= \sum_{h=1}^{H+1} \mathbb{I}\{A_h^{\mathsf{c}}\}\mathbb{I}\{\cap_{j=1}^{h-1} A_j\}.
\end{aligned}$$

This closes the induction. $\qquad\square$

The next lemma is basically the same as the union bound.

**Lemma E.2.** *For any $x_1, \ldots, x_n \in \mathbb{R}$ and $R \in \mathbb{R}$,*

$$\mathbb{I}\left\{\sum_{i=1}^{n} x_i > R\right\} \leq \sum_{i=1}^{n} \mathbb{I}\{x_i > R/n\}.$$

*Proof.* We upper bound the sum by the max. In particular,

$$\mathbb{I}\left\{\sum_{i=1}^{n} x_i > R\right\} \leq \mathbb{I}\{n \cdot \max_i(x_i) > R\} = \mathbb{I}\{\cup_{i=1}^{n}(x_i > R/n)\} \leq \sum_{i=1}^{n} \mathbb{I}\{x_i > R/n\}.$$

This is the bound that we wanted. $\qquad\square$

### E.2. Tail Bound for the Bad Event

We use the following bound on the tail probability of the norm of a Gaussian vector.

**Lemma E.3.** *For any $n \in [N]$, $h \in [H]$ and $u \geq 2\sigma\sqrt{d_s}$,*

$$\mathbb{P}\big(\|\varepsilon_{n,h}\|_2 > u\big) \leq \exp\left(-\frac{u^2}{8\sigma^2}\right).$$

*Proof.* Since the Euclidean norm is 1-Lipschitz, Lemma 4.3 tells us that $\|\varepsilon_{n,h}\|_2$ is $\sigma$-sub-Gaussian. By Jensen's inequality,

$$\mathbb{E}[\|\varepsilon_{n,h}\|_2] \leq \big(\mathbb{E}[\|\varepsilon_{n,h}\|_2^2]\big)^{1/2} = \sigma\sqrt{d_s}.$$

Since $u \geq 2\sigma\sqrt{d_s}$, we have

$$\mathbb{P}\big(\|\varepsilon_{n,h}\|_2 > u\big) \leq \mathbb{P}\big(\|\varepsilon_{n,h}\|_2 - \mathbb{E}[\|\varepsilon_{n,h}\|_2] > u/2\big) \leq \exp\left(-\frac{u^2}{8\sigma^2}\right).$$

This concludes the proof. $\qquad\square$

Let $R_1, R_2, \ldots, R_H$ be a sequence of radii, whose values we will choose later. For each $n \in [N]$ and $h \in [H]$, let us define the event

$$A_{n,h} := \{\|\boldsymbol{s}_{n,h}\|_2 \le R_h\}.$$

In this section, we will be interested in the event

$$A := \cap_{n=1}^{N} \cap_{h=1}^{H} A_{n,h}.$$

This is the event that the Euclidean norm of each state $\boldsymbol{s}_{n,h}$ is at most $R_h$. We want to show that if the radii $R_1, \ldots, R_H$ are large enough, then the complement $A^{\mathsf{c}}$ can be made to occur with arbitrarily small probability. For each $h \in [H]$, let us define $\widetilde{R}_h := \sqrt{R_h^2 + R_a^2}$. For certain choices of $R_1, \ldots, R_H$, the following lemma provides an upper bound on the probability that $A^{\mathsf{c}}$ occurs.

**Lemma E.4.** *For any sequence of radii $R_1, \ldots, R_H$ such that $R_1 > 2\sigma\sqrt{d_s}$ and for all $h \ge 1$,*

$$R_h \ge 168\alpha^{-1/2}\sqrt{\max(C, \sigma^2)(d_s + d_a)\log(5 + 5\widetilde{R}_{h-1}^{\alpha}L/C)}, \tag{17}$$

*we have*

$$\mathbb{P}(A^{\mathsf{c}}) \le N\exp\left(-\frac{R_1^2}{8\sigma^2}\right) + N\sum_{h=2}^{H}\left[\exp\left(-\frac{R_h^2}{32C}\right) + \exp\left(-\frac{R_h^2}{32\sigma^2}\right)\right].$$

*Proof.* Using Lemma E.1,

$$\begin{aligned}
\mathbb{P}(A^{\mathsf{c}}) &= \mathbb{E}\left[\mathbb{I}\{\cup_{n=1}^{N}\cup_{h=1}^{H}A_{n,h}^{\mathsf{c}}\}\right] \\
&= \sum_{n=1}^{N}\sum_{h=1}^{H}\mathbb{E}\left[\mathbb{I}\{A_{n,h}^{\mathsf{c}}\}\mathbb{I}\{(\cap_{i=1}^{n-1}\cap_{j=1}^{H}A_{i,j})\cap(\cap_{j=1}^{h-1}A_{n,j})\}\right] \\
&\le \sum_{n=1}^{N}\sum_{h=1}^{H}\mathbb{E}\left[\mathbb{I}\{A_{n,h}^{\mathsf{c}}\}\mathbb{I}\{A_{n,h-1}\}\right].
\end{aligned}$$

Here, we use the convention that $\mathbb{I}\{A_{n,0}\} := 1$. For any $n \in [N]$ and $h = 1$,

$$\mathbb{E}\left[\mathbb{I}\{A_{n,1}^{\mathsf{c}}\}\mathbb{I}\{A_{n,0}\}\right] = \mathbb{P}\left(\|\boldsymbol{s}_{n,1}\|_2 > R_1\right) = \mathbb{P}\left(\|\boldsymbol{\varepsilon}_{n,1}\|_2 > R_1\right).$$

By Lemma E.3, we have

$$\mathbb{P}\left(\|\boldsymbol{\varepsilon}_{n,1}\|_2 > R_1\right) \le \exp\left(-\frac{R_1^2}{8\sigma^2}\right).$$

We turn our attention to the case where $h > 1$. Since $\|\boldsymbol{x}_{n,h}\|_2^2 = \|\boldsymbol{s}_{n,h}\|_2^2 + \|\boldsymbol{a}_{n,h}\|_2^2$, we have $\|\boldsymbol{x}_{n,h}\|_2 \le \widetilde{R}_h$ whenever $\|\boldsymbol{s}_{n,h}\|_2 \le R_h$. For any $n \in [N]$ and $h > 1$,

$$\begin{aligned}
\mathbb{I}\{A_{n,h}^{\mathsf{c}}\}\mathbb{I}\{A_{n,h-1}\} &= \mathbb{I}\{\|f^{\star}(\boldsymbol{s}_{n,h-1}, \boldsymbol{a}_{n,h-1}) + \boldsymbol{\varepsilon}_{n,h}\|_2 > R_h\}\mathbb{I}\{\|\boldsymbol{s}_{n,h-1}\|_2 \le R_{h-1}\} \tag{18} \\
&\le \mathbb{I}\left\{\sup_{\boldsymbol{x}\in\mathbb{B}_2^{d_s+d_a}(\widetilde{R}_{h-1})}\|f^{\star}(\boldsymbol{x})\|_2 + \|\boldsymbol{\varepsilon}_{n,h}\|_2 > R_h\right\}.
\end{aligned}$$

Using Lemma E.2, we obtain

$$\mathbb{I}\left\{\sup_{\boldsymbol{x}\in\mathbb{B}_2^{d_s+d_a}(\widetilde{R}_{h-1})}\|f^{\star}(\boldsymbol{x})\|_2 + \|\boldsymbol{\varepsilon}_{n,h}\|_2 > R_h\right\} \le \mathbb{I}\left\{\sup_{\boldsymbol{x}\in\mathbb{B}_2^{d_s+d_a}(\widetilde{R}_{h-1})}\|f^{\star}(\boldsymbol{x})\|_2 > \frac{R_h}{2}\right\} + \mathbb{I}\left\{\|\boldsymbol{\varepsilon}_{n,h}\|_2 > \frac{R_h}{2}\right\}.$$

Thus for any $n \in [N]$ and $h > 1$,

$$\mathbb{E}[\mathbb{I}\{A_{n,h}^{\mathsf{c}}\}\mathbb{I}\{A_{n,h-1}\}] \le \mathbb{P}\left(\sup_{\boldsymbol{x}\in\mathbb{B}_2^{d_s+d_a}(\widetilde{R}_{h-1})}\|f^{\star}(\boldsymbol{x})\|_2 > \frac{R_h}{2}\right) + \mathbb{P}\left(\|\boldsymbol{\varepsilon}_{n,h}\|_2 > \frac{R_h}{2}\right).$$

Following the same reasoning as in the $h = 1$ case, we have

$$\mathbb{P}\left( \|\varepsilon_{n,h}\|_2 > \frac{R_h}{2} \right) \leq \exp\left( -\frac{R_h^2}{32\sigma^2} \right).$$

Since $R_h$ satisfies the inequality in (17), by Lemma 4.6,

$$\mathbb{P}\left( \sup_{\boldsymbol{x} \in \mathbb{B}_2^{d_s + d_a}(\widetilde{R}_{h-1})} \|f^\star(\boldsymbol{x})\|_2 > \frac{R_h}{2} \right) \leq \exp\left( -\frac{R_h^2}{32C} \right).$$

This means that

$$\mathbb{E}[\mathbb{I}\{A_{n,h}^{\mathsf{c}}\}\mathbb{I}\{A_{n,h-1}\}] \leq \exp\left( -\frac{R_h^2}{32C} \right) + \exp\left( -\frac{R_h^2}{32\sigma^2} \right).$$

Combining everything so far results in the statement that we wanted. $\qquad\square$

Next, we determine how large $R_1, \ldots, R_H$ need to be to ensure that $\mathbb{P}(A^{\mathsf{c}})$ is of the order $1/T$.

**Lemma E.5.** *If we set $R_1 = \max(2\sigma\sqrt{d_s}, \sqrt{16\sigma^2\log(T)})$, and for each $h \in \{2, \ldots, H\}$, we set $R_h = \max(Z_1, Z_2)$, where*

$$Z_1 = \sqrt{64\max(C, \sigma^2)\log(T)},$$
$$Z_2 = 168\alpha^{-1/2}\sqrt{\max(C, \sigma^2)(d_s + d_a)\log(5 + 5\widetilde{R}_{h-1}^\alpha L/C)},$$

*then $\mathbb{P}(A^{\mathsf{c}}) \leq 2/T$.*

*Proof.* The idea is to verify that these choices of $R_1, \ldots, R_H$ ensure that each term that appears in the upper bound in Lemma E.4 is of the order $1/T^2$. By rearranging the following inequality, we can ensure that

$$\exp\left( -\frac{R_1^2}{8\sigma^2} \right) \leq \frac{1}{T^2},$$

if

$$R_1 \geq \sqrt{16\sigma^2\log(T)}.$$

Suppose now that the value of $R_{h-1}$ has already been fixed. We require that $R_h$ is large enough to ensure that

$$\exp\left( -\frac{R_h^2}{32\sigma^2} \right) \leq \frac{1}{T^2}.$$

By rearranging this inequality, we see that it is satisfied whenever

$$R_h \geq \sqrt{64\sigma^2\log(T)}.$$

Next, we require that

$$\exp\left( -\frac{R_h^2}{32C} \right) \leq \frac{1}{T^2}.$$

By rearranging this inequality, we see that it is satisfied whenever

$$R_h \geq \sqrt{64C\log(T)}.$$

Finally, we also require that

$$R_h \geq 168\alpha^{-1/2}\sqrt{\max(C, \sigma^2)(d_s + d_a)\log(5 + 5\widetilde{R}_{h-1}^\alpha L/C)}.$$

By Lemma E.4, we have

$$\mathbb{P}(A^{\mathsf{c}}) \leq \frac{N}{T^2} + N\sum_{h=2}^{H}\frac{2}{T^2} = \frac{2NH}{T^2} = \frac{2}{T}.$$

This concludes the proof. $\qquad\square$

To derive explicit bounds on how large $R_1, \ldots, R_H$ are, we assume that $T$ is sufficiently large. The following lemma will help us to determine how large is sufficiently large.

**Lemma E.6.** *If, for some constants $D \geq 1$, $L \geq 1$, $R \geq 0$, we have $T \geq \max(D\sqrt{2\log(DL(R+1))}, 1)$, then $T$ satisfies*

$$T \geq D\sqrt{\log((T+R)L)}.$$

*Proof.* Since $D\sqrt{2\log(DL(R+1))} \geq 0$, we have

$$\frac{1}{2}T^2 \geq D^2\log(DL(R+1)).$$

We add $T^2/2$ to both sides to obtain

$$
\begin{aligned}
T^2 &\geq \frac{1}{2}T^2 + D^2\log(DL(R+1)) \\
&= \frac{D^2}{2}\frac{T^2}{D^2} + D^2\log(DL(R+1)) \\
&\geq \frac{D^2}{2}\log(T^2/D^2) + D^2\log(DL(R+1)) \\
&= D^2\log(TL(R+1)) \\
&\geq D^2\log((T+R)L).
\end{aligned}
$$

Since $D^2\log((T+R)L) \geq 0$, and $T \geq 0$, we can conclude that $T \geq D\sqrt{\log((T+R)L)}$, which is the inequality that we wanted to prove. $\qquad\square$

Finally, we derive some explicit bounds on how large $R_1, \ldots, R_H$ are if we set them according to Lemma E.5.

**Lemma E.7.** *Suppose that we set $R_1 = \max(2\sigma\sqrt{d_s}, \sqrt{16\sigma^2\log(T)})$, and for each $h \in \{2, \ldots, H\}$, we set $R_h = \max(Z_1, Z_2)$, where*

$$Z_1 = \sqrt{64\max(C, \sigma^2)\log(T)},$$

$$Z_2 = 168\alpha^{-1/2}\sqrt{\max(C, \sigma^2)(d_s + d_a)\log(5 + 5\widetilde{R}^\alpha_{h-1}L/C)}.$$

*Let $D = 168\alpha^{-1/2}\sqrt{\max(C, \sigma^2)(d_s + d_a)}$. If $T$ is a positive integer that satisfies*

$$T \geq D\sqrt{2\log(10D\max(1, L/C)(R_a + 1))}, \tag{19}$$

*then for all $h \in [H]$,*

$$R_h \leq 168\alpha^{-1/2}\sqrt{\max(C, \sigma^2)(d_s + d_a)\log(10(T + R_a)\max(1, L/C))}.$$

*Proof.* We prove by induction on $h$ that if $T$ satisfies (19), then for all $h \in [H]$,

$$R_h \leq 168\alpha^{-1/2}\sqrt{\max(C, \sigma^2)(d_s + d_a)\log(10(T + R_a)\max(1, L/C))}.$$

For the base case $h = 1$, we use some extremely decadent inequalities to upper bound $R_1$. First, we have

$$2\sigma\sqrt{d_s} \leq 2\alpha^{-1/2}\sqrt{\max(C, \sigma^2)(d_s + d_a)\log(10(T + R_a)\max(1, L/C))}$$

Next, we have

$$\sqrt{16\sigma^2\log(T)} \leq 4\alpha^{-1/2}\sqrt{\max(C, \sigma^2)(d_s + d_a)\log(10(T + R_a)\max(1, L/C))}.$$

Thus we have

$$R_1 = \max(2\sigma\sqrt{d_s}, \sqrt{16\sigma^2\log(T)}) \leq 168\alpha^{-1/2}\sqrt{\max(C, \sigma^2)(d_s + d_a)\log(10(T + R_a)\max(1, L/C))}.$$

Now suppose inductively that for some $h \geq 1$,

$$R_h \leq 168\alpha^{-1/2}\sqrt{\max(C,\sigma^2)(d_s + d_a)\log(10(T + R_a)\max(1, L/C))}\,.$$

We need to show that $R_{h+1}$ satisfies the same upper bound. There are two cases. In the first case,

$$R_{h+1} = \sqrt{64\max(C,\sigma^2)\log(T)} \leq 8\alpha^{-1/2}\sqrt{\max(C,\sigma^2)(d_s + d_a)\log(10(T + R_a)\max(1, L/C))}\,.$$

and the induction is closed. In the second case,

$$R_{h+1} = 168\alpha^{-1/2}\sqrt{\max(C,\sigma^2)(d_s + d_a)\log(5 + 5\widetilde{R}_h^{\alpha}L/C)}\,.$$

We bound the logarithmic term first. By Lemma E.6, since $T$ satisfies the lower bound in (19), $T$ also satsifies the inequality

$$T \geq 168\alpha^{-1/2}\sqrt{\max(C,\sigma^2)(d_s + d_a)\log(10(T + R_a)\max(1, L/C))}\,.$$

From our inductive hypothesis and this condition on $T$, it follows that

$$\widetilde{R}_h = \sqrt{R_h^2 + R_a^2} \leq 168\alpha^{-1/2}\sqrt{\max(C,\sigma^2)(d_s + d_a)\log(10(T + R_a)\max(1, L/C))} + R_a \leq T + R_a\,.$$

Therefore,

$$\log(5 + 5\widetilde{R}_h^{\alpha}L/C) \leq \log(5 + 5(T + R_a)^{\alpha}L/C) \leq \log(10(T + R_a)\max(1, L/C))\,.$$

With this, we have

$$R_{h+1} \leq 168\alpha^{-1/2}\sqrt{\max(C,\sigma^2)(d_s + d_a)\log(10(T + R_a)\max(1, L/C))}\,.$$

This closes the induction. □

### E.3. Proof of Lemma 4.7

With access to the results in Appendix E.2, it is straightforward to prove Lemma 4.7.

*Proof of Lemma 4.7.* Let us define

$$R := 168\alpha^{-1/2}\sqrt{\max(C,\sigma^2)(d_s + d_a)\log(10(T + R_a)\max(1, L/C))}\,.$$

If we set $R_1, \ldots, R_H$ according to Lemma E.5, then by Lemma E.5,

$$\mathbb{P}(A^c) \leq \frac{2}{T}\,.$$

By Lemma E.7, for all $h \in [H]$, $R_h \leq R$. Therefore,

$$\begin{aligned}
\mathbb{P}(A^c) &= \mathbb{P}(\cup_{n=1}^{N} \cup_{h=1}^{H} \{\|\boldsymbol{s}_{n,h}\|_2 > R_h\}) \\
&\geq \mathbb{P}(\cup_{n=1}^{N} \cup_{h=1}^{H} \{\|\boldsymbol{s}_{n,h}\|_2 > R\}) \\
&= \mathbb{P}\left(\sup_{n \in [N], h \in [H]} \|\boldsymbol{s}_{n,h}\|_2 > R\right)\,.
\end{aligned}$$

□

## F. Bounding the Regret by the Estimation Error

We prove a slight extension of a result from a Section 5.1 of Osband et al. (2013) which expresses the value estimation error as a sum of Bellman errors. We then give a shorter (but less general) proof of an inequality that is essentially the same as Lemma 1 in Fan & Ming (2021), and which allows us upper bound the Bellman errors by the model estimation errors. In Section F.1, we use these results to prove Lemma 4.9.

For brevity, we define $P_{n,h}^\star := P^\star(\boldsymbol{x}_{n,h})$ and $P_{n,h}^{(n)} := P^{(n)}(\boldsymbol{x}_{n,h})$. The following lemma allows us to upper bound the cumulative value estimation error by a sum of Bellman errors. The main difference between Lemma F.1 and the result from Section 5.1 of Osband et al. (2013) is that both the value estimation errors and the Bellman errors are multiplied by an indicator variable. Without this, we would have an identity instead of an inequality. Repeated application of the Bellman operator introduces a martingale difference sequence, which has expectation 0. However, the product of the indicator variable and this martingale difference sequence may not have expectation 0. Fortunately, one can still upper bound this expected value by a quantity that turns out to be negligible compared to the dominant term of the final regret bound.

**Lemma F.1.** *For any event A,*

$$\mathbb{E}\left[\mathbb{I}\{A\}\sum_{n=1}^N\left(V_{\pi_n,1}^{\mathcal{M}_n}(\boldsymbol{s}_{n,1}) - V_{\pi_n,1}^{\mathcal{M}_\star}(\boldsymbol{s}_{n,1})\right)\right] \le \mathbb{E}\left[\mathbb{I}\{A\}\sum_{n=1}^N\sum_{h=1}^{H-1}\langle P_{n,h}^{(n)} - P_{n,h}^\star, V_{\pi_n,h+1}^{\mathcal{M}_n}\rangle\right] + 2R_{\max}H\sqrt{2\pi T}\,.$$

*Proof.* First, we use the Bellman equation (cf. (1)) to write

$$
\begin{aligned}
V_{\pi_n,1}^{\mathcal{M}_n}(\boldsymbol{s}_{n,1}) - V_{\pi_n,1}^{\mathcal{M}_\star}(\boldsymbol{s}_{n,1}) &= \mathsf{T}_{\pi_n,1}^{\mathcal{M}_n}V_{\pi_n,2}^{\mathcal{M}_n}(\boldsymbol{s}_{n,1}) - \mathsf{T}_{\pi_n,1}^{\mathcal{M}_\star}V_{\pi_n,2}^{\mathcal{M}_\star}(\boldsymbol{s}_{n,1}) \\
&= \int_{\mathcal{A}}\left(\langle P^{(n)}(\boldsymbol{s}_{n,1},\boldsymbol{a}), V_{\pi_n,2}^{\mathcal{M}_n}\rangle - \langle P^\star(\boldsymbol{s}_{n,1},\boldsymbol{a}), V_{\pi_n,2}^{\mathcal{M}_\star}\rangle\right)\pi_n(\boldsymbol{a}|\boldsymbol{s}_{n,1},1)\mathrm{d}\boldsymbol{a} \\
&= \int_{\mathcal{A}}\langle P^{(n)}(\boldsymbol{s}_{n,1},\boldsymbol{a}) - P^\star(\boldsymbol{s}_{n,1},\boldsymbol{a}), V_{\pi_n,2}^{\mathcal{M}_n}\rangle\pi_n(\boldsymbol{a}|\boldsymbol{s}_{n,1},1)\mathrm{d}\boldsymbol{a} \\
&\quad + \int_{\mathcal{A}}\langle P^\star(\boldsymbol{s}_{n,1},\boldsymbol{a}), V_{\pi_n,2}^{\mathcal{M}_n} - V_{\pi_n,2}^{\mathcal{M}_\star}\rangle\pi_n(\boldsymbol{a}|\boldsymbol{s}_{n,1},1)\mathrm{d}\boldsymbol{a} \\
&= \int_{\mathcal{A}}\langle P^{(n)}(\boldsymbol{s}_{n,1},\boldsymbol{a}) - P^\star(\boldsymbol{s}_{n,1},\boldsymbol{a}), V_{\pi_n,2}^{\mathcal{M}_n}\rangle\pi_n(\boldsymbol{a}|\boldsymbol{s}_{n,1},1)\mathrm{d}\boldsymbol{a} \\
&\quad + V_{\pi_n,2}^{\mathcal{M}_n}(\boldsymbol{s}_{n,2}) - V_{\pi_n,2}^{\mathcal{M}_\star}(\boldsymbol{s}_{n,2}) + D_{n,1}\,,
\end{aligned}
$$

where $D_{n,h} := \int_{\mathcal{A}}\langle P^\star(\boldsymbol{s}_{n,h},\boldsymbol{a}), V_{\pi_n,h+1}^{\mathcal{M}_n} - V_{\pi_n,h+1}^{\mathcal{M}_\star}\rangle\pi_n(\boldsymbol{a}|\boldsymbol{s}_{n,h},h)\mathrm{d}\boldsymbol{a} - V_{\pi_n,h+1}^{\mathcal{M}_n}(\boldsymbol{s}_{n,h+1}) + V_{\pi_n,h+1}^{\mathcal{M}_\star}(\boldsymbol{s}_{n,h+1})$. From this identity, it follows that

$$
\begin{aligned}
\mathbb{E}\left[\mathbb{I}\{A\}\sum_{n=1}^N\left(V_{\pi_n,1}^{\mathcal{M}_n}(\boldsymbol{s}_{n,1}) - V_{\pi_n,1}^{\mathcal{M}_\star}(\boldsymbol{s}_{n,1})\right)\right] &= \mathbb{E}\left[\mathbb{I}\{A\}\sum_{n=1}^N\sum_{h=1}^{H-1}D_{n,h}\right] \\
&\quad + \mathbb{E}\left[\mathbb{I}\{A\}\sum_{n=1}^N\sum_{h=1}^{H-1}\int_{\mathcal{A}}\langle P^{(n)}(\boldsymbol{s}_{n,h},\boldsymbol{a}) - P^\star(\boldsymbol{s}_{n,h},\boldsymbol{a}), V_{\pi_n,h+1}^{\mathcal{M}_n}\rangle\pi_n(\boldsymbol{a}|\boldsymbol{s}_{n,h},h)\mathrm{d}\boldsymbol{a}\right]\,.
\end{aligned}
\tag{20}
$$

To upper bound the first term, we notice that $\sum_{n=1}^N\sum_{h=1}^{H-1}D_{n,h}$ is the sum of a martingale difference sequence with bounded increments. Let us define the random variable

$$Z_{n,h} := V_{\pi_n,h+1}^{\mathcal{M}_n}(\boldsymbol{s}_{n,h+1}) - V_{\pi_n,h+1}^{\mathcal{M}_\star}(\boldsymbol{s}_{n,h+1})\,,$$

and the $\sigma$-algebra

$$\mathcal{G}_{n,h-1} := \sigma(\boldsymbol{s}_{1,1}, \boldsymbol{a}_{1,1}, \ldots, \boldsymbol{s}_{n,h-1}, \boldsymbol{a}_{n,h-1}, \boldsymbol{s}_{n,h}, f^\star, f^{(1)}, \ldots, f^{(n)})\,.$$

We notice that

$$\mathbb{E}\left[\mathbb{I}\{A\}\sum_{n=1}^N\sum_{h=1}^{H-1}D_{n,h}\right] = \mathbb{E}\left[\mathbb{I}\{A\}\sum_{n=1}^N\sum_{h=1}^{H-1}\mathbb{E}[Z_{n,h}|\mathcal{G}_{n,h-1}] - Z_{n,h}\right]\,.$$

Moreover, given any $\mathcal{G}_{n,h-1}$, $Z_{n,h}$ is bounded between $-2R_{\max}H$ and $2R_{\max}H$, and is therefore conditionally $2R_{\max}H$-

sub-Gaussian. This means that, for any $x > 0$,

$$\mathbb{P}\left(\sum_{n=1}^{N}\sum_{h=1}^{H-1}\mathbb{E}[Z_{n,h}|\mathcal{G}_{n,h-1}] - Z_{n,h} \geq x\right) = \mathbb{P}\left(\exp\left(\lambda\sum_{n=1}^{N}\sum_{h=1}^{H-1}\mathbb{E}[Z_{n,h}|\mathcal{G}_{n,h-1}] - Z_{n,h}\right) \geq \exp(\lambda x)\right)$$

$$\leq \mathbb{E}\left[\exp\left(\lambda\sum_{n=1}^{N}\sum_{h=1}^{H-1}\mathbb{E}[Z_{n,h}|\mathcal{G}_{n,h-1}] - Z_{n,h}\right)\right]\exp(-\lambda x)$$

$$= \mathbb{E}\left[\prod_{n=1}^{N}\prod_{h=1}^{H-1}\mathbb{E}[\exp(\lambda(\mathbb{E}[Z_{n,h}|\mathcal{G}_{n,h-1}] - Z_{n,h}))|\mathcal{G}_{n,h-1}]\right]\exp(-\lambda x)$$

$$\leq \exp(2TR_{\max}^2 H^2\lambda^2 - \lambda x).$$

Choosing $\lambda = \frac{x}{4TR_{\max}^2 H^2}$, this becomes

$$\mathbb{P}\left(\sum_{n=1}^{N}\sum_{h=1}^{H-1}D_{n,h} \geq x\right) \leq \exp\left(-\frac{x^2}{8TR_{\max}^2 H^2}\right).$$

To save space, let us write $Y := \sum_{n=1}^{N}\sum_{h=1}^{H-1}D_{n,h}$.

$$\mathbb{E}[\mathbb{I}\{A\}Y] = \mathbb{E}[\mathbb{I}\{A\}Y\mathbb{I}\{Y \geq 0\}] + \mathbb{E}[\mathbb{I}\{A\}Y\mathbb{I}\{Y < 0\}]$$

$$\leq \mathbb{E}[\mathbb{I}\{A\}Y\mathbb{I}\{Y \geq 0\}]$$

$$\leq \int_0^\infty \mathbb{P}(Y\mathbb{I}\{Y \geq 0\} \geq x)\mathrm{d}x$$

$$= \int_0^\infty \mathbb{P}(Y \geq x)\mathrm{d}x$$

$$\leq \int_0^\infty \exp\left(-\frac{x^2}{8TR_{\max}^2 H^2}\right)\mathrm{d}x$$

$$= R_{\max}H\sqrt{2\pi T}.$$

In the second term on the right-hand side of (20), we would like to take the conditional expectations w.r.t. $\pi_n(\cdot|\boldsymbol{s}_{n,h}, h)$ outside the sum. It turns out that this can be done by introducing another martingale difference sequence. This time we define the random variable $Z_{n,h}$ by

$$Z_{n,h} = \langle P_{n,h}^{(n)} - P_{n,h}^\star, V_{\pi_n,h+1}^{\mathcal{M}_n}\rangle.$$

With the same $\sigma$-algebra $\mathcal{G}_{n,h-1}$, we notice that

$$\mathbb{E}\left[\mathbb{I}\{A\}\sum_{n=1}^{N}\sum_{h=1}^{H-1}\int_{\mathcal{A}}\langle P^{(n)}(\boldsymbol{s}_{n,h}, \boldsymbol{a}) - P^\star(\boldsymbol{s}_{n,h}, \boldsymbol{a}), V_{\pi_n,h+1}^{\mathcal{M}_n}\rangle\pi_n(\boldsymbol{a}|\boldsymbol{s}_{n,h}, h)\mathrm{d}\boldsymbol{a}\right] = \mathbb{E}\left[\mathbb{I}\{A\}\sum_{n=1}^{N}\sum_{h=1}^{H-1}\mathbb{E}[Z_{n,h}|\mathcal{G}_{n,h-1}]\right].$$

Note also that given any $\mathcal{G}_{n,h-1}$, $Z_{n,h}$ is bounded between $-2R_{\max}H$ and $2R_{\max}H$. Therefore, using the same argument as before, we have

$$\mathbb{E}\left[\mathbb{I}\{A\}\sum_{n=1}^{N}\sum_{h=1}^{H-1}\mathbb{E}[Z_{n,h}|\mathcal{G}_{n,h-1}]\right] = \mathbb{E}\left[\mathbb{I}\{A\}\sum_{n=1}^{N}\sum_{h=1}^{H-1}Z_{n,h}\right] + \mathbb{E}\left[\mathbb{I}\{A\}\sum_{n=1}^{N}\sum_{h=1}^{H-1}\mathbb{E}[Z_{n,h}|\mathcal{G}_{n,h-1}] - Z_{n,h}\right]$$

$$\leq \mathbb{E}\left[\mathbb{I}\{A\}\sum_{n=1}^{N}\sum_{h=1}^{H-1}Z_{n,h}\right] + R_{\max}H\sqrt{2\pi T}.$$

This concludes the proof. $\qquad\square$

The following lemma is essentially the same as Lemma 1 in Fan & Ming (2021), which is referred to as "Lemma on the property derived from noises with a symmetric probability distribution". Instead, we prefer to use Pinsker's inequality to prove this result.

**Lemma F.2.** *For any $n \in [N]$ and $h \in [H-1]$, with probability 1,*

$$\langle P_{n,h}^{(n)} - P_{n,h}^{\star}, V_{\pi_n,h+1}^{\mathcal{M}_n} \rangle \leq \frac{HR_{\max}}{\sigma} \|f^{(n)}(\boldsymbol{x}_{n,h}) - f^{\star}(\boldsymbol{x}_{n,h})\|_2 .$$

*Proof.* Using Hölder's inequality and then Pinsker's inequality, we obtain

$$
\begin{aligned}
\langle P_{n,h}^{(n)} - P_{n,h}^{\star}, V_{\pi_n,h+1}^{\mathcal{M}_n} \rangle &= \int_{\mathcal{S}} V_{\pi_n,h+1}^{\mathcal{M}_n}(\boldsymbol{s})(P_{n,h}^{(n)}(\boldsymbol{s}) - P_{n,h}^{\star}(\boldsymbol{s}))\mathrm{d}\boldsymbol{s} \\
&\leq \|V_{\pi_n,h+1}^{\mathcal{M}_n}\|_{\infty} \int |P_{n,h}^{(n)}(\boldsymbol{s}) - P_{n,h}^{\star}(\boldsymbol{s})|\mathrm{d}\boldsymbol{s} \\
&= 2\|V_{\pi_n,h+1}^{\mathcal{M}_n}\|_{\infty} D_{\mathrm{TV}}(P_{n,h}^{(n)}\|P_{n,h}^{\star}) \\
&\leq 2HR_{\max}\sqrt{\tfrac{1}{2}D_{\mathrm{KL}}(P_{n,h}^{(n)}\|P_{n,h}^{\star})} \\
&= \frac{HR_{\max}}{\sigma}\|f^{(n)}(\boldsymbol{x}_{n,h}) - f^{\star}(\boldsymbol{x}_{n,h})\|_2 .
\end{aligned}
$$

This concludes the proof. $\qquad\square$

### F.1. Proof of Lemma 4.9

Lemma 4.9 is a straightforward consequence of Lemma F.1 and F.2.

*Proof of Lemma 4.9.* By Lemma F.1,

$$\mathbb{E}\left[\mathbb{I}\{A\}\sum_{n=1}^{N}\left(V_{\pi_n,1}^{\mathcal{M}_n}(\boldsymbol{s}_{n,1}) - V_{\pi_n,1}^{\mathcal{M}_{\star}}(\boldsymbol{s}_{n,1})\right)\right] \leq \mathbb{E}\left[\mathbb{I}\{A\}\sum_{n=1}^{N}\sum_{h=1}^{H-1}\langle P_{n,h}^{(n)} - P_{n,h}^{\star}, V_{\pi_n,h+1}^{\mathcal{M}_n}\rangle\right] + 2R_{\max}H\sqrt{2\pi T} .$$

By Lemma F.2,

$$\mathbb{E}\left[\mathbb{I}\{A\}\sum_{n=1}^{N}\sum_{h=1}^{H-1}\langle P_{n,h}^{(n)} - P_{n,h}^{\star}, V_{\pi_n,h+1}^{\mathcal{M}_n}\rangle\right] \leq \frac{HR_{\max}}{\sigma}\mathbb{E}\left[\mathbb{I}\{A\}\sum_{n=1}^{N}\sum_{h=1}^{H-1}\|f^{(n)}(\boldsymbol{x}_{n,h}) - f^{\star}(\boldsymbol{x}_{n,h})\|_2\right] .$$

This concludes the proof. $\qquad\square$

## G. Upper Bounds for the Cumulative Estimation Error

We prove upper bounds on the cumulative estimation error, which is

$$\mathbb{E}\left[\mathbb{I}\{A\}\sum_{n=1}^{N}\sum_{h=1}^{H-1}\|f^{(n)}(\boldsymbol{x}_{n,h}) - f^{\star}(\boldsymbol{x}_{n,h})\|_2\right] ,$$

where $A := \{\sup_{n \in [N], h \in [H]} \|\boldsymbol{s}_{n,h}\|_2 \leq R\}$ for some $R > 0$. The challenge is to exploit the fact $f^{(n)}$ concentrates around $f^{\star}$, without having to work with the posterior covariance kernel. The general idea is to separate the estimation error into a discretized estimation error and two discretization error terms via a single-step discretization. Let $\widetilde{R} = \sqrt{R^2 + R_a^2}$. For some $\varepsilon \in (0, (7/(e\sqrt{8}))^{\alpha/2}\widetilde{R}]$, let $B_\varepsilon$ be a minimal $\varepsilon$-cover of $\mathbb{B}^{d_s + d_a}(\widetilde{R})$ w.r.t. the Euclidean metric $d_2$ and let $\omega : \mathbb{B}^{d_s + d_a}(\widetilde{R}) \to B_\varepsilon$ defined by $\omega(\boldsymbol{x}) := \arg\min_{\boldsymbol{y} \in B_\varepsilon} d_2(\boldsymbol{x}, \boldsymbol{y})$ be a function that maps each $\boldsymbol{x} \in \mathbb{B}^{d_s + d_a}(\widetilde{R})$ to the closest point in $B_\varepsilon$ (with ties broken arbitrarily). Using the triangle inequality, we obtain

$$
\begin{aligned}
\mathbb{E}\left[\mathbb{I}\{A\}\sum_{n=1}^{N}\sum_{h=1}^{H-1}\|f^{(n)}(\boldsymbol{x}_{n,h}) - f^{\star}(\boldsymbol{x}_{n,h})\|_2\right] &\leq \mathbb{E}\left[\mathbb{I}\{A\}\sum_{n=1}^{N}\sum_{h=1}^{H-1}\|f^{(n)}(\omega(\boldsymbol{x}_{n,h})) - f^{\star}(\omega(\boldsymbol{x}_{n,h}))\|_2\right] \quad (21) \\
&+ \mathbb{E}\left[\mathbb{I}\{A\}\sum_{n=1}^{N}\sum_{h=1}^{H-1}\|f^{(n)}(\boldsymbol{x}_{n,h}) - f^{(n)}(\omega(\boldsymbol{x}_{n,h}))\|_2\right] \\
&+ \mathbb{E}\left[\mathbb{I}\{A\}\sum_{n=1}^{N}\sum_{h=1}^{H-1}\|f^{\star}(\boldsymbol{x}_{n,h}) - f^{\star}(\omega(\boldsymbol{x}_{n,h}))\|_2\right] .
\end{aligned}
$$

We call the first term on the right-hand side the discretized estimation error. The remaining two terms are the discretization errors. In Appendix G.1, we use an elliptical potential lemma and an exponential moment inequality for chi-squared random variables to upper bound the discretized estimation error. In Appendix G.2, we the chaining argument in Lemma 4.5 to upper bound the discretization errors. Finally, in Appendix G.3, we prove Lemma 4.10.

### G.1. Bounding the Discretized Estimation Error

We upper bound the discretized estimation error. First, we show that the supremum of the normalized estimation error is conditionally sub-Gaussian.

**Lemma G.1.** *The random variable* $\sup_{\boldsymbol{x} \in B_\varepsilon} \frac{\|f^{(n)}(\boldsymbol{x}) - f^\star(\boldsymbol{x})\|_2}{\sigma_{n-1}(\boldsymbol{x})}$ *is conditionally (on* $\mathcal{F}_{n-1}$*)* $\sqrt{2}$*-sub-Gaussian.*

*Proof.* Since $B_\varepsilon$ is finite, we can write it as $B_\varepsilon = \{\boldsymbol{x}_1, \ldots, \boldsymbol{x}_m\}$. For any $i \in [d_s]$ and $j \in [m]$, the condtional distribution of $f_i^{(n)}(\boldsymbol{x}_j) - f_i^\star(\boldsymbol{x}_j)$ is Gaussian with mean 0 and variance $2\sigma_{n-1}^2(\boldsymbol{x}_j)$. Therefore, $(f_i^{(n)}(\boldsymbol{x}_j) - f_i^\star(\boldsymbol{x}_j))/\sigma_{n-1}(\boldsymbol{x}_j)$ is a (conditionally) Gaussian random variable with mean 0 and variance 2. Since the noise variables added to each dimension of the states are independent, and the components $f_1, \ldots, f_{d_s}$ are independent under the prior, the random variables $(f_i^{(n)}(\boldsymbol{x}_j) - f_i^\star(\boldsymbol{x}_j))/\sigma_{n-1}(\boldsymbol{x}_j)$ for $i = 1, \ldots, d_s$ are conditionally mutually independent. In addition, one can check that the joint distribution of the random variables

$$\frac{f_i^{(n)}(\boldsymbol{x}_1) - f_i^\star(\boldsymbol{x}_1)}{\sigma_{n-1}(\boldsymbol{x}_1)}, \ldots, \frac{f_i^{(n)}(\boldsymbol{x}_m) - f_i^\star(\boldsymbol{x}_m)}{\sigma_{n-1}(\boldsymbol{x}_m)}$$

does not depend on the choice of $i \in [d_s]$. Let $\boldsymbol{\Sigma} \in \mathbb{R}^{m \times m}$ be the covariance matrix with $(j, k)^{\text{th}}$ element $\Sigma_{j,k}$ given by

$$\Sigma_{j,k} := \mathbb{E}\left[\frac{f_1^{(n)}(\boldsymbol{x}_j) - f_1^\star(\boldsymbol{x}_j)}{\sigma_{n-1}(\boldsymbol{x}_j)} \frac{f_1^{(n)}(\boldsymbol{x}_k) - f_1^\star(\boldsymbol{x}_k)}{\sigma_{n-1}(\boldsymbol{x}_k)} \,\Big|\, \mathcal{F}_{n-1}\right].$$

We have already verified that for every $j \in [m]$, $\Sigma_{j,j} = 2$. Summarising everything so far, conditioned on $\mathcal{F}_{n-1}$, we have

$$\begin{bmatrix} (f_1^{(n)}(\boldsymbol{x}_1) - f_1^\star(\boldsymbol{x}_1))/\sigma_{n-1}(\boldsymbol{x}_1) \\ \vdots \\ (f_1^{(n)}(\boldsymbol{x}_m) - f_1^\star(\boldsymbol{x}_m))/\sigma_{n-1}(\boldsymbol{x}_m) \\ \vdots \\ (f_{d_s}^{(n)}(\boldsymbol{x}_1) - f_{d_s}^\star(\boldsymbol{x}_1))/\sigma_{n-1}(\boldsymbol{x}_1) \\ \vdots \\ (f_{d_s}^{(n)}(\boldsymbol{x}_m) - f_{d_s}^\star(\boldsymbol{x}_m))/\sigma_{n-1}(\boldsymbol{x}_m) \end{bmatrix} \sim \mathcal{N}\left(\begin{bmatrix} 0 \\ \vdots \\ 0 \\ \vdots \\ 0 \\ \vdots \\ 0 \end{bmatrix}, \begin{bmatrix} \Sigma_{1,1} & \cdots & \Sigma_{1,m} & \cdots & 0 & \cdots & 0 \\ \vdots & \ddots & \vdots & & \vdots & \ddots & \vdots \\ \Sigma_{m,1} & \cdots & \Sigma_{m,m} & \cdots & 0 & \cdots & 0 \\ \vdots & & \vdots & \ddots & \vdots & & \vdots \\ 0 & \cdots & 0 & \cdots & \Sigma_{1,1} & \cdots & \Sigma_{1,m} \\ \vdots & \ddots & \vdots & & \vdots & \ddots & \vdots \\ 0 & \cdots & 0 & \cdots & \Sigma_{m,1} & \cdots & \Sigma_{m,m} \end{bmatrix}\right).$$

From here, we can follow the steps taken in the proof of Lemma 4.4. In particular, we define the function $g : \mathbb{R}^{md_s} \to \mathbb{R}$ by

$$g(z_{1,1}, \ldots, z_{d_s,m}) := \max_{j \in [m]} \left(\sum_{i=1}^{d_s} \sum_{k=1}^m \Sigma_{j,k}^{1/2} z_{i,k}\right)^{1/2},$$

where $\Sigma_{j,k}^{1/2}$ is the $(j, k)^{\text{th}}$ entry of $\boldsymbol{\Sigma}^{1/2}$, and $\boldsymbol{\Sigma}^{1/2}$ is a symmetric square root of $\boldsymbol{\Sigma}$. Since $\max_{j \in [m]} \sqrt{\Sigma_{j,j}} = \sqrt{2}$, the same proof as before can be used to show that $g$ is $\sqrt{2}$-Lipschitz with respect to the Euclidean metric. Let $Z_{1,1}, \ldots, Z_{d_s,m}$ be i.i.d. standard Gaussian random variables. Conditioned on $\mathcal{F}_{n-1}$, we have

$$\sup_{\boldsymbol{x} \in B_\varepsilon} \frac{\|f^{(n)}(\boldsymbol{x}) - f^\star(\boldsymbol{x})\|_2}{\sigma_{n-1}(\boldsymbol{x})} \stackrel{\text{d}}{=} g(Z_{1,1}, \ldots, Z_{d_s,m}).$$

By Lemma 4.3, we can conclude that $\sup_{\boldsymbol{x} \in B_\varepsilon} \frac{\|f^{(n)}(\boldsymbol{x}) - f^\star(\boldsymbol{x})\|_2}{\sigma_{n-1}(\boldsymbol{x})}$ is conditionally $\sqrt{2}$-sub-Gaussian. $\square$

Using the previous lemma, we prove a tail bound for the supremum of the normalized estimation error.

**Lemma G.2.** *For any $\varepsilon \leq \widetilde{R}$ and $u \geq (32(d_s + d_a) \log(5\widetilde{R}/\varepsilon))^{1/2}$,*

$$\mathbb{P}\left(\sup_{\boldsymbol{x} \in B_\varepsilon} \frac{\|f^{(n)}(\boldsymbol{x}) - f^\star(\boldsymbol{x})\|_2}{\sigma_{n-1}(\boldsymbol{x})} \geq u \,\Big|\, \mathcal{F}_{n-1}\right) \leq \exp(-u^2/16)\,.$$

*Proof.* By Lemma G.1, $\sup_{\boldsymbol{x} \in B_\varepsilon} \|f^{(n)}(\boldsymbol{x}) - f^\star(\boldsymbol{x})\|_2/\sigma_{n-1}(\boldsymbol{x})$ is conditionally $\sqrt{2}$-sub-Gaussian. Therefore, for every $u \geq 0$,

$$\mathbb{P}\left(\sup_{\boldsymbol{x} \in B_\varepsilon} \frac{\|f^{(n)}(\boldsymbol{x}) - f^\star(\boldsymbol{x})\|_2}{\sigma_{n-1}(\boldsymbol{x})} - \mathbb{E}\left[\sup_{\boldsymbol{x} \in B_\varepsilon} \frac{\|f^{(n)}(\boldsymbol{x}) - f^\star(\boldsymbol{x})\|_2}{\sigma_{n-1}(\boldsymbol{x})} \,\Big|\, \mathcal{F}_{n-1}\right] \geq u \,\Big|\, \mathcal{F}_{n-1}\right) \leq \exp(-u^2/4)\,.$$

From here, we take the same approach used in the proof of Lemma 4.6. In particular, we upper bound the conditional expectation of the supremum and then show that when $u$ is large enough relative to this expected supremum, we get the tail bound that we wanted.

Following the reasoning in the proof of Lemma G.1, we see that the conditional distribution of the random vector $\frac{f^{(n)}(\boldsymbol{x}) - f^\star(\boldsymbol{x})}{\sigma_{n-1}(\boldsymbol{x})}$ is an isotropic Gaussian with mean $0$ and covariance $2\boldsymbol{I}$. Since the Euclidean norm is 1-Lipschitz with respect to the Euclidean metric, Lemma 4.3 tells us that $\frac{\|f^{(n)}(\boldsymbol{x}) - f^\star(\boldsymbol{x})\|_2}{\sigma_{n-1}(\boldsymbol{x})}$ is conditionally $\sqrt{2}$-sub-Gaussian. Thus we need to find the expected supremum of a finite collection of sub-Gaussian random variables. First, we use the inequality

$$\mathbb{E}\left[\sup_{\boldsymbol{x} \in B_\varepsilon} \frac{\|f^{(n)}(\boldsymbol{x}) - f^\star(\boldsymbol{x})\|_2}{\sigma_{n-1}(\boldsymbol{x})} \,\Big|\, \mathcal{F}_{n-1}\right] \leq \mathbb{E}\left[\sup_{\boldsymbol{x} \in B_\varepsilon} \left\{\frac{\|f^{(n)}(\boldsymbol{x}) - f^\star(\boldsymbol{x})\|_2}{\sigma_{n-1}(\boldsymbol{x})} - \mathbb{E}\left[\frac{\|f^{(n)}(\boldsymbol{x}) - f^\star(\boldsymbol{x})\|_2}{\sigma_{n-1}(\boldsymbol{x})} \,\Big|\, \mathcal{F}_{n-1}\right]\right\} \,\Big|\, \mathcal{F}_{n-1}\right]$$
$$+ \sup_{\boldsymbol{x} \in B_\varepsilon} \mathbb{E}\left[\frac{\|f^{(n)}(\boldsymbol{x}) - f^\star(\boldsymbol{x})\|_2}{\sigma_{n-1}(\boldsymbol{x})} \,\Big|\, \mathcal{F}_{n-1}\right]\,.$$

By the maximal inequality in Lemma 5.1 of van Handel (2016), the first term satisfies the upper bound

$$\mathbb{E}\left[\sup_{\boldsymbol{x} \in B_\varepsilon} \left\{\frac{\|f^{(n)}(\boldsymbol{x}) - f^\star(\boldsymbol{x})\|_2}{\sigma_{n-1}(\boldsymbol{x})} - \mathbb{E}\left[\frac{\|f^{(n)}(\boldsymbol{x}) - f^\star(\boldsymbol{x})\|_2}{\sigma_{n-1}(\boldsymbol{x})} \,\Big|\, \mathcal{F}_{n-1}\right]\right\} \,\Big|\, \mathcal{F}_{n-1}\right] \leq 2\sqrt{\log|B_\varepsilon|}\,.$$

By Jensen's inequality, the second term satisfies the upper bound

$$\sup_{\boldsymbol{x} \in B_\varepsilon} \mathbb{E}\left[\frac{\|f^{(n)}(\boldsymbol{x}) - f^\star(\boldsymbol{x})\|_2}{\sigma_{n-1}(\boldsymbol{x})} \,\Big|\, \mathcal{F}_{n-1}\right] \leq \sup_{\boldsymbol{x} \in B_\varepsilon} \left(\mathbb{E}\left[\frac{\|f^{(n)}(\boldsymbol{x}) - f^\star(\boldsymbol{x})\|_2^2}{\sigma_{n-1}^2(\boldsymbol{x})} \,\Big|\, \mathcal{F}_{n-1}\right]\right)^{1/2} = \sqrt{2d_s}\,.$$

Since $\varepsilon \leq \widetilde{R}$, Lemma D.1 tells us that
$$\log|B_\varepsilon| \leq (d_s + d_a) \log(3\widetilde{R}/\varepsilon)\,.$$

Using the inequality $\sqrt{a} + \sqrt{b} \leq \sqrt{2(a+b)}$, we obtain

$$\mathbb{E}\left[\sup_{\boldsymbol{x} \in B_\varepsilon} \frac{\|f^{(n)}(\boldsymbol{x}) - f^\star(\boldsymbol{x})\|_2}{\sigma_{n-1}(\boldsymbol{x})} \,\Big|\, \mathcal{F}_{n-1}\right] \leq 2\sqrt{(d_s + d_a)\log(3\widetilde{R}/\varepsilon)} + \sqrt{2d_s}$$
$$\leq \sqrt{8(d_s + d_a)\log(3\widetilde{R}/\varepsilon) + 4d_s}$$
$$\leq \sqrt{8(d_s + d_a)\log(3\sqrt{e}\widetilde{R}/\varepsilon)}\,.$$

Since $3\sqrt{e} \leq 5$, we can replace this with the slightly prettier upper bound

$$\mathbb{E}\left[\sup_{\boldsymbol{x} \in B_\varepsilon} \frac{\|f^{(n)}(\boldsymbol{x}) - f^\star(\boldsymbol{x})\|_2}{\sigma_{n-1}(\boldsymbol{x})} \,\Big|\, \mathcal{F}_{n-1}\right] \leq \sqrt{8(d_s + d_a)\log(5\widetilde{R}/\varepsilon)}\,.$$

Let us define $H$ to be the RHS of this inequality. For any $v \geq 0$, we know that

$$\mathbb{P}\left(\sup_{\boldsymbol{x} \in B_\varepsilon} \frac{\|f^{(n)}(\boldsymbol{x}) - f^\star(\boldsymbol{x})\|_2}{\sigma_{n-1}(\boldsymbol{x})} \geq v + H \,\Big|\, \mathcal{F}_{n-1}\right) \leq \exp(-u^2/4)\,.$$

Letting $u = v + H$, we have that for any $u \geq H$,

$$\mathbb{P}\left(\sup_{\boldsymbol{x} \in B_\varepsilon} \frac{\|f^{(n)}(\boldsymbol{x}) - f^\star(\boldsymbol{x})\|_2}{\sigma_{n-1}(\boldsymbol{x})} \geq u \,\Big|\, \mathcal{F}_{n-1}\right) \leq \exp(-(u - H)^2/4) \,.$$

Whenever $u \geq 2H$, we have $u - H \geq u/2$, which means $\exp(-(u-H)^2/4) \leq \exp(-u^2/16)$. $\qquad \square$

We will use the fact that whenever the kernel satisfies Assumption 3.3 and Assumption 3.4, the posterior standard deviation is Hölder continuinuous.

**Lemma G.3.** *Suppose that Assumption 3.3 and Assumption 3.4 are satisfied. Then for any $n \geq 0$,*

$$|\sigma_n(\boldsymbol{x}) - \sigma_n(\boldsymbol{y})| \leq \sqrt{2L}\|\boldsymbol{x} - \boldsymbol{y}\|_2^{\alpha/2} \,.$$

*Proof.* We begin by re-writing $\sigma_n^2(\boldsymbol{x})$. Let $\mathcal{H}$ be the reproducing kernel Hilbert space associated with the kernel $c$. For any $\boldsymbol{x}$, let us define $c_{\boldsymbol{x}} := c(\cdot, \boldsymbol{x}) \in \mathcal{H}$. We define the linear operator $\boldsymbol{\Phi}_n : \mathcal{H} \to \mathbb{R}^{n(H-1)}$ by

$$\boldsymbol{\Phi}_n h := [\langle c_{\boldsymbol{x}_{1,1}}, h \rangle_\mathcal{H}, \ldots, \langle c_{\boldsymbol{x}_{n,H-1}}, h \rangle_\mathcal{H}]^\top = [h(\boldsymbol{x}_{1,1}), \ldots, h(\boldsymbol{x}_{n,H-1})]^\top \,.$$

We can re-write $\sigma_n^2(\boldsymbol{x})$ as the quadratic form

$$\sigma_n^2(\boldsymbol{x}) = c(\boldsymbol{x}, \boldsymbol{x}) - \boldsymbol{c}_n^\top (\boldsymbol{C}_n + \sigma^2 \boldsymbol{I})^{-1} \boldsymbol{c}_n(\boldsymbol{x}) = \langle c_{\boldsymbol{x}}, (\boldsymbol{I}_\mathcal{H} - \boldsymbol{\Phi}_n^*(\boldsymbol{\Phi}_n \boldsymbol{\Phi}_n^* + \sigma^2 \boldsymbol{I})^{-1} \boldsymbol{\Phi}_n)c_{\boldsymbol{x}} \rangle_\mathcal{H} \,,$$

where $\boldsymbol{I}_\mathcal{H}$ is the identity function on $\mathcal{H}$. By the matrix inversion lemma, we have

$$\boldsymbol{\Phi}_n^*(\boldsymbol{\Phi}_n \boldsymbol{\Phi}_n^* + \sigma^2 \boldsymbol{I})^{-1} \boldsymbol{\Phi}_n = \boldsymbol{\Phi}_n^* \boldsymbol{\Phi}_n (\boldsymbol{\Phi}_n^* \boldsymbol{\Phi}_n + \sigma^2 \boldsymbol{I}_\mathcal{H})^{-1} \,.$$

Since $\boldsymbol{\Phi}_n^* \boldsymbol{\Phi}_n$ is positive semi-definite, we have $\|\boldsymbol{\Phi}_n^*(\boldsymbol{\Phi}_n \boldsymbol{\Phi}_n^* + \sigma^2 \boldsymbol{I})^{-1} \boldsymbol{\Phi}_n\|_{\mathrm{op}} \leq 1$. Since $\boldsymbol{\Phi}_n^* \boldsymbol{\Phi}_n (\boldsymbol{\Phi}_n^* \boldsymbol{\Phi}_n + \sigma^2 \boldsymbol{I}_\mathcal{H})^{-1}$ is also positive semi-definite, we also have $\|\boldsymbol{I} - \boldsymbol{\Phi}_n^*(\boldsymbol{\Phi}_n \boldsymbol{\Phi}_n^* + \sigma^2 \boldsymbol{I})^{-1} \boldsymbol{\Phi}_n\|_{\mathrm{op}} \leq 1$. For any $\boldsymbol{x}$ and $\boldsymbol{y}$, by the reverse triangle inequality and then Cauchy-Schwarz,

$$
\begin{aligned}
|\sigma_n(\boldsymbol{x}) - \sigma_n(\boldsymbol{y})| &= |\sqrt{\langle c_{\boldsymbol{x}}, (\boldsymbol{I}_\mathcal{H} - \boldsymbol{\Phi}_n^*(\boldsymbol{\Phi}_n \boldsymbol{\Phi}_n^* + \sigma^2 \boldsymbol{I})^{-1} \boldsymbol{\Phi}_n)c_{\boldsymbol{x}} \rangle_\mathcal{H}} - \sqrt{\langle c_{\boldsymbol{y}}, (\boldsymbol{I}_\mathcal{H} - \boldsymbol{\Phi}_n^*(\boldsymbol{\Phi}_n \boldsymbol{\Phi}_n^* + \sigma^2 \boldsymbol{I})^{-1} \boldsymbol{\Phi}_n)c_{\boldsymbol{y}} \rangle_\mathcal{H}}| \\
&\leq \sqrt{\langle c_{\boldsymbol{x}} - c_{\boldsymbol{y}}, (\boldsymbol{I}_\mathcal{H} - \boldsymbol{\Phi}_n^*(\boldsymbol{\Phi}_n \boldsymbol{\Phi}_n^* + \sigma^2 \boldsymbol{I})^{-1} \boldsymbol{\Phi}_n)(c_{\boldsymbol{x}} - c_{\boldsymbol{y}}) \rangle_\mathcal{H}} \\
&\leq \sqrt{\|c_{\boldsymbol{x}} - c_{\boldsymbol{y}}\|_\mathcal{H} \|(\boldsymbol{I}_\mathcal{H} - \boldsymbol{\Phi}_n^*(\boldsymbol{\Phi}_n \boldsymbol{\Phi}_n^* + \sigma^2 \boldsymbol{I})^{-1} \boldsymbol{\Phi}_n)(c_{\boldsymbol{x}} - c_{\boldsymbol{y}})\|_\mathcal{H}} \\
&\leq \|c_{\boldsymbol{x}} - c_{\boldsymbol{y}}\|_\mathcal{H} \\
&\leq \sqrt{2L\|\boldsymbol{x} - \boldsymbol{y}\|_2^\alpha} \,.
\end{aligned}
$$

This concludes the proof. $\qquad \square$

We can now prove an upper bound for the discretized estimation error at a single step. To do so, we combine Lemma G.2 and Lemma G.3 with a trick from the proof of Lemma 18 in Schwartz et al. (2025). The trick allows us to upper bound the expectation of the product of a bounded, non-negative random variable and another random variable for which we have a suitable tail bound (which comes from Lemma G.2).

**Lemma G.4.** *For any $\varepsilon \leq \widetilde{R}$,*

$$
\begin{aligned}
\mathbb{E}[\|f^{(n)}(\omega(\boldsymbol{x}_{n,h})) - f^\star(\omega(\boldsymbol{x}_{n,h}))\|_2 | \mathcal{F}_{n-1}] &\leq \max(\sqrt{32(d_s + d_a)\log(5\widetilde{R}/\varepsilon)}, 8\sqrt{\log(T)})\mathbb{E}[\sigma_{n-1}(\boldsymbol{x}_{n,h}) | \mathcal{F}_{n-1}] \\
&\quad + \sqrt{2L\varepsilon^\alpha} \max(\sqrt{32(d_s + d_a)\log(5\widetilde{R}/\varepsilon)}, 8\sqrt{\log(T)}) + C/T \,.
\end{aligned}
$$

*Proof.* We begin with the inequality

$$\mathbb{E}[\|f^{(n)}(\omega(\boldsymbol{x}_{n,h})) - f^\star(\omega(\boldsymbol{x}_{n,h}))\|_2 \,|\, \mathcal{F}_{n-1}] \leq \mathbb{E}\left[\sup_{\boldsymbol{x} \in B_\varepsilon} \frac{\|f^{(n)}(\boldsymbol{x}) - f^\star(\boldsymbol{x})\|_2}{\sigma_{n-1}(\boldsymbol{x})} \sigma_{n-1}(\omega(\boldsymbol{x}_{n,h})) \,\Big|\, \mathcal{F}_{n-1}\right] \,.$$

To save space, let us introduce the shorthand $X := \sup_{\boldsymbol{x} \in B_\varepsilon} \frac{\|f^{(n)}(\boldsymbol{x}) - f^\star(\boldsymbol{x})\|_2}{\sigma_{n-1}(\boldsymbol{x})}$ and $Y := \sigma_{n-1}(\omega(\boldsymbol{x}_{n,h}))$. Fix any

$$u \geq (32(d_s + d_a) \log(5\widetilde{R}/\varepsilon))^{1/2} \,. \tag{22}$$

Since $X$ and $Y$ are both non-negative we can upper bound the RHS as

$$\mathbb{E}[XY|\mathcal{F}_{n-1}] = \mathbb{E}[XY\mathbb{I}\{X < u\}|\mathcal{F}_{n-1}] + \mathbb{E}[XY\mathbb{I}\{X \geq u\}|\mathcal{F}_{n-1}] \leq u\mathbb{E}[Y|\mathcal{F}_{n-1}] + \mathbb{E}[XY\mathbb{I}\{X \geq u\}|\mathcal{F}_{n-1}] \,.$$

Since $\sup_{\boldsymbol{x} \in \mathbb{R}^{d_s+d_a}} c(\boldsymbol{x}, \boldsymbol{x}) \leq C$, $Y$ is upper bounded a.s. by $C$. Therefore, the second term satisfies the upper bound

$$\mathbb{E}[XY\mathbb{I}\{X \geq u\}|\mathcal{F}_{n-1}] \leq C\mathbb{E}[X\mathbb{I}\{X \geq u\}|\mathcal{F}_{n-1}] \,.$$

Depending on whether $x$ is above or below $u$, we have

$$\mathbb{P}(X\mathbb{I}\{X \geq u\} \geq x|\mathcal{F}_{n-1}) = \begin{cases} \mathbb{P}(X \geq x|\mathcal{F}_{n-1}) & x \geq u \\ \mathbb{P}(X \geq u|\mathcal{F}_{n-1}) & x < u \end{cases} \,.$$

Using this expression for the tail probability of $X\mathbb{I}\{X \geq u\}$, we have

$$\mathbb{E}[X\mathbb{I}\{X \geq u\}|\mathcal{F}_{n-1}] \leq \int_0^u \mathbb{P}(X \geq u|\mathcal{F}_{n-1})\mathrm{d}x + \int_u^\infty \mathbb{P}(X \geq x|\mathcal{F}_{n-1})\mathrm{d}x$$

$$\leq u \exp(-u^2/16) + \int_u^\infty \exp(-x^2/16)\mathrm{d}x \,,$$

where we used Lemma G.2 to upper bound the tail probability. Let us define $\phi(x) := \exp(-x^2/16)$. Using the identity $\phi(x) = -8\phi'(x)/x$ and then integrating by parts, we obtain

$$\int_u^\infty \exp(-x^2/16)\mathrm{d}x = -8 \int_u^\infty \frac{1}{x}\phi'(x)\mathrm{d}x = -8\left[\frac{1}{x}\phi(x)\right]_u^\infty - 8 \int_u^\infty \frac{1}{x^2}\phi(x)\mathrm{d}x \leq \frac{8}{u}\phi(u) \,.$$

Therefore, we have the upper bound

$$\mathbb{E}[X\mathbb{I}\{X \geq u\}|\mathcal{F}_{n-1}] \leq (u + 8/u) \exp(-u^2/16) \,.$$

Combining everything so far, we have

$$\mathbb{E}[\|f^{(n)}(\omega(\boldsymbol{x}_{n,h})) - f^\star(\omega(\boldsymbol{x}_{n,h}))\|_2 \,|\mathcal{F}_{n-1}] \leq u\mathbb{E}[\sigma_{n-1}(\omega(\boldsymbol{x}_{n,h})) \,|\mathcal{F}_{n-1}] + C(u + 8/u) \exp(-u^2/16) \,.$$

By Lemma G.3 and the fact that $B_\varepsilon$ is an $\varepsilon$-cover,

$$u\mathbb{E}[\sigma_{n-1}(\omega(\boldsymbol{x}_{n,h})) \,|\mathcal{F}_{n-1}] \leq u\mathbb{E}[\sigma_{n-1}(\boldsymbol{x}_{n,h}) \,|\mathcal{F}_{n-1}] + \sqrt{2L}u\varepsilon^{\alpha/2} \,.$$

Finally, we need to find a value of $u$ such that $(u + 8/u) \exp(-u^2/16) \leq 1/T$. By (22), we already have $u \geq 10$. For all $u \geq 10$, $u + 8/u \leq (54/5)u$. Therefore, it suffices to choose $u$ such that $54uT/5 \leq \exp(u^2/16)$. Taking logarithms, this becomes $\log(54u/5) + \log(T) \leq u^2/16$. For all $u \geq 10$, we have $\log(54u/5) \leq \frac{\log(108)}{10}u \leq \frac{\log(108)}{100}u^2$. Therefore, it suffices to choose $u$ such that

$$u^2\left(\frac{1}{16} - \frac{\log 108}{100}\right) \geq \log T \,.$$

Since $\frac{1}{16} - \frac{\log 108}{100} \geq \frac{1}{64}$, this inequality is satisfied whenever $u \geq 8\sqrt{\log T}$ (and $u \geq 10$). If we take

$$u = \max((32(d_s + d_a) \log(5\widetilde{R}/\varepsilon))^{1/2}, 8\sqrt{\log(T)}) \,,$$

then we get the inequality that we wanted. $\qquad \square$

So far, we have shown that we can upper bound the discretized estimation error by a sum of posterior standard deviations. To upper bound the sum of standard deviations, we need a version of the elliptical potential lemma that accounts for the fact that $f^{(n)}$ is only re-sampled at the end of each episode. First we re-label the observed state-action pairs

$x_{1,1}, \ldots, x_{1,H}, \ldots, x_{N,1}, \ldots, x_{N,H}$ with a single index $t \in [T]$. To try and avoid confusion, we will use $z_1, \ldots, z_T$ to denote the re-labeled sequence of state-action pairs. For any $t \in [T]$, the $t^{\text{th}}$ observation $z_t$ must occur in episode $\lfloor \frac{t-1}{H} \rfloor + 1$ at step $(t-1) \bmod H + 1$. Let us introduce the functions $n : [T] \to [N]$ and $h : [T] \to [H]$ given by

$$n(t) := \left\lfloor \frac{t-1}{H} \right\rfloor + 1 , \quad h(t) := (t-1) \bmod H + 1 .$$

We can then define $z_t := x_{n(t),h(t)}$. We can also map each state action pair $x_{n,h}$ to the corresponding element $z_t$ in the sequence $(z_t)_{t \in [T]}$. We define the function $t : [N] \times [H] \to [T]$ given by $t(n,h) := (n-1)H + h$. We then have $x_{n,h} = z_{t(n,h)}$. Next, we define the covariance function $\widetilde{\sigma}_t : \mathbb{R}^{d_s+d_a} \to \mathbb{R}$ by

$$\widetilde{\sigma}_t^2(z) := c(z,z) - \widetilde{c}_t^\top(z)(\widetilde{C}_t + \sigma^2 I)^{-1} \widetilde{c}_t(z) ,$$

where $\widetilde{c}_t(z) := [c(z, z_1), \ldots, c(z, z_t)]^\top \in \mathbb{R}^t$ and the matrix $\widetilde{C}_t \in \mathbb{R}^{t \times t}$ has $(i,j)^{\text{th}}$ element $c(z_i, z_j)$. With this notation in place, we recall the standard elliptical potential lemma (cf. e.g., Lemma 5.3 and Lemma 5.4 in Srinivas et al., 2012).

**Lemma G.5.** *For any sequence $z_1, \ldots, z_T \in \mathbb{R}^{d_s+d_a}$,*

$$\sum_{t=1}^{T} \widetilde{\sigma}_{t-1}^2(z_t) \le \frac{C}{\log(1 + C/\sigma^2)} \log \det \left( \frac{1}{\sigma^2} \widetilde{C}_T + I \right) .$$

Using Lemma G.5, one can prove a version of the elliptical potential lemma that accounts for the fact that $f^{(n)}$ is only re-sampled at the end of each episode. The result below is proved in the proof of Theorem 1 in Fan & Ming (2021).

**Lemma G.6.** *For the event $A$ defined at the beginning of Appendix G,*

$$\mathbb{E}\left[ \mathbb{I}\{A\} \sum_{t=1}^{T} \widetilde{\sigma}_{\lfloor \frac{t-1}{H} \rfloor H}^2(z_t) \right] \le \frac{2CH}{\log(1 + C/\sigma^2)} \gamma_N(\sigma^2, \widetilde{R}) .$$

*Proof.* First, we can re-write the LHS of the desired inequality as

$$\mathbb{I}\{A\} \sum_{t=1}^{T} \widetilde{\sigma}_{\lfloor \frac{t-1}{H} \rfloor H}^2(z_t) = \mathbb{I}\{A\} \sum_{n=1}^{N} \sum_{h=1}^{H} \widetilde{\sigma}_{(n-1)H}^2(z_{t(n,h)}) .$$

Let $h^\star(n) := \arg\max_{h \in [H]} \widetilde{\sigma}_{(n-1)H}^2(z_{t(n,h)})$ and let $z_{n,\max} := z_{t(n,h^\star(n))}$. For any $z$ and any $n$, we define the posterior variance at $z$ conditioned on only $z_{1,\max}, \ldots, z_{n,\max}$ as

$$\overline{\sigma}_n^2(z) := c(z,z) - \overline{c}_n(z)^\top \left( \overline{C}_n + \sigma^2 I \right)^{-1} \overline{c}_n(z) ,$$

where $\overline{c}_n(z) := [c(z, z_{1,\max}), \ldots, c(z, z_{n,\max})]^\top$ and $\overline{C}_n$ is the $n \times n$ kernel matrix with $(i,j)^{\text{th}}$ element $\{\overline{C}_n\}_{i,j} = c(z_{i,\max}, z_{j,\max})$. Since conditioning on more data can never increase the posterior variance, we have $\widetilde{\sigma}_{(n-1)H}^2(z) \le \overline{\sigma}_{n-1}^2(z)$. Using Lemma G.5, we obtain

$$\mathbb{E}\left[ \mathbb{I}\{A\} \sum_{n=1}^{N} \sum_{h=1}^{H} \widetilde{\sigma}_{(n-1)H}^2(z_{t(n,h)}) \right] \le \mathbb{E}\left[ \mathbb{I}\{A\} H \sum_{n=1}^{N} \widetilde{\sigma}_{(n-1)H}^2(z_{n,\max}) \right]$$

$$\le \mathbb{E}\left[ \mathbb{I}\{A\} H \sum_{n=1}^{N} \overline{\sigma}_{n-1}^2(z_{n,\max}) \right]$$

$$\le \mathbb{E}\left[ \mathbb{I}\{A\} \frac{CH}{\log(1 + C/\sigma^2)} \log \det \left( \frac{1}{\sigma^2} \overline{C}_N + I \right) \right]$$

$$= \frac{2CH}{\log(1 + C/\sigma^2)} \gamma_N(\sigma^2, \widetilde{R}) .$$

This concludes the proof. $\qquad \square$

Note that the quantity $\gamma_N(\sigma^2, \widetilde{R})$ that appears here is technically not the same as the quantity $\gamma_T(\sigma^2, \widetilde{R})$ that was defined in (5), even when accounting for the different subscripts. In particular, each quantity depends on the joint distribution of $N$ or $T$ state action pairs, but in each case, the joint distribution is different. In (5) $\gamma_T(\sigma^2, \widetilde{R})$ depends on the joint distribution of the first $T$ state-action pairs, which we can equivalently write as either $\boldsymbol{x}_{1,1}, \ldots, \boldsymbol{x}_{N,H}$ or $\boldsymbol{z}_1, \ldots, \boldsymbol{z}_T$. In Lemma G.6, $\gamma_N(\sigma^2, \widetilde{R})$ depends on the joint distribution of $\boldsymbol{z}_{1,\max}, \ldots, \boldsymbol{z}_{N,\max}$, which are not the first $N$ state-action pairs. Strictly speaking, we should therefore change our notation to distinguish between these quantities. However, we elect not to do this because $\gamma_N(\sigma^2, \widetilde{R})$ will only end up contributing to a lower order term in our final regret bound. The reason is that (for any kernel that satisfies Assumption 3.3 and Assumption 3.4), the expected information gain is sublinear in $T$ (or $N$) regardless of the joint distribution of the points $\boldsymbol{z}_1, \ldots, \boldsymbol{z}_T$ or $\boldsymbol{z}_{1,\max}, \ldots, \boldsymbol{z}_{N,\max}$ (cf. Lemma G.9). In any case, both $\gamma_T(\sigma^2, \widetilde{R})$ in (5) and $\gamma_N(\sigma^2, \widetilde{R})$ in Lemma G.6 are the expected information gain for $T$ or $N$ points drawn from some distribution.

Lemma G.6 already allows us to upper bound the sum of posterior standard deviations by a quantity of the order $(H\gamma_N(\sigma^2, \widetilde{R})T)^{1/2}$. However, we can use an idea from Vakili & Olkhovskaya (2024) (cf. their Lemma 4) to derive a better upper bound of the order $(\gamma_T(\sigma^2, \widetilde{R})T)^{1/2}$. To do so, we will need the inequality in Lemma G.7, which controls the ratio of posterior variances at different sample sizes, and is a restatement of Lemma 4 from Calandriello et al. (2020). Note that Calandriello et al. (2020) use a scaled version of the posterior variance. Accounting for this introduces the factor of $1/\sigma^2$ on the RHS of the inequality in Lemma G.7.

**Lemma G.7.** *For any $t' < t$ and any $\boldsymbol{z} \in \mathbb{R}^{d_s + d_a}$,*

$$\widetilde{\sigma}_{t'}^2(\boldsymbol{z}) \leq \widetilde{\sigma}_t^2(\boldsymbol{z})\left(1 + \frac{1}{\sigma^2}\sum_{s=t'+1}^{t}\widetilde{\sigma}_{t'}^2(\boldsymbol{z}_s)\right).$$

We can now prove the version of the elliptical potential lemma that we actually use. As elluded to earlier, Lemma G.8 is more or less the same as Lemma 4 from Vakili & Olkhovskaya (2024). The only (small) difference is that we are after an inequality in expectation.

**Lemma G.8.** *For the event $A$ defined at the beginning of Appendix G,*

$$\mathbb{E}\left[\mathbb{I}\{A\}\sum_{n=1}^{N}\sum_{h=1}^{H-1}\sigma_{n-1}(\boldsymbol{x}_{n,h})\right] \leq \sqrt{\frac{2C}{\log(1+C/\sigma^2)}\gamma_T(\sigma^2, \widetilde{R})\left(T + \frac{2CH^2}{\sigma^2\log(1+C/\sigma^2)}\gamma_N(\sigma^2, \widetilde{R})\right)}.$$

*Proof.* We begin with the inequality

$$\mathbb{I}\{A\}\sum_{n=1}^{N}\sum_{h=1}^{H-1}\sigma_{n-1}(\boldsymbol{x}_{n,h}) \leq \mathbb{I}\{A\}\sum_{n=1}^{N}\sum_{h=1}^{H}\widetilde{\sigma}_{(n-1)H}(\boldsymbol{z}_{t(n,h)}) = \mathbb{I}\{A\}\sum_{t=1}^{T}\widetilde{\sigma}_{\lfloor\frac{t-1}{H}\rfloor H}(\boldsymbol{z}_t).$$

We apply Lemma G.7 to each summand, with $t' = \lfloor\frac{t-1}{H}\rfloor H$ for the $t^{\text{th}}$ summand, to obtain

$$\mathbb{I}\{A\}\sum_{t=1}^{T}\widetilde{\sigma}_{\lfloor\frac{t-1}{H}\rfloor H}(\boldsymbol{z}_t) \leq \mathbb{I}\{A\}\sum_{t=1}^{T}\widetilde{\sigma}_t(\boldsymbol{z}_t)\left(1 + \frac{1}{\sigma^2}\sum_{s=\lfloor\frac{t-1}{H}\rfloor H + 1}^{t}\widetilde{\sigma}_{\lfloor\frac{t-1}{H}\rfloor H}^2(\boldsymbol{z}_s)\right)^{1/2}.$$

By the Cauchy-Schwarz inequality,

$$\mathbb{I}\{A\}\sum_{t=1}^{T}\widetilde{\sigma}_t(\boldsymbol{z}_t)\left(1 + \frac{1}{\sigma^2}\sum_{s=\lfloor\frac{t-1}{H}\rfloor H + 1}^{t}\widetilde{\sigma}_{\lfloor\frac{t-1}{H}\rfloor H}^2(\boldsymbol{z}_s)\right)^{1/2} \leq \mathbb{I}\{A\}\sqrt{\sum_{t=1}^{T}\widetilde{\sigma}_t^2(\boldsymbol{z}_t)}\sqrt{T + \frac{1}{\sigma^2}\sum_{t=1}^{T}\sum_{s=\lfloor\frac{t-1}{H}\rfloor H + 1}^{t}\widetilde{\sigma}_{\lfloor\frac{t-1}{H}\rfloor H}^2(\boldsymbol{z}_s)}$$

$$\leq \mathbb{I}\{A\}\sqrt{\sum_{t=1}^{T}\widetilde{\sigma}_t^2(\boldsymbol{z}_t)}\sqrt{T + \frac{H}{\sigma^2}\sum_{t=1}^{T}\widetilde{\sigma}_{\lfloor\frac{t-1}{H}\rfloor H}^2(\boldsymbol{z}_t)}.$$

Using Cauchy-Schwarz once more (in the form $\mathbb{E}[\sqrt{X}\sqrt{Y}] \leq \sqrt{\mathbb{E}[X]\mathbb{E}[Y]}$), we get

$$\mathbb{E}\left[\mathbb{I}\{A\}\sqrt{\sum_{t=1}^{T}\widetilde{\sigma}_t^2(\boldsymbol{z}_t)}\sqrt{T + \frac{H}{\sigma^2}\sum_{t=1}^{T}\widetilde{\sigma}_{\lfloor\frac{t-1}{H}\rfloor H}^2(\boldsymbol{z}_t)}\right] \leq \sqrt{\mathbb{E}\left[\mathbb{I}\{A\}\sum_{t=1}^{T}\widetilde{\sigma}_t^2(\boldsymbol{z}_t)\right]\mathbb{E}\left[\mathbb{I}\{A\}\left(T + \frac{H}{\sigma^2}\sum_{t=1}^{T}\widetilde{\sigma}_{\lfloor\frac{t-1}{H}\rfloor H}^2(\boldsymbol{z}_t)\right)\right]}$$

$$\leq \sqrt{\frac{2C}{\log(1+C/\sigma^2)}\gamma_T(\sigma^2,\widetilde{R})\left(T + \frac{2CH^2\sigma^{-2}}{\log(1+C/\sigma^2)}\gamma_N(\sigma^2,\widetilde{R})\right)}\,,$$

where the final inequality follows from Lemma G.6 and Lemma G.5, since

$$\mathbb{E}\left[\mathbb{I}\{A\}\sum_{t=1}^{T}\widetilde{\sigma}_t^2(\boldsymbol{z}_t)\right] \leq \mathbb{E}\left[\mathbb{I}\{A\}\sum_{t=1}^{T}\widetilde{\sigma}_{t-1}^2(\boldsymbol{z}_t)\right] \leq \frac{2C}{\log(1+C/\sigma^2)}\gamma_T(\sigma^2,\widetilde{R})\,.$$

This is the inequality that we wanted. $\qquad\square$

Before combining everything from this subsection to prove a bound on the discretized estimation error, we show that for any kernel that satisfies Assumption 3.3 and Assumption 3.4, the expected information gain has a sublinear growth rate in $T$, which means that the upper bound in Lemma G.8 is of the order $(\gamma_T(\sigma^2,\widetilde{R})T)^{1/2}$.

**Lemma G.9.** *If Assumption 3.3 and Assumption 3.4 are satisfied, then for any $\sigma \in (0,\infty)$ and any $R \in [0,\infty)$,*

$$\gamma_T(\sigma^2, R) = o(T)\,.$$

*Proof.* For the sake of clarity, we will re-write the event $A$ as $A = \{\sup_{t\in[T]}\|\boldsymbol{z}_t\|_2 \leq R\}$. First, we show that with probability 1,

$$\lim_{T\to\infty}\frac{1}{T}\mathbb{I}\{\sup_{t\in[T]}\|\boldsymbol{z}_t\|_2 \leq R\}\frac{1}{2}\log\det\left(\frac{1}{\sigma^2}\widetilde{\boldsymbol{C}}_T + \boldsymbol{I}\right) = 0\,.$$

Fix any $\varepsilon > 0$ and any sequence $(\boldsymbol{z}_t)_{t=1}^{\infty}$ of elements in $\mathbb{R}^{d_s+d_a}$. If for any $t' \geq 1$, $\boldsymbol{z}_{t'} \notin \mathbb{B}^{d_s+d_a}(R)$, then for all $T \geq t'$,

$$\mathbb{I}\{\sup_{t\in[T]}\|\boldsymbol{z}_t\|_2 \leq R\} = 0\,.$$

Therefore, we may assume that $\boldsymbol{z}_t \in \mathbb{B}^{d_s+d_a}(R)$ for all $t \geq 1$. By assumption, the kernel function $c$ is continuous, positive semi-definite and satisfies the Hilbert-Schmidt condition

$$\int_{\mathbb{B}^{d_s+d_a}(R)}\int_{\mathbb{B}^{d_s+d_a}(R)}c^2(\boldsymbol{x},\boldsymbol{y})\mathrm{d}\nu(\boldsymbol{x})\mathrm{d}\nu(\boldsymbol{y}) < \infty\,,$$

where $\nu$ is (for instance) the Lebesgue measure on $\mathbb{R}^{d_s+d_a}$. Since the closed ball $\mathbb{B}^{d_s+d_a}(R)$ is compact, Mercer's theorem (see e.g. Theorem 12.20 in Wainwright, 2019) ensures that the kernel function has the expansion

$$c(\boldsymbol{x},\boldsymbol{y}) = \sum_{m=1}^{\infty}\lambda_m\phi_m(\boldsymbol{x})\phi_m(\boldsymbol{y})\,,$$

where $(\lambda_m)_{m=1}^{\infty}$ is a sequence of non-negative real numbers. Since the kernel function is bounded by $C$,

$$\sum_{m=1}^{\infty}\lambda_m\sum_{t=1}^{T}\phi_m^2(\boldsymbol{z}_t) = \sum_{t=1}^{T}\sum_{m=1}^{\infty}\lambda_m\phi_m^2(\boldsymbol{z}_t) = \mathrm{tr}(\widetilde{C}_T) \leq CT\,.$$

Let us define the sequence $(a_m)_{m=1}^{\infty}$ by $a_m := \frac{\lambda_m}{\sigma^2}\sum_{t=1}^{T}\phi_m^2(\boldsymbol{z}_t)$. From the inequality above and the fact that $a_m \geq 0$, it follows that the sequence $(\sum_{m=1}^{M}a_m)_{M=1}^{\infty}$ of partial sums is monotone increasing and upper bounded by $CT/\sigma^2$. Therefore, the series $\sum_{m=1}^{\infty}a_m$ converges to some limit $L \leq CT/\sigma^2$. This means that there exists $M \geq 1$ such that $|\sum_{m=1}^{M}a_m - L| < \varepsilon$. This being so,

$$\sum_{m=M+1}^{\infty}a_m = \sum_{m=1}^{\infty}a_m - \sum_{m=1}^{M}a_m < \sum_{m=1}^{\infty}a_m - L + \varepsilon = \varepsilon\,.$$

By Lemma 1 from Seeger et al. (2008),

$$\log\det\left(\frac{1}{\sigma^2}\widetilde{C}_T + I\right) \leq \sum_{m=1}^{\infty}\log\left(1 + \frac{\lambda_m}{\sigma^2}\sum_{t=1}^{T}\phi_m^2(z_t)\right) = \sum_{m=1}^{\infty}\log(1 + a_m)\,.$$

By splitting the sum and then using the inequality $\log(1 + x) \leq x$, we get

$$\sum_{m=1}^{\infty}\log(1 + a_m) = \sum_{m=1}^{M}\log(1 + a_m) + \sum_{m=M+1}^{\infty}\log(1 + a_m)$$

$$\leq M\log(1 + CT/\sigma^2) + \sum_{m=M+1}^{\infty}a_m$$

$$\leq M\log(1 + CT/\sigma^2) + \varepsilon\,.$$

Combining everything so far,

$$\limsup_{T\to\infty}\frac{1}{T}\mathbb{I}\{\sup_{t\in[T]}\|z_t\|_2 \leq R\}\frac{1}{2}\log\det\left(\frac{1}{\sigma^2}\widetilde{C}_T + I\right) \leq \limsup_{T\to\infty}\frac{M\log(1 + CT/\sigma^2) + \varepsilon}{2T} = 0\,.$$

Since $\widetilde{C}_T$ is positive semi-definite, we can conclude that the sequence converges and that the limit is 0. By the AM-GM inequality, for any positive semi-definite $T \times T$ matrix $A$,

$$\det(A) = \prod_{t=1}^{T}\lambda_t \leq \left(\frac{1}{T}\sum_{t=1}^{T}\lambda_t\right)^T = (\mathrm{tr}(A)/T)^T\,.$$

Therefore, for every $T \geq 1$,

$$\left|\frac{1}{T}\mathbb{I}\{\sup_{t\in[T]}\|z_t\|_2 \leq R\}\frac{1}{2}\log\det\left(\frac{1}{\sigma^2}\widetilde{C}_T + I\right)\right| \leq \frac{1}{2}\log\frac{\mathrm{tr}(\frac{1}{\sigma^2}\widetilde{C}_T + I)}{T} \leq \frac{1}{2}\log(1 + C/\sigma^2)\,.$$

Finally, by the dominated convergence theorem,

$$\lim_{T\to\infty}\frac{\gamma_T(\sigma^2, R)}{T} = \mathbb{E}\left[\lim_{T\to\infty}\frac{1}{T}\mathbb{I}\{\sup_{t\in[T]}\|z_t\|_2 \leq R\}\frac{1}{2}\log\det\left(\frac{1}{\sigma^2}\widetilde{C}_T + I\right)\right] = 0\,.$$

This concludes the proof. $\qquad\square$

Using Lemma G.4 and G.8 we can prove the following upper bound on the discretized estimation error.

**Lemma G.10.** *For every $\varepsilon \leq \widetilde{R}$,*

$$\mathbb{E}\left[\mathbb{I}\{A\}\sum_{n=1}^{N}\sum_{h=1}^{H-1}\|f^{(n)}(\omega(x_{n,h})) - f^{\star}(\omega(x_{n,h}))\|_2\right] \leq T\sqrt{2L\varepsilon^{\alpha}}\sqrt{\max(32(d_s + d_a)\log(5\widetilde{R}/\varepsilon), 64\log(T))} + C$$

$$+ \sqrt{\max(32(d_s + d_a)\log(5\widetilde{R}/\varepsilon), 64\log(T))}\sqrt{\frac{2C}{\log(1 + C/\sigma^2)}\gamma_T(\sigma^2, \widetilde{R})\left(T + \frac{2CH^2}{\sigma^2\log(1 + C/\sigma^2)}\gamma_N(\sigma^2, \widetilde{R})\right)}\,.$$

For a suitable choice of $\varepsilon$, the RHS is of the order $\sqrt{(d_s + d_a)\gamma_T(\sigma^2, \widetilde{R})T\log(T)}$.

*Proof.* By the tower rule and Lemma G.4,

$$\mathbb{E}\left[\mathbb{I}\{A\}\sum_{n=1}^{N}\sum_{h=1}^{H-1}\|f^{(n)}(\omega(x_{n,h})) - f^{\star}(\omega(x_{n,h}))\|_2\right] = \mathbb{E}\left[\mathbb{I}\{A\}\sum_{n=1}^{N}\sum_{h=1}^{H-1}\mathbb{E}[\|f^{(n)}(\omega(x_{n,h})) - f^{\star}(\omega(x_{n,h}))\|_2|\mathcal{F}_{n-1}]\right]$$

$$\leq \max(\sqrt{32(d_s + d_a)\log(5\widetilde{R}/\varepsilon)}, 8\sqrt{\log(T)})\mathbb{E}\left[\mathbb{I}\{A\}\sum_{n=1}^{N}\sum_{h=1}^{H-1}\sigma_{n-1}(x_{n,h})\right]$$

$$+ T\sqrt{2L\varepsilon^{\alpha}}\max(\sqrt{32(d_s + d_a)\log(5\widetilde{R}/\varepsilon)}, 8\sqrt{\log(T)}) + C\,.$$

By Lemma G.8,

$$\mathbb{E}\left[\mathbb{I}\{A\} \sum_{n=1}^{N} \sum_{h=1}^{H-1} \sigma_{n-1}(\boldsymbol{x}_{n,h})\right] \le \sqrt{\frac{2C}{\log(1+C/\sigma^2)}\gamma_T(\sigma^2, \widetilde{R})\left(T + \frac{2CH^2}{\sigma^2 \log(1+C/\sigma^2)}\gamma_N(\sigma^2, \widetilde{R})\right)}.$$

This concludes the proof. $\qquad\square$

### G.2. Bounding the Discretization Errors

Let $f = (f_1, \ldots, f_{d_s})$, where $f_1, \ldots, f_{d_s} \sim \mathcal{GP}(0, c(\boldsymbol{x}, \boldsymbol{y}))$, and consider the random process $\widetilde{f}$ given by $\widetilde{f}(\boldsymbol{x}) := f(\boldsymbol{x}) - f(\omega(\boldsymbol{x}))$. It can be seen that each component $\widetilde{f}_i$ of $\widetilde{f}$ is a centered Gaussian princess with the covariance kernel $\widetilde{c} : \mathbb{R}^{d_s+d_a} \times \mathbb{R}^{d_s+d_a} \to \mathbb{R}$ given by

$$\widetilde{c}(\boldsymbol{x}, \boldsymbol{y}) := c(\omega(\boldsymbol{x}), \omega(\boldsymbol{y})) - c(\omega(\boldsymbol{x}), \boldsymbol{y}) - c(\boldsymbol{x}, \omega(\boldsymbol{y})) + c(\boldsymbol{x}, \boldsymbol{y}).$$

Since $f^\star$ and $f^{(n)}$ (and $f$) have the same marginal distribution, both of the discretisation error terms are equal to

$$\mathbb{E}\left[\mathbb{I}\{A\} \sum_{n=1}^{N} \sum_{h=1}^{H-1} \|\widetilde{f}(\boldsymbol{x}_{n,h})\|_2\right] \le \sum_{n=1}^{N} \sum_{h=1}^{H-1} \mathbb{E}\left[\sup_{\boldsymbol{x} \in \mathbb{B}^{d_s+d_a}(\widetilde{R})} \|\widetilde{f}(\boldsymbol{x})\|_2\right].$$

We upper bound the expected supremum on the right-hand side using the chaining method. We will first need to establish some properties of $\widetilde{c}$ and the corresponding natural distance $d_{\widetilde{c}}$. We would like to have an analogue of Lemma D.2, which gives an upper bound on the covering number of $\mathbb{B}^{d_s+d_a}(\widetilde{R})$ with respect to the distance $d_{\widetilde{c}}$. However, since $\omega$ is in general not continuous, neither is $d_{\widetilde{c}}$. This means we that cannot use exactly the same argument as we did in the proof of Lemma D.2. Fortunately, $d_{\widetilde{c}}$ is still piecewise Hölder continuous, and this fact can be exploited in a similar manner to before. To do so, we will use the following lemma, which establishes a relationship between covering numbers and proper covering numbers (see Appendix C.2 for the distinction between covering numbers and proper covering numbers).

**Lemma G.11.** *For any set $\mathcal{Z} \subset \mathbb{R}^{d_s+d_a}$ and any $\varepsilon > 0$ such that $\mathsf{N}(\mathcal{Z}, d_2, \varepsilon/2) < \infty$,*

$$\mathsf{N}_{\mathrm{pr}}(\mathcal{Z}, d_2, \varepsilon) \le \mathsf{N}(\mathcal{Z}, d_2, \varepsilon/2).$$

*Proof.* Let $\{\boldsymbol{x}_1, \ldots, \boldsymbol{x}_M\} \subset \mathbb{R}^{d_s+d_a}$ be an $\varepsilon/2$-cover of $\mathcal{Z}$. We partition $\mathcal{Z}$ into at most $M$ sets as follows. We define

$$B_1 := \{\boldsymbol{x} \in \mathcal{Z} : d_2(\boldsymbol{x}, \boldsymbol{x}_1) \le \varepsilon/2\}.$$

Then, for $i = 2, 3, \ldots, M$, we define

$$B_i := \{\boldsymbol{x} \in \mathcal{Z} : d_2(\boldsymbol{x}, \boldsymbol{x}_i) \le \varepsilon/2\} \setminus \left(\cup_{j=1}^{i-1} B_j\right).$$

If for any $i \in [M]$, $B_i = \emptyset$, we can remove it from the partition. In any case, the resulting partition will have at most $M$ elements. Assuming, as we may, that each $B_i \ne \emptyset$ for all $i \in [M]$, we can choose a set of points $\{\boldsymbol{y}_1, \ldots, \boldsymbol{y}_M\} \subset \mathcal{Z}$ such that for all $i \in [M]$, $\boldsymbol{y}_i \in B_i$. Fix a point $\boldsymbol{x} \in \mathcal{Z}$. Since $(B_i)_{i=1}^{M}$ is a partition of $\mathcal{Z}$, there exists a unique $j \in [M]$ such that $\boldsymbol{x} \in B_j$. Since $\boldsymbol{y}_j \in B_j$ and $\mathrm{diam}_{d_2}(B_j) \le \varepsilon$, $d_2(\boldsymbol{x}, \boldsymbol{y}_j) \le \varepsilon$. Therefore, $\{\boldsymbol{y}_1, \ldots, \boldsymbol{y}_M\}$ is a proper $\varepsilon$-cover of $\mathcal{Z}$, and the claim follows. $\qquad\square$

We can now prove an analogue of Lemma D.2.

**Lemma G.12.** *For any $\varepsilon \in (0, \widetilde{R}]$ and $\delta \in (0, \sqrt{8L\varepsilon^\alpha}]$,*

$$\mathsf{N}(\mathbb{B}^{d_s+d_a}(\widetilde{R}), d_{\widetilde{c}}, \delta) \le \left(\frac{7L^{1/2}\widetilde{R}^{\alpha/2}}{\delta}\right)^{2(d_s+d_a)/\alpha}.$$

*Proof.* Recall that $B_\varepsilon$ is a minimal $\varepsilon$-cover of $\mathbb{B}^{d_s+d_a}(\widetilde{R})$ w.r.t. the Euclidean metric $d_2$, and let $\{\boldsymbol{x}_1, \ldots, \boldsymbol{x}_M\}$ be the points in $B_\varepsilon$. We partition $\mathbb{B}^{d_s+d_a}(\widetilde{R})$ into $M$ sets $(B_i)_{i=1}^{M}$ as follows. For each $i \in [M]$, we define $B_i := \{\boldsymbol{x} \in \mathbb{B}^{d_s+d_a}(\widetilde{R}) : \omega(\boldsymbol{x}) = i\}$. We may assume that each set $B_i$ is non-empty, since if this was not the case, then $B_\varepsilon$ would not be a minimal

$\varepsilon$-cover. Since $B_\varepsilon$ is an $\varepsilon$-cover of $\mathbb{B}^{d_s+d_a}(\widetilde{R})$ w.r.t. $d_2$, for each $i$ we have $B_i \subseteq \mathbb{B}^{d_s+d_a}_{\boldsymbol{x}_i}(\varepsilon)$. For each $i$, we construct a proper $\delta_0$-cover $C_i \subseteq B_i$ of $B_i$ (w.r.t. the metric $d_2$). Fix a point $\boldsymbol{x} \in \mathbb{B}^{d_s+d_a}(\widetilde{R})$. Since $(B_i)_{i=1}^M$ is a partition of $\mathbb{B}^{d_s+d_a}(\widetilde{R})$, there exists $j \in [M]$ such that $\boldsymbol{x} \in B_j$, which means $\omega(\boldsymbol{x}) = \boldsymbol{x}_j$. Since $C_j$ is a proper $\delta_0$-cover of $B_j$, there exists $\boldsymbol{y} \in C_j \subseteq B_j$ such that $\omega(\boldsymbol{y}) = \boldsymbol{x}_j = \omega(\boldsymbol{x})$ and $\|\boldsymbol{x} - \boldsymbol{y}\|_2 \le \delta_0$. By Assumption 3.4, we have

$$
\begin{aligned}
|\widetilde{c}(\boldsymbol{x}, \boldsymbol{x}) - \widetilde{c}(\boldsymbol{x}, \boldsymbol{y})| &\le |c(\omega(\boldsymbol{x}), \omega(\boldsymbol{x})) - c(\omega(\boldsymbol{x}), \omega(\boldsymbol{y}))| + |c(\omega(\boldsymbol{x}), \boldsymbol{x}) - c(\omega(\boldsymbol{x}), \boldsymbol{y})| \\
&\quad + |c(\boldsymbol{x}, \omega(\boldsymbol{x})) - c(\boldsymbol{x}, \omega(\boldsymbol{y}))| + |c(\boldsymbol{x}, \boldsymbol{x}) - c(\boldsymbol{x}, \boldsymbol{y})| \\
&\le 2L\|\omega(\boldsymbol{x}) - \omega(\boldsymbol{y})\|_2^\alpha + 2L\|\boldsymbol{x} - \boldsymbol{y}\|_2^\alpha \le 2L\delta_0^\alpha .
\end{aligned}
$$

Therefore, if we choose $\delta_0 = (\delta^2/(4L))^{1/\alpha}$, then

$$
d_{\widetilde{c}}(\boldsymbol{x}, \boldsymbol{y}) \le \sqrt{|\widetilde{c}(\boldsymbol{x}, \boldsymbol{x}) - \widetilde{c}(\boldsymbol{x}, \boldsymbol{y})| + |\widetilde{c}(\boldsymbol{y}, \boldsymbol{x}) - \widetilde{c}(\boldsymbol{y}, \boldsymbol{y})|} \le \sqrt{4L\delta_0^\alpha} = \delta .
$$

Therefore, $\cup_{i=1}^M C_i$ is a $\delta$-cover of $\mathbb{B}^{d_s+d_a}(\widetilde{R})$ w.r.t. the metric $d_{\widetilde{c}}$. By combining Lemma G.11 and Lemma D.1, we see that the cardinality of each $C_i$ satisfies

$$
\left(1 + \frac{4\varepsilon}{\delta_0}\right)^{d_s+d_a} .
$$

Therefore, using Lemma D.1 again,

$$
|\cup_{i=1}^M C_i| \le M\left(1 + \frac{2\varepsilon}{\delta_0}\right)^{d_s+d_a} \le \left(1 + \frac{2\widetilde{R}}{\varepsilon}\right)^{d_s+d_a}\left(1 + \frac{4(4L)^{1/\alpha}\varepsilon}{\delta^{2/\alpha}}\right)^{d_s+d_a} .
$$

Since $\varepsilon \le \widetilde{R}$, we have $1 \le \widetilde{R}/\varepsilon$. Similarly, since $\delta \le \sqrt{8L\varepsilon^\alpha}$, we have $1 \le 8^{1/\alpha}L^{1/\alpha}\varepsilon/\delta^{2/\alpha}$. Therefore,

$$
\begin{aligned}
|\cup_{i=1}^M C_i| &\le \left(\frac{3\widetilde{R}}{\varepsilon}\right)^{d_s+d_a}\left(\frac{(4 \cdot 4^{1/\alpha} + 8^{1/\alpha})L^{1/\alpha}\varepsilon}{\delta^{2/\alpha}}\right)^{d_s+d_a} \\
&= \left(\frac{3(4 \cdot 4^{1/\alpha} + 8^{1/\alpha})L^{1/\alpha}\widetilde{R}}{\delta^{2/\alpha}}\right)^{d_s+d_a} \\
&= \left(\frac{3^{\alpha/2}(4 \cdot 4^{1/\alpha} + 8^{1/\alpha})^{\alpha/2}L^{1/2}\widetilde{R}^{\alpha/2}}{\delta}\right)^{2(d_s+d_a)/\alpha} .
\end{aligned}
$$

The factor of $3^{\alpha/2}(4 \cdot 4^{1/\alpha} + 8^{1/\alpha})^{\alpha/2}$ can be simplified. Since $\alpha \le 1$,

$$
3^{\alpha/2}(4 \cdot 4^{1/\alpha} + 8^{1/\alpha})^{\alpha/2} \le \sqrt{3}(2 \cdot 2^{1/\alpha} + 8^{1/(2\alpha)})^\alpha \le \sqrt{3}(4 + \sqrt{8}) \le 7 .
$$

This concludes the proof. $\qquad\square$

We would also like to have an analogue of Lemma D.3, which controls the diameter $\text{diam}_{d_{\widetilde{c}}}(\mathbb{B}^{d_s+d_a}(\widetilde{R}))$.

**Lemma G.13.** *For any $\varepsilon > 0$,*
$$
\text{diam}_{d_{\widetilde{c}}}(\mathbb{B}^{d_s+d_a}(\widetilde{R})) \le \sqrt{8L\varepsilon^\alpha} .
$$

*Proof.* Let $\boldsymbol{x}, \boldsymbol{y}$ be any pair of points in $\mathbb{B}^{d_s+d_a}(\widetilde{R})$. Due to Assumption 3.4 and the definition of $\omega$, it follows that

$$
|\widetilde{c}(\boldsymbol{x}, \boldsymbol{y})| \le |c(\omega(\boldsymbol{x}), \omega(\boldsymbol{y})) - c(\omega(\boldsymbol{x}), \boldsymbol{y})| + |c(\boldsymbol{x}, \omega(\boldsymbol{y})) - c(\boldsymbol{x}, \boldsymbol{y})| \le 2L\varepsilon^\alpha . \tag{23}
$$

Therefore,

$$
d_{\widetilde{c}}(\boldsymbol{x}, \boldsymbol{y}) = \sqrt{\widetilde{c}(\boldsymbol{x}, \boldsymbol{x}) - 2\widetilde{c}(\boldsymbol{x}, \boldsymbol{y}) + \widetilde{c}(\boldsymbol{y}, \boldsymbol{y})} \le \sqrt{8L\varepsilon^\alpha} .
$$

Since $\boldsymbol{x}$ and $\boldsymbol{y}$ were arbitrary points in $\mathbb{B}^{d_s+d_a}(\widetilde{R})$, we conclude that $\text{diam}_{d_{\widetilde{c}}}(\mathbb{B}^{d_s+d_a}(R)) \le \sqrt{8L\varepsilon^\alpha}$. $\qquad\square$

To apply the chaining method to the discretization error, we also need an upper bound for the entropy integral. For this purpose, we will use the following technical lemma.

**Lemma G.14.** *For any $y \in (0, 1/e]$,*

$$\int_0^y \sqrt{\log(1/x)}\mathrm{d}x \le 2y\sqrt{\log(1/y)}\,.$$

*Proof.* Using the substitution $u = \log(y/x)$, we have

$$\int_0^y \sqrt{\log(1/x)}\mathrm{d}x = \int_0^\infty \sqrt{\log(1/y) + u}\,y\exp(-u)\mathrm{d}u = y\sqrt{\log(1/y)}\int_0^\infty \sqrt{1 + u/\log(1/y)}\exp(-u)\mathrm{d}u\,.$$

Since $u/\log(1/y) \ge 0$, we have $1 + u/\log(1/y) \le \exp(u/\log(1/y))$. Also, since $y \le 1/e$, we have $\log(1/y) \ge 1$. Using these inequalities, we obtain

$$\int_0^\infty \sqrt{1 + u/\log(1/y)}\exp(-u)\mathrm{d}u \le \int_0^\infty \exp\left(\frac{u}{2\log(1/y)} - u\right)\mathrm{d}u \le \int_0^\infty \exp(-u/2)\mathrm{d}x = 2\,.$$

This concludes the proof. $\qquad\square$

This lemma has the following consequence.

**Corollary G.15.** *For any $z > 0$ and $y \in (0, z/e]$,*

$$\int_0^y \sqrt{\log(z/x)}\mathrm{d}x \le 2y\sqrt{\log(z/y)}\,.$$

*Proof.* Using the substitution $u = x/z$, we have

$$\int_0^y \sqrt{\log(z/x)}\mathrm{d}x = z\int_0^{y/z} \sqrt{\log(1/u)}\mathrm{d}u\,.$$

Since $y/z \le 1/e$, the claim now follows from Lemma G.14 $\qquad\square$

We can at last state and prove an upper bound for the entropy integral.

**Lemma G.16.** *For any $\varepsilon \in (0, (7/(e\sqrt{8}))^{\alpha/2}\widetilde{R}]$,*

$$\int_0^\infty \sqrt{\log \mathsf{N}(\mathbb{B}^{d_s+d_a}(\widetilde{R}), d_{\widetilde{c}}, \delta)}\mathrm{d}\delta \le 8\alpha^{-1/2}\sqrt{(d_s + d_a)L\varepsilon^\alpha \log(3\widetilde{R}^{\alpha/2}/\varepsilon^{\alpha/2})}\,.$$

*Proof.* By Lemma G.13, $\mathrm{diam}_{d_{\widetilde{c}}}(\mathbb{B}^{d_s+d_a}(\widetilde{R})) \le \sqrt{8L\varepsilon^\alpha}$, so we can replace the upper limit of the integral by $\sqrt{8L\varepsilon^\alpha}$. Since $7/(e\sqrt{8}) \le 1$, $\varepsilon \le \widetilde{R}$. Thus by Lemma G.12,

$$\int_0^{\sqrt{8L\varepsilon^\alpha}} \sqrt{\log \mathsf{N}(\mathbb{B}^{d_s+d_a}(\widetilde{R}), d_{\widetilde{c}}, \delta)}\mathrm{d}\delta \le \sqrt{2(d_s + d_a)/\alpha}\int_0^{\sqrt{8L\varepsilon^\alpha}} \sqrt{\log(7L^{1/2}\widetilde{R}^{\alpha/2}/\delta)}\mathrm{d}\delta\,.$$

Since $\varepsilon \le (7/(e\sqrt{8}))^{\alpha/2}\widetilde{R}$, $\sqrt{8L\varepsilon^\alpha} \le 7L^{1/2}\widetilde{R}^{\alpha/2}/e$. Thus by Corollary G.15,

$$\int_0^{\sqrt{8L\varepsilon^\alpha}} \sqrt{\log(7L^{1/2}\widetilde{R}^{\alpha/2}/\delta)}\mathrm{d}\delta \le 2\sqrt{8L\varepsilon^\alpha \log\left(\frac{7\widetilde{R}^{\alpha/2}}{8^{1/2}\varepsilon^{\alpha/2}}\right)} \le 2\sqrt{8L\varepsilon^\alpha \log(3\widetilde{R}^{\alpha/2}/\varepsilon^{\alpha/2})}\,.$$

This concludes the proof. $\qquad\square$

Finally, we are ready to upper bound each of the discretization errors.

**Lemma G.17.** *Suppose that Assumption 3.3 and Assumption 3.4 are satisfied. Let $f_1, \ldots, f_{d_s} \sim \mathcal{GP}(0, c(\boldsymbol{x}, \boldsymbol{y}))$ be independent, centered Gaussian processes and let $f = (f_1, \ldots, f_{d_s})$. Let $\widetilde{f}$ be the random process defined by $\widetilde{f}(\boldsymbol{x}) := f(\boldsymbol{x}) - f(\omega(\boldsymbol{x}))$. For any $\varepsilon \in (0, (7/(e\sqrt{8}))^{\alpha/2}\widetilde{R}]$,*

$$\mathbb{E}\left[\sup_{\boldsymbol{x}\in\mathbb{B}^{d_s+d_a}(\widetilde{R})} \|\widetilde{f}(\boldsymbol{x})\|_2\right] \le 96\alpha^{-1/2}\sqrt{(d_s + d_a)L\varepsilon^\alpha \log(3\widetilde{R}^{\alpha/2}/\varepsilon^{\alpha/2})} + 6\sqrt{8L\varepsilon^\alpha} + \sqrt{2d_s L\varepsilon^\alpha}\,.$$

*Proof.* First, we apply the chaining argument from Lemma 4.5, except with the kernel $\widetilde{c}$. The proof of Lemma 4.5 (with the kernel $c$) uses the fact that $\sup_{\boldsymbol{x}\in\mathbb{R}^{d_s+d_a}} c(\boldsymbol{x},\boldsymbol{x}) \leq C$ and $\mathrm{diam}_{d_c}(\mathbb{B}^{d_s+d_a}(R)) \leq 2\sqrt{C}$. Lemma G.13 gives us an upper bound for $\mathrm{diam}_{d_{\widetilde{c}}}(\mathbb{B}^{d_s+d_a}(\widetilde{R}))$. From (23), we have $\sup_{\boldsymbol{x}\in\mathbb{R}^{d_s+d_a}} \widetilde{c}(\boldsymbol{x},\boldsymbol{x}) \leq 2L\varepsilon^\alpha$. Therefore, Lemma 4.5 tells us that

$$\mathbb{E}\left[\sup_{\boldsymbol{x}\in\mathbb{B}^{d_s+d_a}(\widetilde{R})}\|\widetilde{f}(\boldsymbol{x})\|_2\right] \leq 12\int_0^\infty \sqrt{\log\mathsf{N}(\mathbb{B}^{d_s+d_a}(\widetilde{R}), d_{\widetilde{c}}, \delta)}\mathrm{d}\delta + 6\sqrt{8L\varepsilon^\alpha} + \sqrt{2d_s L\varepsilon^\alpha}\,.$$

Using Lemma G.16 to upper bound the entropy integral, we obtain

$$\mathbb{E}\left[\sup_{\boldsymbol{x}\in\mathbb{B}^{d_s+d_a}(\widetilde{R})}\|\widetilde{f}(\boldsymbol{x})\|_2\right] \leq 96\alpha^{-1/2}\sqrt{(d_s+d_a)L\varepsilon^\alpha\log(3\widetilde{R}^{\alpha/2}/\varepsilon^{\alpha/2})} + 6\sqrt{8L\varepsilon^\alpha} + \sqrt{2d_s L\varepsilon^\alpha}\,.$$

This concludes the proof. $\qquad\square$

## G.3. Proof of Lemma 4.10

We prove Lemma 4.10 by combining the inequalities in Lemma G.10 and Lemma G.17, and then finding a good value for $\varepsilon$.

*Proof of Lemma 4.10.* Recall that, for some $\varepsilon \in (0, (7/(e\sqrt{8}))^{\alpha/2}\widetilde{R}]$, $B_\varepsilon$ is a minimal $\varepsilon$-cover of $\mathbb{B}^{d_s+d_a}(\widetilde{R})$ w.r.t. the Euclidean metric $d_2$, and $\omega : \mathbb{B}^{d_s+d_a}(\widetilde{R}) \to B_\varepsilon$ is given by $\omega(\boldsymbol{x}) := \arg\min_{\boldsymbol{y}\in B_\varepsilon} d_2(\boldsymbol{x},\boldsymbol{y})$. Using the triangle inequality, we obtain

$$\mathbb{E}\left[\mathbb{I}\{A\}\sum_{n=1}^N\sum_{h=1}^{H-1}\|f^{(n)}(\boldsymbol{x}_{n,h}) - f^\star(\boldsymbol{x}_{n,h})\|_2\right] \leq \mathbb{E}\left[\mathbb{I}\{A\}\sum_{n=1}^N\sum_{h=1}^{H-1}\|f^{(n)}(\omega(\boldsymbol{x}_{n,h})) - f^\star(\omega(\boldsymbol{x}_{n,h}))\|_2\right]$$
$$+ \mathbb{E}\left[\mathbb{I}\{A\}\sum_{n=1}^N\sum_{h=1}^{H-1}\|f^{(n)}(\boldsymbol{x}_{n,h}) - f^{(n)}(\omega(\boldsymbol{x}_{n,h}))\|_2\right]$$
$$+ \mathbb{E}\left[\mathbb{I}\{A\}\sum_{n=1}^N\sum_{h=1}^{H-1}\|f^\star(\boldsymbol{x}_{n,h}) - f^\star(\omega(\boldsymbol{x}_{n,h}))\|_2\right]\,.$$

Using Lemma G.10 and G.17 to upper bound each term on the right-hand side, we obtain

$$\mathbb{E}\left[\mathbb{I}\{A\}\sum_{n=1}^N\sum_{h=1}^{H-1}\|f^{(n)}(\boldsymbol{x}_{n,h}) - f^\star(\boldsymbol{x}_{n,h})\|_2\right] \leq T\sqrt{2L\varepsilon^\alpha}\sqrt{\max(32(d_s+d_a)\log(5\widetilde{R}/\varepsilon), 64\log(T))} + C$$
$$+ \sqrt{\max(32(d_s+d_a)\log(5\widetilde{R}/\varepsilon), 64\log(T))}\sqrt{\frac{2C}{\log(1+C/\sigma^2)}\gamma_T(\sigma^2,\widetilde{R})\left(T + \frac{2CH^2}{\sigma^2\log(1+C/\sigma^2)}\gamma_N(\sigma^2,\widetilde{R})\right)}$$
$$+ 192T\alpha^{-1/2}\sqrt{(d_s+d_a)L\varepsilon^\alpha\log(3\widetilde{R}^{\alpha/2}/\varepsilon^{\alpha/2})}$$
$$+ 12T\sqrt{8L\varepsilon^\alpha} + 2T\sqrt{2d_s L\varepsilon^\alpha}\,.$$

We choose $\varepsilon = 1/(L^{1/\alpha}T^{2/\alpha})$. The upper bound on the estimation error becomes

$$\mathbb{E}\left[\mathbb{I}\{A\}\sum_{n=1}^N\sum_{h=1}^{H-1}\|f^{(n)}(\boldsymbol{x}_{n,h}) - f^\star(\boldsymbol{x}_{n,h})\|_2\right] \leq \alpha^{-1/2}\sqrt{64(d_s+d_a)\log(\max(1, 5\widetilde{R}^\alpha L)T^2)} + C$$
$$+ \alpha^{-1/2}\sqrt{\frac{64C(d_s+d_a)}{\log(1+C/\sigma^2)}\gamma_T(\sigma^2,\widetilde{R})\left(T + \frac{2CH^2}{\sigma^2\log(1+C/\sigma^2)}\gamma_N(\sigma^2,\widetilde{R})\right)\log(\max(1, 5\widetilde{R}^\alpha L)T^2)}$$
$$+ 192\alpha^{-1/2}\sqrt{(d_s+d_a)\log(3\widetilde{R}^{\alpha/2}L^{1/2}T)} + 12\sqrt{8} + 2\sqrt{2d_s}\,.$$

This concludes the proof. $\qquad\square$

# H. Proof of Theorem 4.11

*Proof of Theorem 4.11.* By Lemma 4.8, we can re-write the Bayesian regret as a sum of value estimation errors. In particular,

$$
\begin{aligned}
\mathcal{R}_T &= \mathbb{E}\left[\sum_{n=1}^{N} V_{\pi^\star,1}^{\mathcal{M}^\star}(\boldsymbol{s}_{n,1}) - V_{\pi_n,1}^{\mathcal{M}^\star}(\boldsymbol{s}_{n,1})\right] \\
&= \mathbb{E}\left[\sum_{n=1}^{N} V_{\pi^\star,1}^{\mathcal{M}^\star}(\boldsymbol{s}_{n,1}) - V_{\pi_n,1}^{\mathcal{M}_n}(\boldsymbol{s}_{n,1}) + V_{\pi_n,1}^{\mathcal{M}_n}(\boldsymbol{s}_{n,1}) - V_{\pi_n,1}^{\mathcal{M}^\star}(\boldsymbol{s}_{n,1})\right] \\
&= \mathbb{E}\left[\sum_{n=1}^{N} \mathbb{E}\left[V_{\pi^\star,1}^{\mathcal{M}^\star}(\boldsymbol{s}_{n,1}) - V_{\pi_n,1}^{\mathcal{M}_n}(\boldsymbol{s}_{n,1}) + V_{\pi_n,1}^{\mathcal{M}_n}(\boldsymbol{s}_{n,1}) - V_{\pi_n,1}^{\mathcal{M}^\star}(\boldsymbol{s}_{n,1})\big|\mathcal{F}_{n-1}\right]\right] \\
&= \mathbb{E}\left[\sum_{n=1}^{N} V_{\pi_n,1}^{\mathcal{M}_n}(\boldsymbol{s}_{n,1}) - V_{\pi_n,1}^{\mathcal{M}^\star}(\boldsymbol{s}_{n,1})\right].
\end{aligned}
$$

Let us define

$$
R := 168\alpha^{-1/2}\sqrt{\max(C,\sigma^2)(d_s + d_a)\log(10(T + R_a)\max(1, L/C))}.
$$

Also, let $\widetilde{R} = \sqrt{R^2 + R_a^2}$. We define the event $A$ as

$$
A := \left\{\sup_{n\in[N],h\in[H]} \|\boldsymbol{s}_{n,h}\|_2 \le R\right\}.
$$

Since the reward function is bounded between $-R_{\max}$ and $R_{\max}$, we have $|V_{\pi_n,1}^{\mathcal{M}_n}(\boldsymbol{s}_{n,1})| \le R_{\max}H$ and $|V_{\pi_n,1}^{\mathcal{M}^\star}(\boldsymbol{s}_{n,1})| \le R_{\max}H$ almost surely. Thus by Lemma 4.7

$$
\begin{aligned}
\mathbb{E}\left[\sum_{n=1}^{N} V_{\pi_n,1}^{\mathcal{M}_n}(\boldsymbol{s}_{n,1}) - V_{\pi_n,1}^{\mathcal{M}^\star}(\boldsymbol{s}_{n,1})\right] &= \mathbb{E}\left[(\mathbb{I}\{A\} + \mathbb{I}\{A^c\})\sum_{n=1}^{N} V_{\pi_n,1}^{\mathcal{M}_n}(\boldsymbol{s}_{n,1}) - V_{\pi_n,1}^{\mathcal{M}^\star}(\boldsymbol{s}_{n,1})\right] \\
&\le \mathbb{E}\left[\mathbb{I}\{A\}\sum_{n=1}^{N} V_{\pi_n,1}^{\mathcal{M}_n}(\boldsymbol{s}_{n,1}) - V_{\pi_n,1}^{\mathcal{M}^\star}(\boldsymbol{s}_{n,1})\right] + 2R_{\max}T\mathbb{P}(A^c) \\
&\le \mathbb{E}\left[\mathbb{I}\{A\}\sum_{n=1}^{N} V_{\pi_n,1}^{\mathcal{M}_n}(\boldsymbol{s}_{n,1}) - V_{\pi_n,1}^{\mathcal{M}^\star}(\boldsymbol{s}_{n,1})\right] + 4R_{\max}.
\end{aligned}
$$

Next, by Lemma 4.9, we have

$$
\mathbb{E}\left[\mathbb{I}\{A\}\sum_{n=1}^{N} V_{\pi_n,1}^{\mathcal{M}_n}(\boldsymbol{s}_{n,1}) - V_{\pi_n,1}^{\mathcal{M}^\star}(\boldsymbol{s}_{n,1})\right] \le \frac{R_{\max}H}{\sigma}\mathbb{E}\left[\mathbb{I}\{A\}\sum_{n=1}^{N}\sum_{h=1}^{H-1} \|f^{(n)}(\boldsymbol{x}_{n,h}) - f^\star(\boldsymbol{x}_{n,h})\|_2\right] + 2R_{\max}H\sqrt{2\pi T}.
$$

Finally, by Lemma 4.10, we have

$$
\begin{aligned}
\frac{R_{\max}H}{\sigma}\mathbb{E}\left[\mathbb{I}\{A\}\sum_{n=1}^{N}\sum_{h=1}^{H-1} \|f^{(n)}(\boldsymbol{x}_{n,h}) - f^\star(\boldsymbol{x}_{n,h})\|_2\right] &\le \frac{R_{\max}H}{\sigma\alpha^{1/2}}\sqrt{64(d_s + d_a)\log(\max(1, 5\widetilde{R}^\alpha L)T^2)} + \frac{CR_{\max}H}{\sigma} \\
&\quad + \frac{R_{\max}H}{\sigma\alpha^{1/2}}\sqrt{\frac{64C(d_s + d_a)}{\log(1 + C/\sigma^2)}\gamma_T(\sigma^2, \widetilde{R})\left(T + \frac{2CH^2}{\sigma^2\log(1 + C/\sigma^2)}\gamma_N(\sigma^2, \widetilde{R})\right)\log(\max(1, 5\widetilde{R}^\alpha L)T^2)} \\
&\quad + \frac{192R_{\max}H}{\sigma\alpha^{1/2}}\sqrt{(d_s + d_a)\log(3\widetilde{R}^{\alpha/2}L^{1/2}T)} + \frac{12\sqrt{8}R_{\max}H}{\sigma} + \frac{2\sqrt{2d_s}R_{\max}H}{\sigma}.
\end{aligned}
$$

Therefore, the Bayesian regret satisfies

$$
\begin{aligned}
\mathcal{R}_T \leq{} & \frac{8R_{\max}H}{\sigma\alpha^{1/2}}\sqrt{\frac{C(d_s+d_a)}{\log(1+C/\sigma^2)}\gamma_T(\sigma^2,\widetilde{R})\left(T+\frac{2CH^2}{\sigma^2\log(1+C/\sigma^2)}\gamma_N(\sigma^2,\widetilde{R})\right)\log(\max(1,5\widetilde{R}^\alpha L)T^2)} \\
& + \frac{8R_{\max}H}{\sigma\alpha^{1/2}}\sqrt{(d_s+d_a)\log(\max(1,5\widetilde{R}^\alpha L)T^2)} + \frac{CR_{\max}H}{\sigma} \\
& + \frac{192R_{\max}H}{\sigma\alpha^{1/2}}\sqrt{(d_s+d_a)\log(3\widetilde{R}^{\alpha/2}L^{1/2}T)} + \frac{12\sqrt{8}R_{\max}H}{\sigma} + \frac{2\sqrt{2d_s}R_{\max}H}{\sigma} + 4R_{\max} + 2R_{\max}H\sqrt{2\pi T}\,.
\end{aligned}
$$

The dominant term on the right-hand side is the first one, and its growth-rate matches the one stated in Theorem 4.11.

$\square$

Reward, $r(\mathbf{s})$

Episodic state trajectories

Evaluation state trajectories

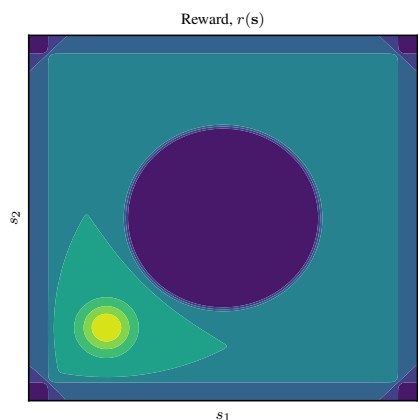

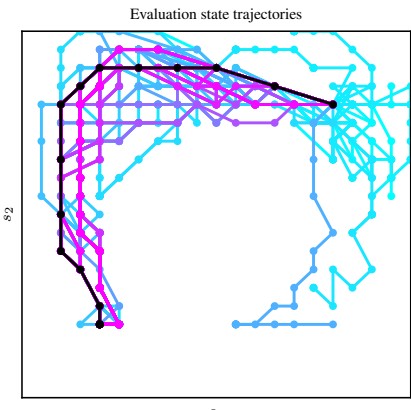

*Figure 4.* The reward function for our navigation-based experimental study, with a goal state, state limit boundary and central obstacle.

*Figure 5.* Exploration of GP-PSRL for one seed over 200 episodes with a SE kernel prior. Episode progression is shown from cyan to magenta.

*Figure 6.* Illustrating the episodic performance improvement of GP-PSRL for one seed over 200 episodes with a SE kernel prior and a constant initial state. Episode progression is shown from cyan to magenta.

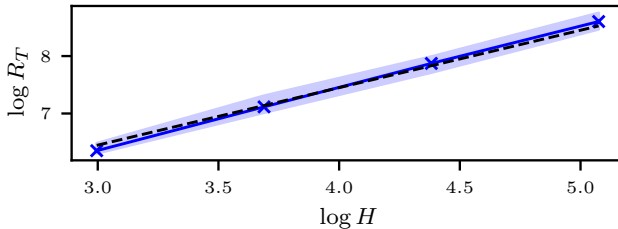

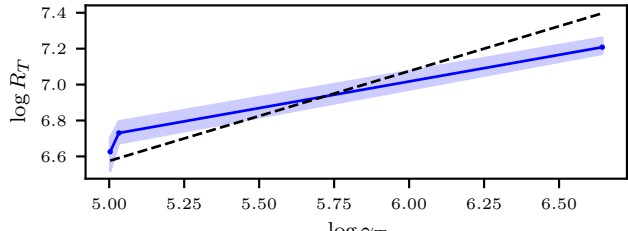

*Figure 7.* For the squared exponential kernel, we ran PSRL across horizons 20, 40, 80 and 160 for 20 seeds. Our predicted rate of $\mathcal{O}(H)$ (dashed line) compares favorably to the actual rate.

*Figure 8.* For the Matérn kernels, we can use the approximation of $\Gamma_T$ in Lemma 4.12 to estimate the growth rate of the Bayesian regret w.r.t. $\Gamma_T$. The empirical rate appears to be slightly better than $\sqrt{\Gamma_T}$ (dashed line).

## I. Experimental Results

For the experimental study, we considered a 2D navigation task where $d_s = 2$ and $d_a = 2$. The reward function contained potential functions for a goal state, a central circle 'obstacle' and barrier functions inside a state boundary (Fig 4). For the optimal control oracle and regret analysis, we discretized the system within this state boundary and performed value iteration to compute approximate optimal policies and compute the exact regret for the approximated MDP. For the function prior, we used stationary GPs of the form $\mathbf{s}_{h+1} = \mathbf{s}_h + \Delta \cdot f(\mathbf{s}_h, \mathbf{a}_h)$, essentially a stationary velocity prior under Euler integration. To estimate the Bayesian regret, we sampled a GP from this prior to be the 'ground truth', and then performed GP-PSRL, computing regret by evaluating the policy on the 'ground truth' discretized function sample. To approximate the stationary GP with a parametric model and to benefit from explicit function samples, we used random Fourier features (Rahimi & Recht, 2007) constructed from the spectral density of the covariance functions. The GP prior had fixed hyperparameters, with a prior variance of 1, a lengthscale of 0.5 and aleatoric noise variance of $1 \times 10^{-6}$, and used 1000 random features. This prior was designed such that samples from the discretized action space sufficiently explored the state space across several function samples from the prior, and also such that the optimal policy solved the navigation task across function samples. For the MDP, we used a uniform initial state distribution and a nominal task horizon of 20. These were chosen to ensure adequate exploration and optimal solutions, as shown in Figures 5 and 6.

To evaluate our theoretical rates empirically, Figure 7 looks at the Bayesian regret as $H$ is increased from 20 to 160. Figure 8 looks at the Bayesian regret for different maximum information gains for the Matérn kernels.

