# OpenReview forum: "Posterior Sampling Reinforcement Learning with Gaussian Processes for Continuous Control: Sublinear Regret Bounds for Unbounded State Spaces"
_ICML.cc/2026/Conference — ICML 2026 regular_

### Official Review · Reviewer_ufsk · 2026-02-20

**Soundness:** 3
**Presentation:** 3
**Significance:** 3
**Originality:** 3
**Overall Recommendation:** 5
**Confidence:** 2

**Summary:**

This paper analyzes the Bayesian regret bound of the Gaussian process posterior sampling reinforcement learning (PSRL) algorithm, focusing on a more general setting with an unbounded state space. The established regret bound achieves a tighter dependence on the maximum information gain.

**Compliance With Llm Reviewing Policy:**

Affirmed.

**Final Justification:**

I am not an expert in continuous settings, but I think that there has been less focus on rigorous theoretical work in continuous control area, and this paper could contribute to that area.

**Key Questions For Authors:**

See above

**Limitations:**

yes

**Strengths And Weaknesses:**

Strengths:
1. The paper is well written and easy to follow.
2. The use of the Borell–Tsirelson–Ibragimov–Sudakov inequality together with Dudley’s chaining method is a key technical contribution for bounding all states observed by the algorithm with high probability. This technique could potentially be adapted to the analysis of other GP-based algorithms.
3. The paper considers a more general setting with an unbounded state space and mild smoothness assumptions on the kernel, while achieving a tighter regret bound in terms of the maximum information gain.


Weaknesses and questions:
1. How do the authors derive Inequality (4)? It seems to rely on the implication that the event $|s_{n,h-1}|_2 \le R_{h-1}$ implies $\|s_{n,h}\|_2 \le R_h$.  If this is the case, it is unclear why if $\|s_{n,h-1}\|_2 \le R_{h-1}$ holds, we have $\|s_{n,h}\|_2 \le R_{h}$.
2. Are there any existing results on lower bounds for the regret of PSRL? It would be useful to understand whether the obtained upper bound is minimax optimal or if a gap remains.
3. Can the results of this paper be extended to high-probability regret bounds, rather than Bayesian regret bounds?

---

> ### Author Rebuttal · Authors · 2026-03-31
>
> Thank you for your your comments and questions, and for your positive assessment of our work. We address the points under "weaknesses and questions" below.
>
> *How do the authors derive Inequality (4)?*
>
> It is correct that $\|s\_{n,h-1}\|_2 \leq R\_{h-1}$ does not necessarily imply that $\|s\_{n,h}\|_2 \leq R\_{h}$. Informally, (4) relies on the fact that if $\|s\_{n,h-1}\|_2 \leq R\_{h-1}$ then the probability that the event $\|s\_{n,h}\|_2 > R\_{h}$ occurs is "small" (for suitable values of $R\_{h-1}$ and $R\_h$). If the previous state is bounded, then $f^{\star}(s\_{n,h-1}, a\_{n,h-1})$ is upper bounded by the supremum of a Gaussian process over a bounded domain (which is sub-Gaussian thanks to the BTIS inequality). Therefore, the norm of the next state is upper bounded by a sum of two sub-Gaussian random variables, which is also sub-Gaussian. This means that the tail probability $\mathbb{P}(\|s\_{n,h}\|_2 > R\_{h})$ decays rapidly as $R\_h$ increases.
>
> *Are there any existing results on lower bounds for the regret of PSRL?*
>
> We are not aware of any lower bounds for the Bayesian regret in finite-horizon MDPs with a GP prior. However, since GP bandits are a special case (when the horizon is $H = 1$), one can get an idea of what the best achievable Bayesian regret is by looking at lower bounds on the Bayesian regret for linear or Gaussian process bandits. For instance, Rusmevichientong and Tsitsiklis (2010) showed that for linear bandits with $d$-dimensional actions (with $d > 1$) and a Gaussian prior, the Bayesian regret of any algorithm is of the order $d\sqrt{T}$. This means that a Bayesian regret bound of the order $\sqrt{dT}(\gamma_T)^{p}$ is not possible for any power $p < 1/2$. Therefore, we expect that the dependence of our regret bound on $T$ is close to optimal, though it may be possible to replace the maximum information gain with a less pessimistic notion of information gain (see our response to Reviewer Koqh). There may be some room for improvement in the dependence on $H$ (perhaps from $H^{3/2}$ to $H$), since PSRL is known to satisfy Bayesian regret bounds for tabular MDPs with linear dependence on $H$ (Osband and Van Roy, 2017).
>
> Paat Rusmevichientong and John Tsitsiklis, Linearly parameterized bandits, Mathematics of Operations Research (2010)
>
> Ian Osband and Benjamin Van Roy, Why is Posterior Sampling Better than Optimism for Reinforcement Learning?, ICML (2017)
>
> *Can the results of this paper be extended to high-probability regret bounds, rather than Bayesian regret bounds?*
>
> As long as the true transition kernel is still a random draw from a Gaussian process, we expect that one could obtain a regret bound with the same growth-rate that holds with high probability. Controlling the worst-case regret (in expectation or in probability), where, for instance, the true transition kernel is any fixed function belonging to a reproducing kernel Hilbert space, would be trickier. One of the main difficulties is that Lemma 4.5 no longer holds, since $f^{\star}$ and $f^{(n)}$ no longer have the same conditional distribution. A potential solution is to use an "optimistic" or "feel-good" version of posterior sampling (Zhang, 2022; Neu et al., 2024), in which one replaces $f^{(n)}$, a random draw from the Bayesian posterior, with a random draw from an optimistic version of the posterior. This fixes the previous problem (i.e. an inequality similar to the identity in Lemma 4.5 now holds). However, it introduces another problem. The optimistic posterior is not a Gaussian process, which means bounding the estimation error would require some additional ideas or tricks.
>
> Tong Zhang, Feel-Good Thompson Sampling for Contextual Bandits and Reinforcement Learning, SIAM Journal on Mathematics of Data Science (2022)
>
> Gergely Neu, Matteo Papini, and Ludovic Schwartz, Optimistic Information Directed Sampling, COLT (2024)

---

> > ### Author Rebuttal · Reviewer_ufsk · 2026-04-01
> >
> > Thanks for your response. I decide to maintain my positive score.

---

> > > ### Author Response · Authors · 2026-04-06
> > >
> > > Thank you for your answer. We’re happy to hear that your concerns have been resolved.

---

### Official Review · Reviewer_Koqh · 2026-02-25

**Soundness:** 3
**Presentation:** 2
**Significance:** 3
**Originality:** 2
**Overall Recommendation:** 4
**Confidence:** 4

**Summary:**

This paper analyzes the posterior sampling algorithm, so-called Thompson sampling algorithm, for the $H$-horizon reinforcement learning algorithm. They assume the Bayesian condition that the transition probability is defined by Gaussian processes (GPs). The authors provide the regret upper bound without a probabilistic Lipschitz continuity condition for the sample path from GPs, which requires a higher-order smoothness condition for the kernel function compared with the assumption of this paper. The resulting regret upper bound is sublinear at least for Matern kernel families with $\nu > 1/2$.

**Compliance With Llm Reviewing Policy:**

Affirmed.

**Final Justification:**

The authors adequately addressed my concerns.

However, this discussion clarifies the paper's position, which implies that problems 1 and 3 have been addressed separately in existing studies.
In addition, this paper only slightly tightens the regret upper bound, although the first submitted paper can be read as if it obtained a near-optimal regret bound (Problem 2).
Thus, although the theoretical contribution of this paper is solid, I feel that the novelty (originality) of this paper is somewhat limited, and the presentation should be improved.
In addition to those clarifications of the contribution, the paper should be revised in several minor points.

Therefore, although I raised the score from 3 to 4, I also feel that appropriate revisions are necessary for acceptance.

**Key Questions For Authors:**

Please answer the above comments.
Furthermore;

7. Can we obtain the kernel-specific result for the squared exponential kernel?
8. The regret upper bound of Bayrooti et al. (2025b)  shown in Table 1 seems to be tighter than the proposed one (though there are differences in high-probability or expectation and the required assumption). Is my understanding correct?

**Limitations:**

yes

**Strengths And Weaknesses:**

I agree that relaxing the kernel assumption is an important direction.
In addition, though I could not follow the proof entirely, the theoretical statements appear mostly rigorous.

On the other hand, I think the following about the paper.
1. Relaxing the kernel assumption is partially dealt with at least by [1]. Thus, explicit discussion on [1] is required.

[1] Emile Contal, Nicolas Vayatis, Stochastic Process Bandits: Upper Confidence Bounds Algorithms via Generic Chaining, arXiv 2016.

2. I could not understand the literature on the problems 1. I believe that this problem occurs whenever the noise exists. Do the existing studies ignore this problem or consider a noise-free problem?

3. I thought that Problem 2 is not resolved in this paper since this paper concentrates on the Bayesian setting, which does not encompass the frequentist setting, where the function lies in the reproducing kernel Hilbert space. Furthermore, as discussed in the comments below, the near-optimality of the proposed regret bound does not appear to be shown.

4. The authors use the words "tight" and "near-optimal" in the abstract and the introduction. However, the analysis for the regret lower bound in the Bayesian setting is limited to [2]. On the other hand, [3] shows $O(\sqrt{T})$ high-probability regret upper bound for the usual optimization problem. Thus, I believe that near-optimality for this problem setup has not been shown, and I expect that the result $O(\sqrt{T \gamma\_T})$ is not near-optimal. Please rephrase "tight" and "near-optimal" with other words.

[2] Jonathan Scarlett, Tight Regret Bounds for Bayesian Optimization in One Dimension, ICML 2018.

[3] Shogo Iwazaki, Improved Regret Bounds for Gaussian Process Upper Confidence Bound in Bayesian Optimization, NeurIPS 2025.

5. The regret upper bound of the proposed method in Table 1 seems to be incorrect for the subscript of $\gamma$.

6. For Lemma 4.9, I believe that $\nu > 1/2$ is at least required. For example, see Remark 2 of [4]

[4] Sattar Vakili, Kia Khezeli, Victor Picheny, On Information Gain and Regret Bounds in Gaussian Process Bandits, AISTATS 2021.

- (minor) In page 2, there is a typographical error "bounded bounded."
- (minor) There is no definition of $x\_{n, h}$.

---

> ### Author Rebuttal · Authors · 2026-03-31
>
> Thank you for your comments and questions. We address some specific comments below.
>
> *Discussion on [1] is required.*
>
> We agree that [1] uses a version of chaining to prove Bayesian regret bounds for GP bandits under similar smoothness conditions. We will discuss it. There are several subtle differences between each use of the chaining method. For instance, in [1], the tree of discretizations used in the chaining argument is also used to construct the algorithm. In our work, the sequence of covers is used purely for analysis, which means we can analyze the regret of standard algorithms that do not necessarily discretize the domain.
>
> *Literature on problem 1.*
>
> This problem occurs whenever random noise with unbounded support is added to the states, but can also occur with noiseless states. If each state is given by a function $f$ of the previous state and action, and $f$ is a random draw from a GP, then the states still have unbounded support. Some studies do ignore this problem. Others do not, but obtain much worse bounds on the norm of the largest state. E.g., Lemma 26 in Curi et al. (2020) shows that when the mean function $f$ of the transition kernel is $L$-Lipschitz, with high probability, the norm of every state is bounded by a quantity of the order $O(L^{T-1}H\sqrt{d + \log(T)})$.
>
> *Problem 2 is not resolved (Bayesian vs frequentist setting)*
>
> Problem 2 is referring to the fact that existing bounds on the Bayesian regret have at least linear dependence on the information gain. The only exceptions we are aware of are Theorem 3 and Theorem 4 of Chowdhury and Gopalan (2019), which have square root dependence on $\gamma\_T$. However, these regret bounds assume that the state space is a hypercube, which is incompatible with both the Gaussian transition kernel and the GP prior on the dynamics.
>
> *Please rephrase "tight" and "near-optimal".*
>
> We agree that the tightness of our regret bound has not been established in the sense that it matches a lower bound on the Bayesian regret. We propose to describe the rates of our regret bounds as “improved” rather than “tight” or “near-optimal”.
>
> We would be happy to add some of the following comments about references [2,3] and the optimality/sub-optimality of our regret bound. While the lower bound in [2] suggests that $\sqrt{T\gamma\_T}$ is not tight, Rusmevichientong and Tsitsiklis (2010) showed that dimension one is a special case when it comes to the Bayesian regret. The optimal Bayesian regret for one-dimensional linear bandits is of the order $\log(T)$, whereas in dimension $d > 1$ it is of the order $d\sqrt{T}$. This result also shows that (for GP bandits) a Bayesian regret bound of the order $\sqrt{dT}(\gamma\_T)^{p}$ is not possible for any $p < 1/2$. Based on [3], one might guess that (under additional smoothness conditions) a Bayesian regret bound of the order $\sqrt{T\cdot I\_T}$ is possible, where $I\_T$ is the (expected) information gain evaluated at the state-action pairs observed by GP-PSRL. However, this has not been established in any work that we are aware of, and remains an open question.
>
> Paat Rusmevichientong and John Tsitsiklis, Linearly parameterized bandits, Mathematics of Operations Research (2010)
>
> *Incorrect subscript for $\gamma$ in Table 1.*
>
> The subscript should be $T/H$. We will fix this.
>
> *$\nu > 1/2$ required.*
>
> We do not use the bound on the information gain from [4]. We use Lemma 2 from Vakili and Olkhovskaya (2023), which uses a slightly different eigenvalue decay condition. The paragraph following the definition of this eigenvalue decay condition does not say anywhere that $\nu > 1/2$ is required. Therefore, we believe that this bound on the information gain holds for any positive value of $\nu$.
>
> Sattar Vakili and Julia Olkhovskaya, Kernelized Reinforcement Learning with Order Optimal Regret Bounds, NeurIPS 2023
>
> *Kernel-specific result for the squared exponential kernel?*
>
> Yes, one can use Lemma 29 in Curi et al. (2020), which states that $\gamma\_T(R) = O(R^d \log^{d+1}(T))$ for the squared exponential kernel with $d$-dimensional inputs. If we set $d = d\_s + d\_a$ and plug this into the regret bound in Theorem 4.8, we obtain a regret bound of the order $O(H^{3/2}d^{(d+2)/4}T^{1/2}\log^{(3d+4)/4}(T))$.
>
> *The regret upper bound of Bayrooti et al. (2025b) shown in Table 1 seems to be tighter than the proposed one.*
>
> It is correct that the regret bound of Bayrooti et al. (2025b) has better dependence on $H$. However, that work considers deterministic MDPs (i.e. there is no noise added to the states). This makes it easier to estimate the transition kernel of the MDP, and so one would expect to get better regret bounds. Given recent results for noiseless GP bandits (Iwazaki, 2025), one might even expect that constant or logarithmic (in $T$) regret bounds are possible for deterministic MDPs.
>
> Shogo Iwazaki, Gaussian Process Upper Confidence Bound Achieves Nearly-Optimal Regret in Noise-Free Gaussian Process Bandits, NeurIPS 2025

---

> > ### Author Rebuttal · Reviewer_Koqh · 2026-04-02
> >
> > I appreciate the detailed reply.
> > Most concerns are resolved.
> > Thus, I will raise the score to 4.
> >
> > On the other hand, I describe additional comments below:
> >
> > >Discussion on [1] is required.
> >
> > I agree that there are several differences.
> > I believe that adding this discussion will improve the clarity regarding the paper's position.
> > I'm sorry for adding a new point, but the following paper [1], which studies level set estimation rather than BO, also relies on the idea of chaining.
> > Please consider the discussion of this paper, too, though I won't reduce the score for this new point.
> >
> > [1] Shubhanshu Shekhar, Tara Javidi, Multiscale Gaussian Process Level Set Estimation, Proceedings of the 22nd International Conference on Artificial Intelligence and Statistics, 2019.
> >
> > >Bayesian vs frequentist setting
> >
> > I conjecture that the authors call the expectation of regret "Bayesian regret".
> > However, in the frequentist setting where $f$ belongs to an RKHS, I believe the term ``expected regret'' is preferable, since it is not a Bayesian approach.
> > Indeed, Kakade et al. (2020) state, "While we focus on the frequentist regret bounds, we conjecture that a Bayesian regret bound..."
> >
> > > Presentation regarding Problems
> >
> > I probably understand the paper's position from the authors' reply.
> > Then, I still believe that the current descriptions are misleading.
> > Please consider fixing it adequately.
> > That is, please revise so that it is clear that there were prior studies that incorporate unbounded state spaces (problem 1) and limiter priors (problem 2), respectively, and that the proposed analyses are tighter (not necessarily optimal) than the existing results (problem 2).
> > At least, I could not understand these points from the current paper.
> > I believe these points are important for understanding the paper's novelty correctly.
> >
> > >This result also shows that (for GP bandits) a Bayesian regret bound of the order $\sqrt{dT} \gamma\_T^p$ is not possible for any $p < 1/2$.
> >
> > Why can we show this?
> > Can the argument of the linear bandit be extended to any kernels?
> > In addition, why can we obtain the lower bound regarding $\gamma\_T$ regardless of the fact that Rusmevichientong and Tsitsiklis (2010) did not discuss $\gamma\_T$?
> > If you add this sentence, please consider adding rigorous justification.

---

> > > ### Author Response · Authors · 2026-04-06
> > >
> > > Thank you for your reply and for agreeing to raise your score. We're pleased to hear that most of your concerns have been resolved. We address your additional comments below.
> > >
> > > *Discussion on [1]*
> > >
> > > Thank you for bringing this work to our attention. We agree that [1] could fit into a paragraph on previous uses of chaining in sequential Bayesian optimization/estimation problems (or sequential GP-based optimization/estimation under weak kernel smoothness assumptions). We will add such a paragraph to our related work section.
> > >
> > > *Bayesian vs frequentist*
> > >
> > > We agree that it is preferable to use the terms Bayesian regret and frequentist (or worst-case) regret to distinguish between regret bounds (in probability or in expectation) that hold when $f^{\star}$ is a random draw from a GP or some other prior (Bayesian) and regret bounds that hold when $f^{\star}$ is a fixed function in (say) a ball of an RKHS (frequentist). We will update the writing to reflect this.
> > >
> > > *Presentation regarding problems*
> > >
> > > Thank you for highlighting the lack of clarity in the positioning of our work. We agree that when we describe problems 1-3 in the introduction, this could reasonably be interpreted as a statement that we are solving each of these problems for the first time, whereas some of these problems have already been addressed individually. Our contribution is a regret analysis for GP-PSRL that addresses all three problems simultaneously (which we believe is new). We propose to add a sentence or two to the introduction that clarifies this explicitly.
> > >
> > > *Lower bound*
> > >
> > > Suppose we have a regret bound of the order $\sqrt{dT}(\gamma\_T)^{p}$, with $p < 1/2$, that holds for any (sufficiently smooth) kernel. In the special case where $H = 1$ and the kernel is the linear kernel, this gives a regret bound of the order $\sqrt{dT}(d\log T)^{p}$ for Bayesian linear bandits. One can then choose $d$ and $T$ to be large enough such that this upper bound is less than the $d\sqrt{T}$ lower bound for Bayesian linear bandits from Rusmevichientong and Tsitsiklis (2010). Therefore, an upper bound of the order $\sqrt{dT}(\gamma\_T)^{p}$ that holds for every kernel is not possible. Of course, this does not mean that $\sqrt{dT\gamma\_T}$ is necessarily the best possible rate for every kernel, just that it is not possible to uniformly (w.r.t. the choice of kernel) improve upon the rate of $\sqrt{dT\gamma\_T}$. Indeed, this does not rule out the possibility of improving the regret bound by replacing $\gamma\_T$ by a "less worst-case" quantity (which coincides with $\gamma\_T$ for the linear kernel) as done in [3] for the bandit setting.

---

### Official Review · Reviewer_k2yr · 2026-03-12

**Soundness:** 4
**Presentation:** 4
**Significance:** 3
**Originality:** 3
**Overall Recommendation:** 5
**Confidence:** 2

**Summary:**

This paper addresses a gap in the theoretical analysis of Gaussian process posterior sampling reinforcement learning. The setting they analyze is a finite-horizon episodic MDP with continuous, unbounded state space with Gaussian transition noise, and a GP prior over the unknown dynamics. The paper's contribution addresses two primary limitations in prior theoretical work characterizing GP-PSRL by doing the following: 1) handling unbounded state spaces, and 2) obtaining optimal dependence on the maximum information gain. The main result is deriving a Bayesian regret bound resolving prior limitations, with a proof that shows that the states visited by GP-PSRL are actually bounded even if the possible state space is unbounded, and given this the regret can be bounded by decomposition into model estimation error terms. A small experiment supports the theoretical findings.

**Compliance With Llm Reviewing Policy:**

Affirmed.

**Final Justification:**

Maintaining my positive score.

**Key Questions For Authors:**

N/A

**Limitations:**

yes

**Strengths And Weaknesses:**

The paper is technically strong and well written - the intuition for each part of the proof is explained in the main text instead of solely relying on the appendix so the reader can follow the thought process behind the construction of the final regret bound. The result itself is significant, as the authors address the primary limitations of previous approaches. The related work section comprehensively places this work in the context of the field.

I am not familiar enough with the background topic to comment on the correctness of the proofs. Taking the main text at face value, it is overall a strong contribution.

---

> ### Author Rebuttal · Authors · 2026-03-31
>
> Thank you for your kind comments and for your positive assessment of our work.

---

> > ### Author Rebuttal · Reviewer_k2yr · 2026-04-02
> >
> > No resolution needed

---

> > > ### Author Response · Authors · 2026-04-06
> > >
> > > Thank you for your answer. We’re happy to hear that your impression of our work remains positive.

---

### Official Review · Reviewer_TwSS · 2026-03-21

**Soundness:** 4
**Presentation:** 3
**Significance:** 4
**Originality:** 4
**Overall Recommendation:** 5
**Confidence:** 4

**Summary:**

The paper considers a Markov decision process (MDP) in a Bayesian setting, with transition dynamics drawn from a Hölder-continuous kernel. In the work, the reward function is assumed to be known and fixed. Under this setting, the authors derive sublinear regret bounds for an unbounded state-action space, which is otherwise frequently assumed in theoretical RL considerations. The proof idea is to show inductively that the state-action walk remains within a polylogarithmic ball, thereby essentially removing the hardness of the unbounded state-action space.

**Compliance With Llm Reviewing Policy:**

Affirmed.

**Final Justification:**

I believe it's an interesting paper with a few novel approaches to well established problems, I vote for its acceptance.

**Key Questions For Authors:**

(1) Please elaborate on several comparisons with prior works raised earlier.

(2) A question that also ties into question (1) is the assumption that the reward function is fixed (deterministic) and known. This essentially allows for "offline" policy design before each episode. Basically, the reward does not need to be learned, and the algorithm bypasses the problematic backtracking regression argument typically seen in deterministic MDPs (Jin et al., 2020). In RKHS, this is especially problematic as it requires a covering bound on the space of value functions whose norm scales with the information gain, and this covering number bound directly dictates the confidence intervals.  For example, in Yang et al. (2020), this leads to the regret scaling as $\mathcal{O}(T^{\frac{2d}{\nu}})\gg 1$ for high-dimensional spaces and thus can lead to trivial regret bounds.

While incorporating a deterministic, unknown reward function (just a member of a RKHS) might be quite difficult, is there a way to allow for $r$ to be sampled from a GP? Would the analysis and the regret order in the paper remain? Is knowing the reward function a usual assumption in Bayesian MDP's?

Given that Bayrooti et al. (2025b) do not assume bounded rewards (I believe they are indeed sampled from a GP), I don't think the comparison between the two works is quite fair.



*Jin, Chi, Zhuoran Yang, Zhaoran Wang, and Michael I. Jordan. "Provably efficient reinforcement learning with linear function approximation." In Conference on learning theory, pp. 2137-2143. PMLR, 2020.*

*Yang, Zhuoran, Chi Jin, Zhaoran Wang, Mengdi Wang, and Michael I. Jordan. "On function approximation in reinforcement learning: Optimism in the face of large state spaces." arXiv preprint arXiv:2011.04622 (2020).*

**Limitations:**

Yes

**Strengths And Weaknesses:**

Soundness: I found the paper sound, and the outline of ideas natural and easy to follow. The appendix is also well written and allows for a broad overview of the proof. The definition of $\tilde{c}$ auxiliary kernel to handle the ball discretization is new and interesting idea with broader applications.

Presentation: The outline of the proof and its motivation are well written and make for an easy read on a technical topic. The part outlining the difficulty of establishing estimation bounds in the current Thompson sampling literature and emphasizing the need for a very smooth kernel function (which essentially kills applicability for nearly all "interesting" Matern kernels) is especially appreciated. In my mind, this is the main contribution of the paper and should be better highlighted in both the abstract and the related work section, when compared to other works in GP RL. The ability to go from $C^4\left(\mathbb{R}^{2(d_s+d_a)}\right)$ to just Holder continuity for kernel functions certainly fills a gap and should be the main takeaway from the paper.

The comparison to previous works does need some refinement, in my opinion. As it currently stands, I am not sure which works exactly assumes bounded state- action space and what level of kernel regularity is needed. For example, in Bayrooti et al. (2025b), is the ambient space bounded? If it is, the only benefit is the removal of the Lipschitz condition on the value function.

What is the benefit over the work of Curi et al., (2020)? From the table it seems the regret order are the same and the kernel requirements match., Is $\beta_T(\delta)$ a confidence width and thus a function of $\gamma_T$? What is N here?
Given that the main point of the work is to improve the current literature, a more complete and detailed comparison is needed.

A minor presentation issue: In chaining inequality, $\mathcal{N}$ appears before being defined and it's not clear that it is the covering number.

Significance: The work is significant both for the novelty of the techniques used and the improvement over the current literature.

Originality: As outlined before there are several novelties in the paper namely the chaining method to bound the diameter the state-action walk could reach and defining the "new" kernel function that's piece-wise Holder to better take care of discretization.

---

> ### Author Rebuttal · Authors · 2026-03-31
>
> Thank you for your comments and for your positive assessment of our work. We will fix the presentation issues as suggested. We address the comments under "strengths and weaknesses" and "questions" below.
>
> *For example, in Bayrooti et al. (2025b), is the ambient space bounded? If it is, the only benefit is the removal of the Lipschitz condition on the value function.*
>
> In the setting studied by Bayrooti et al. (2025b), the set of possible states is unbounded, and this is not considered in their regret analysis (as far as we can tell). In their setting, each state is given by a function $f$ of the previous state and action (without any added noise). Since $f$ is a random draw from a GP, the states are random variables with unbounded support. The required kernel regularity conditions are not obvious to us. Bayrooti et al. (2025b) assume that all value functions have bounded first and second derivatives, which is presumably only true under some additional regularity conditions.
>
> *What is the benefit over the work of Curi et al., (2020)? Is $\beta\_T$ a confidence width and thus a function of $\gamma\_T$? What is $N$ here?*
>
> One of the benefits of our work over the work of Curi et al. (2020) is a better regret bound, though since Curi et al. (2020) consider the worst-case regret, as opposed to the Bayesian regret, the two regret bounds are not directly comparable. Yes, $\beta\_T$ is a confidence width that depends on $\gamma\_T$. If we plug in the expression for $\beta\_T$, then the regret bound for HUCRL is of the order $O(H^2\gamma\_T^{(H+1)/2}\sqrt{T})$. $N$ is used by Curi et al. (2020) to denote the horizon, which we denote by $H$. We will fix this.
>
> In addition, GP-PSRL is a more practical algorithm than HUCRL. The planning step of HUCRL requires one to find an optimal policy for an augmented MDP with an action space of higher dimension. We would struggle to implement HUCRL in our experimental setup as the memory requirement of the discretized action space would be 2000 times larger. In contrast, the planning step of GP-PSRL requires one to find an optimal policy for an ordinary MDP sampled from the posterior. This may still be expensive, but it is easier than planning in the augmented MDPs used by Curi et al. (2020).
>
> *...is there a way to allow for the reward function to be sampled from a GP? Would the analysis and the regret order in the paper remain? Is knowing the reward function a usual assumption in Bayesian MDP's?*
>
> Yes, the reward function could be modeled as a sample from a GP. The regret analysis would not change drastically, but would be more complicated. The need to estimate the reward function as well as the transition kernel would lead to an additive lower order term in the regret bound. Intuitively, this is because the reward function is determined by a single random draw from a GP, whereas the transition kernel is determined by $d\_s$ random draws from a GP, and is therefore harder to estimate. For this reason, it is quite common to assume that the reward function is known.
>
> The bounded reward assumption is used to upper bound the value function of any policy, which is still possible if the reward is drawn from a GP. Using the same argument as in Lemma 4.4, one could show that the states generated by running any policy for $H$ steps are contained within a ball with high probability. One could then upper bound the value function of any policy using an upper bound on the supremum of the reward function over this ball (via chaining, for instance). We expect that this bound on the value function would be of the order $H\sqrt{(d\_s+d\_a)\log(T)}$ (as opposed to $H$), so the final regret bound would pick up an additional factor of $\sqrt{(d\_s+d\_a)\log(T)}$. We also use the bounded reward assumption to control the regret under the bad event. This could be circumvented by proving a high-probability regret bound, and then integrating the resulting tail bound for the regret to get a regret bound in expectation.
>
> *Given that Bayrooti et al. (2025b) do not assume bounded rewards, I don't think the comparison between the two works is quite fair.*
>
> We agree that a fair comparison is difficult. On the one hand, our assumption that the reward function is known and bounded makes some parts of our analysis simpler, though we believe that it does not make the problem much easier (in the sense that it would allow one to prove a much better regret bound than would otherwise be possible). On the other hand, Bayrooti et al. (2025b) assume that the transition kernel is deterministic (i.e. no noise is added to the states), which makes it easier to estimate the transition kernel. Based on recent results for noiseless GP bandits (Iwazaki, 2025), one might expect that constant or logarithmic (in $T$) regret bounds are possible for reinforcement learning in deterministic MDPs.
>
> Shogo Iwazaki, Gaussian Process Upper Confidence Bound Achieves Nearly-Optimal Regret in Noise-Free Gaussian Process Bandits, NeurIPS 2025

---

> > ### Author Rebuttal · Reviewer_TwSS · 2026-04-02
> >
> > Although it's not entirely clear where the paper fits within the relevant literature, I still find its theoretical novelty sufficient to vote for acceptance. The "discrete kernel" trick to avoid excessive smoothness in the kernel function is entirely new to me, even though the chaining trick might not be.

---

> > > ### Author Response · Authors · 2026-04-06
> > >
> > > Thank you for your answer. We’re happy to hear that your concerns have been resolved. Regarding the position of our work within the literature, we propose to add a sentence or two to the introduction stating that some of the three problems listed in the introduction have already been addressed individually and that the gap the we fill is a regret analysis of GP-PSRL that simultaneously addresses all three of these problems.

---

### Decision · Program_Chairs · 2026-04-30

**Decision:**

Accept (regular)

**Comment:**

The paper studies GP-based posterior sampling for finite-horizon reinforcement learning with continuous, unbounded state spaces, and derives a Bayesian regret bound that improves prior analyses in its dependence on information gain while explicitly handling the unbounded-state issue. Reviewers viewed the paper as technically strong and generally well presented, with the main strengths being the proof strategy for controlling the visited state region and the use of chaining and related Gaussian-process tools to obtain sharper regret guarantees.

The main concerns were about positioning relative to prior work, especially around which limitations had already been addressed separately in the literature, as well as some claims in the original framing that sounded stronger than what is currently established. These concerns were largely addressed in the rebuttal: the authors clarified the comparison to prior work, agreed to soften claims such as “tight” or “near-optimal,” and committed to revising the introduction and related work for accuracy and clarity. Overall, the paper makes a meaningful theoretical contribution, and I recommend acceptance.